# Mysteriously high $\Delta^{14}$C of the glacial atmosphere: Influence of $^{14}$C production and carbon cycle changes

Ashley Dinauer[1,*], Florian Adolphi[1,2], Fortunat Joos[1]

[1]Climate and Environmental Physics, Physics Institute and Oeschger Centre for Climate Change Research, University of Bern, Sidlerstrasse 5, 3012 Bern, Switzerland

[2]Quaternary Sciences, Department of Geology, Lund University, Sölvegatan 12, 22362 Lund, Sweden

*To whom correspondence should be addressed. Email: ashley.dinauer@climate.unibe.ch

**Abstract.** Despite intense focus on the ~190 permil drop in atmospheric $\Delta^{14}$C during Heinrich Stadial 1 ~17.4 to 14.6 kyr BP, the specific mechanisms responsible for the apparent $\Delta^{14}$C excess in the glacial atmosphere have received considerably less attention. The computationally efficient Bern3D earth system model of intermediate complexity, designed for long-term climate simulations, allows us to address a very fundamental but still elusive question concerning the atmospheric $\Delta^{14}$C record: How can we explain the persistence of relatively high $\Delta^{14}$C values during the millennia after the Laschamp event? Large uncertainties in the pre-Holocene $^{14}$C production rate, as well as in the older portion of the $\Delta^{14}$C record, complicate our qualitative and quantitative interpretation of the glacial $\Delta^{14}$C elevation. Here we begin with sensitivity experiments that investigate the controls on atmospheric $\Delta^{14}$C in idealized settings. We show that the interaction with the ocean sediments may be much more important to the simulation of $\Delta^{14}$C than had been previously thought. In order to provide a bounded estimate of glacial $\Delta^{14}$C change, the Bern3D model was integrated with five available estimates of the $^{14}$C production rate as well as reconstructed and hypothetical paleoclimate forcing. Model results demonstrate that none of the available reconstructions of past changes in $^{14}$C production can reproduce the elevated $\Delta^{14}$C levels during the last glacial. In order to increase atmospheric $\Delta^{14}$C to glacial levels, a drastic reduction of air-sea exchange efficiency in the polar regions must be assumed, though discrepancies remain for the portion of the record younger than ~33 kyr BP. We end with an illustration of how the $^{14}$C production rate would have had to evolve to be consistent with the $\Delta^{14}$C record, by combining an atmospheric radiocarbon budget with the Bern3D model. The overall conclusion is that the remaining discrepancies with respect to glacial $\Delta^{14}$C may be linked to an underestimation of $^{14}$C production and/or a biased-high reconstruction of $\Delta^{14}$C over the time period of interest. Alternatively, we appear to still be missing an important carbon cycle process for atmospheric $\Delta^{14}$C.

## 1 Introduction

The cosmogenic radionuclide radiocarbon ($^{14}$C) is a powerful tracer for the study of several ocean processes including deep ocean circulation and ventilation. Past changes in atmospheric $^{14}$C/C (i.e., $\Delta^{14}C_{atm}$, in permil; corresponding to $\Delta$ from Stuiver and Polach, 1977), as recorded in absolutely dated tree rings, plant macrofossils, speleothems, corals,

and foraminifera, have been interpreted as possibly reflecting real changes in the ocean's large-scale overturning circulation (Siegenthaler et al., 1980). The extended 54,000-year record of $\Delta^{14}C_{atm}$ from the latest IntCal compilation (i.e., IntCal13; Reimer et al., 2013) and from two Hulu Cave stalagmites (Cheng et al., 2018), adjusted to the presently accepted value of the radiocarbon half-life of 5700 years (Audi et al., 2003; Bé et al., 2013), suggests that large millennial-scale variations in $\Delta^{14}C_{atm}$ have occurred during the last glacial, compared to the relatively small (~30 ppm) change in atmospheric $CO_2$ over the same time period (Fig. 1). When interpreting the implications of such changes, it is important to note that $\Delta^{14}C_{atm}$ is controlled not only by global carbon cycle processes but also by variations in the atmospheric $^{14}C$ production rate. Therefore, the use of $\Delta^{14}C_{atm}$ as an indicator of past oceanic conditions, particularly those associated with air-sea exchange efficiency and deep ocean ventilation rates, requires reliable estimates of the $^{14}C$ production rate over time.

The vast majority of all $^{14}C$ production changes are the result of either solar or geomagnetic modulation of the cosmic ray flux reaching the Earth (Masarik and Beer, 1999; Poluianov et al., 2016). Fig. 1 shows several different proxy records of the global production rate of $^{14}C$ in relative units covering the full range of the $^{14}C$ dating method, based on geomagnetic field data from marine sediments (Laj et al., 2000; Laj et al., 2004; Nowaczyk et al., 2013; Channell et al., 2018) and on $^{10}Be$ and $^{36}Cl$ measurements in polar ice cores (Adolphi et al., 2018). A fundamental difference between these reconstruction methods is that paleointensity-based estimates of the $^{14}C$ production rate, by definition, do not reflect changes in the solar modulation of the cosmic radiation, whereas ice-core $^{10}Be$-based estimates give the combined influence of solar and geomagnetic modulation on radionuclide production. Of note is the striking coherence in all three records $\Delta^{14}C_{atm}$, paleointensity-based production, and ice-core $^{10}Be$-based production) of the Laschamp excursion (~41 kyr BP), when the Earth's geomagnetic dipole field briefly reversed and its intensity was close to zero (Nowaczyk et al., 2012; Laj et al., 2014). According to reconstructions and production rate models, this large geomagnetic event caused a doubling of the $^{14}C$ production rate, leading to the highest $\Delta^{14}C_{atm}$ values over the last 54 kyr. Relatively high $\Delta^{14}C_{atm}$ values continued until ~25 kyr BP, then gradually diminished to preindustrial levels, interrupted by a sharp drop in $\Delta^{14}C_{atm}$ during Heinrich Stadial 1 (HS1) ~17.4 to 14.6 kyr BP (sometimes called the "mystery interval"; Broecker and Barker, 2007). While the Laschamp geomagnetic excursion appears to be responsible for the $\Delta^{14}C_{atm}$ peak at ~41 kyr BP, the production rate estimates during much of the pre-Holocene period are subject to considerable uncertainty.

Paleointensity-based reconstructions are sensitive to coring disturbances of poorly consolidated sediments. The last 50 kyr are represented by the relatively slushy uppermost few meters of recovered marine sediment cores (Channell et al., 2018). Channell et al. (2018) preferentially selected cores recovered using conventional piston and square barrel gravity coring methods, and from sites with high mean (> 15 cm $kyr^{-1}$) sedimentation rates, so as to minimize the influence of drilling disturbance, and reached very different production rates than, e.g., Laj et al. (2000). On the other hand, ice-core $^{10}Be$ records are affected by changes in the transport and deposition of atmospheric $^{10}Be$, which may overprint the production rate changes (e.g., Heikkilä et al., 2013). Furthermore, in order to calculate the ice-core $^{10}Be$ deposition fluxes, snow accumulation rates must be known for each specific ice core, which themselves

have uncertainties on the order of 10 to 20 percent that propagate into the ice-core [10]Be fluxes (Gkinis et al., 2014;
Rasmussen et al., 2013). The large uncertainties associated with the reconstruction of past changes in [14]C production
hamper our ability to predict reliably the extent to which production changes contributed to high glacial $\Delta^{14}C_{atm}$ levels.
Only if estimates of past changes in [14]C production are robust can one improve assessments of the relative importance
of the two fundamental mechanisms responsible for glacial-interglacial $\Delta^{14}C$ changes: (1) production changes and (2)
carbon cycle changes.

Earlier model studies have focused heavily on the ~190 permil drop in $\Delta^{14}C_{atm}$ during HSI and on the deglacial

trends in $\Delta^{14}C_{atm}$ after HS1 (Muscheler et al., 2004; Broecker and Barker, 2007; Skinner et al., 2010; Mariotti et al.,
2016; Delaygue et al., 2003; Marchal et al., 2001; Huiskamp and Meissner, 2012; Hain et al., 2014). Historically, the
younger portion of the $\Delta^{14}C_{atm}$ record has received more attention than the glacial section because of the early emphasis
on the general climatic trends of the North Atlantic stadials (HS1 and the Younger Dryas [YD]) and the Bølling-
Allerød (BA) warm period, and on the important role of an exceptionally aged ([14]C-depleted) deep-water mass in the
pulsed rise of atmospheric $CO_2$ during the last glacial termination (e.g., Skinner et al., 2017). Less research over the
last few decades has studied the specific mechanisms responsible for high glacial $\Delta^{14}C_{atm}$ levels. The model studies
that are available point out the difficulties in simulating the correct glacial $\Delta^{14}C_{atm}$ levels (Hughen et al., 2004; Köhler
et al., 2006). These studies demonstrate with box models that glacial levels of $\Delta^{14}C_{atm}$ cannot be attained without
invoking significant changes in ocean circulation, air-sea gas exchange, and carbonate sedimentation. However, the
box models were not able to reproduce $\Delta^{14}C_{atm}$ values higher than 700 permil, and these results still need to be
scrutinized with models of higher complexity. To our knowledge, no three-dimensional ocean biogeochemical model
has yet simulated the 50,000-year record of $\Delta^{14}C_{atm}$. Many questions remain unanswered, in particular: What
mechanism can account for the persistence of relatively high $\Delta^{14}C_{atm}$ values during the millennia after the Laschamp
excursion?

The expected time scale for sustaining elevated levels of $\Delta^{14}C_{atm}$ after a production peak is on the order of

thousands of years, a time scale tied to the mean lifetime of [14]C (~8223 years; Audi et al., 2003; Bé et al., 2013) and
the time required for deep ocean ventilation (on the order of 1000 years or more; Primeau, 2005). Specifically,
Muscheler et al. (2004) demonstrate that the characteristic time constant for equilibration of $\Delta^{14}C_{atm}$ after a
perturbation in atmospheric production is 5000 years. By this analysis, the Laschamp event, which lasted only about
1500 to 2000 years (Laj et al., 2000), was insufficient to sustain the high $\Delta^{14}C_{atm}$ values observed over the next ~15,000
years. The lack of significant changes (only ~10 percent) in atmospheric $CO_2$ during the time period of interest raises
the question of what causes variations in $\Delta^{14}C_{atm}$, but not $CO_2$, on millennial time scales? The obvious answers are:
cosmic ray modulation and air-sea gas exchange. Ultimately, no explanation for high glacial $\Delta^{14}C_{atm}$ levels can be
complete in the absence of more robust estimates of the pre-Holocene [14]C production rate, as well as a good
understanding of the ocean carbon cycle under glacial climate conditions.

One of the major challenges associated with modelling glacial-interglacial climate cycles is that it is currently
not possible to reproduce climate and atmospheric $CO_2$ variations on the basis of orbital forcing alone. Problems
include the complexity of the Earth system, making it difficult to represent all the relevant processes in models, and
the long time scales involved, making simulations covering tens of thousands of years costly in computation time.
Glacial-interglacial simulations with dynamic ocean and land models of intermediate complexity have begun to
emerge, but these models are not yet able to reproduce the reconstructed variations in important proxy data or the
timing of $CO_2$ variations during the last glacial termination (Brovkin et al., 2012; Ganopolski and Brovkin, 2017;
Menviel et al., 2012). A wide variety of mechanisms, both physical and biological, centered on or connected with the
ocean, as well as exchange processes with the land biosphere, marine sediments, coral reefs, and the lithosphere, are
thought to play a role in explaining the glacial-interglacial variations in atmospheric $CO_2$ (Archer et al., 2000; Fischer
et al., 2010; Wallmann et al., 2016; Galbraith and Skinner, 2020), but how they interacted over time under the influence
of orbital forcing remains elusive. We appear to still be missing a single framework in which these mechanisms are
linked to each other in a predictable manner. As long as there are still large gaps in our understanding of the glacial
climate and associated ocean carbon cycle, a convenient way to examine the impact of the possible mechanisms on
atmospheric $CO_2$ levels, and here on $\Delta^{14}C_{atm}$, is to perform sensitivity experiments and scenario-based simulations
with models. This allows us to investigate specific phenomena in idealized settings, permitting us to investigate in
detail which parameters and processes are most important in controlling $\Delta^{14}C_{atm}$ levels.

In this paper we extend previous modelling efforts concerning the record of $\Delta^{14}C_{atm}$ with respect to three
issues: (1) the sensitivity of the $\Delta^{14}C_{atm}$ response to carbon cycle changes and the potential importance of marine
sediments, (2) the simulation of $\Delta^{14}C_{atm}$ covering the time range of the IntCal13 radiocarbon calibration curve (50,000
years), the primary focus being the explanation of high glacial $\Delta^{14}C_{atm}$ levels, and (3) a new 50,000-year record of the
$^{14}C$ production rate, as inferred by deconvolving the reconstructed histories of $\Delta^{14}C_{atm}$ and $CO_2$ with a prognostic
carbon cycle model and considering the uncertainties associated with the glacial-interglacial ocean carbon cycle. In
the following sections we first introduce the Bern3D earth system model of intermediate complexity and describe the
carbon cycle scenarios for forcing it. We then use step changes in the $^{14}C$ production rate and in selected parameters
of the ocean carbon cycle model to gain insight into the transient and equilibrium response of $\Delta^{14}C_{atm}$. After these
sensitivity experiments we present the results of paleoclimate simulations forced by available reconstructions of past
changes in $^{14}C$ production together with reconstructed and hypothetical carbon cycle changes accompanying glacial-
interglacial climate cycles. Finally, we present results for a first attempt to reconstruct the glacial history of the $^{14}C$
production rate using the Bern3D model forced with reconstructed variations in $\Delta^{14}C_{atm}$ and $CO_2$ as well as a wide
range of carbon cycle scenarios. We end with a comparison of three fundamentally different (model-based,
paleointensity-based, and ice-core $^{10}Be$-based) reconstructions of atmospheric $^{14}C$ production.

**2 Materials and methods**

**2.1 Brief description of the Bern3D model**

Simulations are performed with the computationally efficient Bern3D earth system model of intermediate complexity
(version 2.0), which is designed for long-term climate simulations over several tens of thousands of years. The Bern3D
couples a frictional geostrophic 3-D ocean general circulation model (Edwards et al., 1998; Edwards and Marsh, 2005;
Müller et al., 2006), a 2-D energy-moisture balance atmosphere model (Ritz et al., 2011), an ocean carbon cycle model
(Müller et al., 2008; Tschumi et al., 2008; Parekh et al., 2008), a chemically active 10-layer ocean sediment model
(Heinze et al., 1999; Tschumi et al., 2011; Roth et al., 2014; Jeltsch-Thömmes et al., 2019), and a four-box model
representing carbon stocks in the terrestrial biosphere (Siegenthaler and Oeschger, 1987). The coarse-resolution ocean
model is implemented on a 41 x 40 horizontal grid, with 32 logarithmically spaced layers in the vertical. The seasonal
cycle is resolved with 96 time steps per year. The tracers carried in the ocean model include temperature, salinity,
dissolved inorganic carbon (DIC), dissolved organic carbon (DOC), carbon isotopes ([14]C and [13]C) of DIC and DOC,
alkalinity (Alk), phosphate (P), silicate (Si), iron, dissolved oxygen ($O_2$), preformed dissolved oxygen ($O_{2,pre}$), and an
"ideal age" tracer. The ideal age is set to zero in the surface layer, increased by $\Delta t$ in all interior grid cells at each time
step of duration $\Delta t$, and transported by advection, diffusion, and convection. Atmospheric $CO_2$, $^{14}CO_2$, and $^{13}CO_2$ are
also carried as tracers in the atmosphere model. For a more complete description of the Bern3D model, the reader is
referred to Appendix A.

**2.2 Implementation of the [14]C tracer**

Natural radiocarbon ([14]C) is a cosmogenic radionuclide produced in the atmosphere by cosmic radiation. Once
oxidized to $^{14}CO_2$, it participates in the global carbon cycle. Atmospheric $^{14}CO_2$ invades the ocean by air-sea gas
exchange, where it is subject to the same physical and biogeochemical processes that affect DIC. The only difference
is that [14]C is lost by radioactive decay (half-life of $5700 \pm 30$ years; Audi et al., 2003; Bé et al., 2013). The governing
natural processes, namely, atmospheric [14]C production, air-sea gas exchange, physical transport and mixing in the
water column, biological production and export of particulate and dissolved matter from the surface ocean, particle
flux through the water column, particle deposition on the sea floor, remineralization and dissolution in the water
column and the sediment pore waters, and vertical sediment advection and sediment accumulation, are explicitly
represented in the Bern3D model (see Fig. 2). Air-sea gas exchange is parameterized using a modified version of the
standard gas transfer formulation of OCMIP-2, with exchange rates that vary across time and space (see Appendix A
for more details).

Radiocarbon measurements are generally reported as $\Delta^{14}C$, i.e., the ratio of [14]C to total carbon C relative to
that of the AD 1950 atmosphere, with a correction applied for fractionation effects, e.g., due to gas exchange and
photosynthesis (see Stuiver and Polach, 1977). In this model study, $\Delta^{14}C$ is treated as a diagnostic variable using the
two-tracer approach of OCMIP-2. Rather than treating the [14]C/C ratio as a single tracer, fractionation-corrected [14]C is
carried independently from the carbon tracer. The modelled [14]C concentration is normalized by the standard ratio of
the preindustrial atmosphere ($^{14}r_{std} = 1.170 \times 10^{-12}$; Orr et al., 2017) in order to minimize the numerical error of
carrying very small numbers. For comparison to observations, $\Delta^{14}C$ is calculated from the normalized and
fractionation-corrected modelled $^{14}C$ concentration as follows:

$$\Delta^{14}C = 1000\left(^{14}r' - 1\right) \tag{1}$$

where $^{14}r'$ is the ratio of $^{14}C/C$ in either atmospheric $CO_2$ or oceanic DIC divided by $^{14}r_{std}$, depending on the reservoir
being considered. The approach taken to simulate atmospheric $^{14}CO_2$ is analogous to the approach used for $CO_2$,
except that the equation includes the terms due to atmospheric production and radioactive decay. For simulations
where the sediment model is active, the oceanic DIC tracer sees a constant input from terrestrial weathering, whereas
there is no weathering input of $DI^{14}C$ to the ocean (see Appendix A for more details).

In the preindustrial spin-up simulation needed to initialize the Bern3D model, atmospheric $CO_2$ is held
constant at 278.05 ppm and $\Delta^{14}C_{atm}$ at 0 permil. During this integration time the ocean inventories of carbon and $^{14}C$
adjust to the forcing fields. The resulting changes after >50,000 years of integration are negligibly small. Fig. 3 shows
the steady-state $\Delta^{14}C$ distribution in the surface (< 100 m) and deep (> 1500 m) ocean for the preindustrial control run.
The large-scale distribution of modelled oceanic $\Delta^{14}C$ broadly resembles the observed pattern in the Global Ocean
Data Analysis Project (GLODAP; Key et al., 2004). That final state (i.e., the end of the preindustrial spin-up) is used
to diagnose the $^{14}C$ production rate for the preindustrial atmosphere, such that the rate of $^{14}C$ production is balanced
by radioactive decay and the net fluxes out of the atmosphere. For transient simulations, an adjustable scale factor is
applied to the preindustrial steady-state value of 443.9 mol $^{14}C$ per year (1.66 atoms $cm^{-2}$ $s^{-1}$) in order to account for
production changes induced by solar and/or geomagnetic modulation. These production changes are derived from,
e.g., available reconstructions of the $^{14}C$ production rate in relative units, as detailed in Sect. 2.5. Note the preindustrial
spin-up results in steady-state values for weathering-derived inputs of DIC, Alk, P, and Si of 0.46 Gt C per year, 34.37
Tmol Alk per year, 0.17 Tmol P per year, and 6.67 Tmol Si per year, respectively. These terrestrial weathering rates
were chosen to balance the sedimentation rates on the sea floor and are held fixed and constant throughout the
simulations.

**2.3 Model configurations**

We focus in this paper on the response of $\Delta^{14}C_{atm}$ to changes in $^{14}C$ production and the ocean carbon cycle. For a
deeper mechanistic understanding of the driving processes, step response experiments are first performed (see Sect.
3.1). These simulations include perturbations of the steady-state $^{14}C/C$ distribution under preindustrial conditions. We
investigate the impact of step changes in (1) the $^{14}C$ production rate ("higher production" scenario), (2) wind stress
and vertical diffusivity ("reduced deep ocean ventilation" scenario), and (3) the gas transfer velocity ("enhanced
permanent sea ice cover" scenario). After a step change at time 0, the simulations are run to near-equilibrium over a
50,000-year integration. The following model configurations and therefore exchanging carbon reservoirs are
considered: atmosphere–ocean (OCN), atmosphere–ocean–land (OCN-LND), atmosphere–ocean–sediment (OCN-
SED), and atmosphere–ocean–land–sediment (ALL).

Next we examine the influence of changes that are transient in nature. We simulate $\Delta^{14}C_{atm}$ over the full range
of the $^{14}C$ dating method (i.e., 50 to 0 kyr BP) (see Sect. 3.2 and 3.3). These transient simulations are initialized at 70
kyr BP using model configuration ALL, and forced by reconstructed changes in $^{14}C$ production (see Sect. 2.5) over a
70,000-year integration. The first 20,000 years of the integration are considered a spin-up. Although the full record is
simulated, we focus our analysis on the millennial-scale variation in $\Delta^{14}C_{atm}$ before incipient deglaciation at ~18 kyr
BP. Eight model runs are carried out for each production rate reconstruction, using different combinations of forcing
fields and parameter values as described next.

**2.4 Carbon cycle scenarios**

In our transient simulations with the Bern3D model, eight scenarios based on different assumptions about the global
carbon cycle are considered, the details of which are summarized in Table 1. The goal is to investigate the extent to
which changes in the ocean carbon cycle could explain high glacial $\Delta^{14}C_{atm}$ levels, given available reconstructions of
past changes in $^{14}C$ production. We therefore consider a wide range of carbon cycle scenarios, including some extreme
cases. A note of caution. Because millennial-scale $\Delta^{14}C_{atm}$ variations during the last glacial are what we are interested
in, we do not attempt to reproduce abrupt climate perturbations such as Dansgaard-Oeschger warming events in the
model runs.

In the first scenario (MOD), the model is run with fixed preindustrial boundary conditions for the Earth's
orbital parameters, radiative forcing due to well-mixed greenhouse gases, and ice sheet extent. As a consequence,
atmospheric $CO_2$ remains approximately constant at the preindustrial level of 278.05 ppm over the simulation. The
second scenario (PAL) considers reasonably well-known climate forcing over the last glacial-interglacial cycle.
Simulations under this scenario are initialized with output from a previous spin-up simulation forced by glacial
boundary conditions with respect to orbital parameters (Berger, 1978), ice sheet extent (see below), and greenhouse
gas radiative forcing based on the smoothed dataset of atmospheric greenhouse gases by Köhler et al. (2017) as
constructed from the original data of Ahn and Brook (2014), Ahn et al. (2012), Bauska et al. (2015), Bereiter et al.
(2012), Buizert et al. (2015), Dlugokencky et al. (2016), Lourantou et al. (2010), Lüthi et al. (2010), MacFarling-
Meure et al. (2006), Marcott et al. (2014), Monnin et al. (2001, 2004), Rubino et al. (2013), Schneider et al. (2013),
and Sigl et al. (2016). In simulations under PAL, the model is integrated until 0 kyr BP following the reconstructed
histories of orbital forcing, ice sheets, and radiative forcing due to greenhouse gases. Ice sheets for the preindustrial
and Last Glacial Maximum (LGM) states are taken from Peltier (1994) and linearly scaled using the global benthic
$\delta^{18}O$ stack of Lisiecki and Stern (2016), which is a global ice volume proxy. Changes in the albedo, salinity and latent
heat flux associated with the ice sheet buildup or melting are also taken into account (Ritz et al., 2011). Note that,
although the radiative forcing for $CO_2$ is prescribed, the atmospheric $CO_2$ concentration is allowed to evolve freely,
except in the simulations described in Sect. 2.5.

Model scenario PAL appears to still be missing an important process or feedback for atmospheric $CO_2$, as it

cannot reproduce the observed low glacial $CO_2$ level without invoking additional changes (see, e.g., Tschumi et al.,
2011; Menviel et al., 2012; Roth and Joos, 2013; Jeltsch-Thömmes et al., 2019). Variations in atmospheric $CO_2$ govern
how fast $\Delta^{14}C$ signatures are passed between the atmosphere and ocean. Gross fluxes of $^{14}C$ between the atmosphere
and ocean, and vice versa, scale with atmospheric $pCO_2$ and its $^{14}C/C$ ratio. It is therefore important to reproduce low
glacial atmospheric $CO_2$ concentrations in at least some of the model scenarios, thereby capturing the influence of
temporal changes in $CO_2$ on the air-sea exchange of $^{14}C$. In this study, we consider six scenarios that invoke additional
changes to force the model toward the observed low glacial $CO_2$ concentration. In addition to the PAL forcing, a time-
varying scale factor $F(t)$ is applied to some combination of tunable model parameters: wind stress scale factor $\tau$,
vertical diffusivity $K_V$, gas transfer velocity $k_w$, $CaCO_3$-to-particulate organic carbon (POC) export ratio $rr$, and POC
remineralization length scale $\ell_{POC}$. For the preindustrial period, the value of $F(t)$ is fixed at 1, whereas the theoretical
LGM value was chosen in order to achieve an atmospheric $CO_2$ concentration close to the LGM level of ~190 ppm
(see Table 1), as determined by sensitivity experiments. Note that the same values of $F(t)$ apply to any of the model
parameters considered in a given scenario. To obtain intermediate values, $F(t)$ is linearly scaled using the global
benthic $\delta^{18}O$ stack (see Fig. 1). For the spin-up needed to initialize these simulations, the glacial spin-up simulation
of PAL was integrated for 50,000 model years, with tunable parameters adjusted to their appropriate glacial values.
Atmospheric $CO_2$ drawdown of up to ~100 ppm is achieved over this 50,000-year integration. From that final spun-
up state, the model is run forward in time until 0 kyr BP with PAL and $F(t)$ forcing.

The first of these scenarios (CIRC) allows us to test the sensitivity of the model results with respect to changes

in ocean circulation. Tunable model parameters $\tau$ and $K_V$ were reduced to 40 percent of their preindustrial values
throughout the global ocean during the LGM (i.e., $F_{\tau,K_V} = 0.4$). Such a drastic change in the wind stress field is not
realistic. Rather, these changes should be viewed as "tuning knobs" that force the ocean model into a poorly ventilated
state with an "older" ideal age and $^{14}C$-depleted deep waters, as suggested for the glacial ocean (e.g., Sarnthein et al.,
2013; Skinner et al., 2017). In the model's implementation, a change in wind stress does not affect the gas transfer
velocity $k_w$, unlike in the real ocean where changes in wind stress and wind speed act together. The influence of a
change in air-sea exchange efficiency on the model results was investigated in a second scenario (VENT) where $k_w$
is reduced in the model's north (> 60°N) and south (> 48°S) polar areas in addition to global reductions of $\tau$ and $K_V$
($F_{\tau,K_V,k_w} = 0.4$). A 60 percent reduction of $k_w$ is unlikely to be correct but is a straightforward way to reduce the
model's gas exchange efficiency. In the third scenario (VENTx), reduction of polar $k_w$ to 0 percent of its preindustrial
value was tested ($F_{\tau,K_V} = 0.4$; $F_{k_w} = 0.0$). Here, $k_w$ remains fixed at 0 percent during the last glacial and is adjusted
to its preindustrial value via a linear ramp across the last glacial termination (~18 to 11 kyr BP). In this scenario, sea
ice would permanently cover 100 percent of the Southern Ocean during the last glacial, which is not supported by the
sea ice reconstructions of Gersonde et al. (2005) and Allen et al. (2011), and also the high-latitude (> 60°N) North
Atlantic and Arctic Ocean, for which there is some evidence (Müller and Stein, 2014; Hoff et al., 2016).

We end by investigating the sensitivity of the model results to changes in the parameters controlling the

export production of $CaCO_3$ and the water column remineralization of POC. Model scenario BIO considers changes
of the $CaCO_3$-to-POC export ratio (and thus also the $CaCO_3$-to-POC rain ratio; Archer and Maier-Reimer, 1994)
($F_{rr} = 0.8$) and POC remineralization length scale (Roth et al., 2014) ($F_{\ell_{POC}} = 1.2$). These changes impact the global
carbon cycle by influencing the vertical gradients of DIC, Alk, and nutrients in the water column. A change in the
fluxes of POC and $CaCO_3$ to the sea floor drives a change in the magnitude of their removal by sedimentation on the
sea floor. A modest reduction in the export ratio during the last glacial is compatible with reconstructed variations in
carbonate ion concentrations (Jeltsch-Thömmes et al., 2019). How the depth of POC remineralization changed over
time is still unknown. The last two scenarios consider the combined effect of physical and biogeochemical changes:
PHYS-BIO ($F_{\tau,K_V,k_w,rr} = 0.7$) and PHYS-BIOx ($F_{\tau,K_V,k_w,rr} = 0.8$; $F_{\ell_{POC}} = 1.2$).

**2.5 Measurement- and model-based reconstruction of [14]C production**

Our ability to attribute past changes in $\Delta^{14}C_{atm}$ to climate-related changes in the ocean carbon cycle is limited by our
ability to reconstruct a precise and accurate history of the [14]C production rate. Past changes in [14]C production can be
estimated from geomagnetic field reconstructions and from [10]Be measurements in polar ice cores. For ice-core [10]Be-
based estimates, we use the ice-core radionuclide stack of Adolphi et al. (2018), which is based on [36]Cl data from the
GRIP ice core (Baumgartner et al., 1998), and on [10]Be data from the GRIP (Yiou et al., 1997; Baumgartner et al.,
1997; Wagner et al., 2001; Muscheler et al., 2004; Adolphi et al., 2014) and GISP2 (Finkel and Nishiizumi, 1997) ice
cores. It also includes [10]Be data from the NGRIP, EDML, EDC, and Vostok ice cores around the Laschamp
geomagnetic excursion (Raisbeck et al., 2017). It has been extended to the present using the [10]Be stack of Muscheler
et al. (2016). All ice cores were first placed on the same time scale (GICC05) before [10]Be fluxes were calculated. This
70,000-year [10]Be stack provides relative changes of [14]C production rates under the assumption that [14]C and [10]Be
production rates are directly proportional, as indicated by the most recent production rate models (e.g., Herbst et al.,

2017).


For paleointensity-based estimates, we employ (1) the North Atlantic Paleointensity Stack, or NAPIS, by Laj

et al. (2000) as extended by Laj et al. (2002), (2) the Global Paleointensity Stack, or GLOPIS, by Laj et al. (2004), (3)
a high-resolution paleointensity stack from the Black Sea (Nowaczyk et al., 2013), and (4) a paleointensity stack from
Iberian Margin sediments (Channell et al., 2018). In principle, stacks of widely distributed cores (NAPIS/GLOPIS)
are expected to yield a better representation of the global geomagnetic dipole moment, whereas the paleointensity
stacks from the Black Sea and the Iberian Margin avoid some of the problems associated with coring disturbances.
The four different paleointensity stacks were converted to [14]C production rates using the production rate model of
Herbst et al. (2017), the local interstellar spectrum of Potgieter et al. (2014), and assuming a constant solar modulation
potential of 630 MeV.

An alternative approach to estimating the $^{14}$C production rate is to combine an atmospheric radiocarbon
budget with a prognostic carbon cycle model. Here simulations are performed with the Bern3D model and forced by
reconstructed changes in $\Delta^{14}C_{atm}$ and $CO_2$, as well as reconstructed and hypothetical carbon cycle changes, over the
last 50 kyr. Both the IntCal13 calibration curve (Reimer et al., 2013) and the recent Hulu Cave $\Delta^{14}C_{atm}$ dataset (Cheng
et al., 2018) are used. Note that although the forthcoming IntCal20 calibration curve (Reimer et al., in press) will be
the new standard atmospheric radiocarbon record for the last 55,000 years, essentially all data underlying IntCal20
before 13.9 kyr BP are tied to the Hulu Cave dataset, either via time scales (Lake Suigetsu plant macrofossil data) or
marine reservoir corrections (marine records). Hence, the IntCal20 and Hulu Cave $\Delta^{14}C_{atm}$ records are very similar
and using IntCal20 would not impact our conclusions.

The $^{14}$C production rate $Q$ is calculated, each model year, from the air-sea $^{14}CO_2$ flux ($F_{as}$), the atmosphere-
land $^{14}CO_2$ flux ($F_{ab}$), the loss of $^{14}$C due to radioactive decay, and the change ($\dot{I}_a$) in the atmospheric $^{14}$C inventory
($I_a$):

$$Q = F_{as} + F_{ab} + \lambda I_a + \dot{I}_a \tag{2}$$

where $\lambda$ is the radioactive decay constant for $^{14}$C, i.e., $\lambda = \ln 2/5700$ years $= 1.2160 \times 10^{-4}$ yr$^{-1}$. The radioactive decay
term $\lambda I_a$ and the change in inventory $\dot{I}_a$ follow the reconstructed $\Delta^{14}C_{atm}$ and $CO_2$ records, whereas $F_{as}$ and $F_{ab}$ are
explicitly computed by the model. The $F_{as}$ term depends strongly on the carbon cycle scenario under consideration
(see Sect. 2.4 and Table 1). For comparison with other reconstructions, $Q$ is converted into a relative value by
normalizing it by the preindustrial value.

**3 Results and discussion**

**3.1 Atmospheric $\Delta^{14}$C response to step changes**

We use step changes in the $^{14}$C production rate, and in selected carbon cycle parameters, to gain insight into the
characteristic magnitude and time scale of the corresponding $\Delta^{14}C_{atm}$ changes (Fig. 4). Besides variations of the
production rate, changes in ocean circulation and air-sea gas exchange are considered the most important factors
affecting $\Delta^{14}C_{atm}$. Their effect on $\Delta^{14}C_{atm}$ can be understood in terms of their effect on the reservoir sizes involved in
the global carbon cycle and on the exchange rates between the reservoirs. We investigate the relative importance of
the major global carbon reservoirs (atmosphere, terrestrial biosphere, ocean, and sediments) by considering four
different model configurations (see Sect. 2.3), with particular emphasis on the role of marine sediments.

In model studies, the process of sedimentation (defined here as the difference between deposition and remineralization/dissolution of material on the sea floor) is often neglected because it is a relatively minor flux. In the Bern3D model, sedimentation removes only about 0.46 Gt C and 45.31 mol $^{14}$C per year in the preindustrial steady state. Indeed, the interaction with the ocean sediments has little influence on the global mean value of oceanic $\Delta^{14}$C and therefore $\Delta^{14}C_{atm}$, as long as the total oceanic amount of carbon remains approximately constant (Siegenthaler et al., 1980); however, this is not always true, particularly in the case of millennial-scale climate perturbations. This is demonstrated by the differences between the model runs with and without sediments (i.e., ALL versus OCN-LND, and OCN-SED versus OCN) as shown in Fig. 4. The response of $\Delta^{14}C_{atm}$ to various perturbations depends on the magnitude of the change in the ocean carbon inventory, with a larger change achieved by considering the interaction with the ocean sediments and the imbalance between weathering and sedimentation (see Fig. 5e,f). In order to facilitate our discussion, we will make only direct comparisons between model runs ALL and OCN-LND, which both include the four-box terrestrial biosphere model. We note that the $^{14}$C exchange rate between the atmosphere and the terrestrial biosphere is only of minor importance for long time scales of millennia and more.

**3.1.1 Change of $^{14}$C production**

At steady state, the relative change of $\Delta^{14}C_{atm}$ is equal to the relative change of the $^{14}$C production rate, irrespective of the individual reservoirs considered. Fig. 4 shows that $\Delta^{14}C_{atm}$ increases by about 100 permil (or 10 percent) when the production rate is increased by 10 percent. In model run ALL, $\Delta^{14}C_{atm}$ increases approximately exponentially to its new steady-state value with a characteristic time constant T of about 6170 years (i.e., $1 - 1/e \approx 63$ percent of the total change in $\Delta^{14}C_{atm}$ occurs within 6170 years). This e-folding time scale is close to the mean lifetime of $^{14}$C (~8223 years), which is modulated by the time required for $\Delta^{14}$C to equilibrate between the atmosphere and the ocean (i.e., the time scale for deep ocean ventilation, of the order of hundreds of years to 1000 years or more). In the next section, we will investigate the effect of ocean carbon cycle processes on $\Delta^{14}C_{atm}$.

Note that for simplicity, we investigated only step changes in atmospheric production, although, in reality, $^{14}$C production varies continuously over time due to changes in the solar and/or geomagnetic modulation of the cosmic radiation. This results in a non-steady state value of $\Delta^{14}C_{atm}$.

**3.1.2 Change of ocean circulation**

The exchange rate between the surface and deep ocean is mainly determined by physical transport and mixing processes. The overall effect of these processes is to transport $^{14}$C-enriched surface waters to the thermocline and deep ocean, where waters are typically $^{14}$C-depleted. In addition, the nutrient supply by transport and mixing plays an important role in determining the production and export of biogenic material from the surface ocean, constituting a second pathway for transporting $^{14}$C to the deep ocean.

In the Bern3D model, the tunable model parameters affecting the ventilation of the deep ocean include a
scale factor $\tau$ for the wind stress field and vertical diffusivity $K_V$. Fig. 4 shows the $\Delta^{14}C_{atm}$ response after a sudden
decrease of $\tau$ and $K_V$ by 50 percent. Although a halving of $\tau$ and $K_V$ does not represent a realistic change, the resulting
state of the ocean's large-scale overturning circulation can be interpreted in terms of the "ideal age" of water, which
represents the average time since a water mass last made surface boundary contact. The new steady-state ideal age
after a halving of $\tau$ and $K_V$ is almost three times greater than the preindustrial steady-state value (i.e., ~1664 years
versus ~613 years). This "ageing" of the ocean is achieved through a weakening and shoaling of the global meridional
overturning circulation as evident from a moderate reduction in the meridional overturning stream function for the
Indo-Pacific Ocean from about 14 to 9.5 Sv (1 Sv = $10^6$ m$^3$ s$^{-1}$), and a very strong reduction from about 18 to 8 Sv in
the Atlantic meridional overturning stream function, consistent with evidence for the glacial ocean. Here, as expected,
the overall effect of deep water ageing is a stronger vertical $\Delta^{14}C$ gradient in the water column and a subsequent
increase in $\Delta^{14}C_{atm}$. The exact nature of the $\Delta^{14}C_{atm}$ response, however, depends on the carbon reservoirs considered.
If the ocean sediment reservoir is neglected, the time required for $\Delta^{14}C_{atm}$ to adjust to step changes in $\tau$ and
$K_V$ is relatively short. $\Delta^{14}C_{atm}$ increases rapidly to its new steady-state value of ~159 permil, with a time constant T of
about 600 years. This increase of $\Delta^{14}C_{atm}$ can be explained by the fact that, owing to a weaker and shallower
overturning circulation, a comparatively large amount of carbon is moved from the atmosphere to the ocean. More
specifically, the atmospheric carbon inventory decreases by 14.6 percent, whereas the atmospheric $^{14}C$ inventory
decreases by only 1.1 percent (Fig. 5c). The $^{14}C$ being produced in the atmosphere is therefore diluted by a smaller
carbon inventory, increasing the atmospheric $^{14}C/C$ ratio; this asymmetry in the drawdown of $CO_2$ and $^{14}CO_2$ is what
permits the increase of $\Delta^{14}C_{atm}$. Since the ocean carbon inventory changes by only +0.2 percent, the mean $\Delta^{14}C$ value
for the global ocean is nearly unaffected, a decrease of only ~11 permil in the new steady state (Fig. 6g).
In the model run where the sediment model is active, there are two distinct time constants. A rapid increase
of $\Delta^{14}C_{atm}$ occurs, ~143 permil in the first few hundred years, then $\Delta^{14}C_{atm}$ gradually decreases to its final value of ~91
permil after tens of thousands of years. Reduced deep ocean ventilation is again responsible for the rapid $\Delta^{14}C_{atm}$
change and the respective time constant (T = ~480 years). The second time constant of ~23,390 years is due to the
relatively long time required for the ocean carbon inventory to adjust to the ocean circulation-driven imbalance
between weathering and sedimentation.
The process of ocean circulation interacts with the efficiency of the ocean's biological carbon pump, via its
impact on export production, ocean interior oxygen levels, and seawater carbonate chemistry/equilibria. This has
important implications for the sedimentation of biogenic material on the sea floor and, on a time scale of tens of
thousands of years, the total oceanic amount of carbon. Through this coupling of ocean circulation and sea floor
sedimentation via the biological carbon pump, a halving of $\tau$ and $K_V$ leads to a 9.8 percent increase of the ocean carbon
inventory in the new steady state (Fig. 5e). Qualitatively, a reduction in the ocean's overturning circulation leads to a
lower surface nutrient supply, which limits the production and export of biogenic material from the surface ocean.
This, in turn, decreases the fluxes of POC and $CaCO_3$ to the sea floor, with major consequences for the magnitude of
their removal by sedimentation. At the same time, a constant input of DIC, Alk, and nutrients is added to the ocean
from terrestrial weathering which is no longer balanced by sedimentation on the sea floor (this is what permits a larger
ocean carbon inventory). The overall effect is a gradual reduction of oceanic $\Delta^{14}C$ by ~76 permil (Fig. 6g), which
dilutes the initial $\Delta^{14}C_{atm}$ peak by 52 permil.

**3.1.3 Change of gas transfer velocity**

It takes about a decade for the isotopic ratios of carbon to equilibrate between the atmosphere and a ~75-m thick
surface mixed layer by air-sea gas exchange (Broecker and Peng, 1974). A consequence of this is that the surface
ocean is undersaturated with respect to $\Delta^{14}C_{atm}$ (see Fig. 3). The choice of gas transfer velocity $k_w$ as a function of
wind speed is critical for the efficiency of air-sea gas exchange. A reduction of $k_w$ corresponds to a higher resistance
for gas transfer across the air-sea interface, which means that the $^{14}C$ produced in the atmosphere escapes into the
surface ocean at a slower rate. The effect of a lower $k_w$ is a larger air-sea gradient of $\Delta^{14}C$ and higher $\Delta^{14}C_{atm}$ values.
In contrast, the $\Delta^{14}C$ value for the surface ocean is nearly unaffected so long as the ocean carbon inventory remains
approximately constant, since the vertical gradient of $\Delta^{14}C$ in the ocean is dominated by physical transport and mixing
processes. Although the exact nature of the gas transfer velocity under glacial climate conditions remains unclear, $k_w$
represents a straightforward way to reduce the model's air-sea exchange efficiency due to theoretical changes in wind
stress, sea ice, etc.

Fig. 4 shows how $\Delta^{14}C_{atm}$ responds to a perturbation in the gas transfer velocity. In the model run without

sediments, reduction of $k_w$ to 0 percent of its preindustrial value, in the model's north (> 60°N) and south (> 48°S)
polar areas, leads to a moderate increase of $\Delta^{14}C_{atm}$ in the new steady state. The amplitude of $\Delta^{14}C_{atm}$ change is ~42
permil, which is achieved with an e-folding time scale T of about 180 years. This relatively short time constant can be
explained by the multidecadal time scale required for $\Delta^{14}C$ to equilibrate between the model's atmosphere, upper
ocean, and terrestrial biosphere. As shown in Fig. 6, the mean $\Delta^{14}C$ values for the surface, deep, and global ocean in
the new steady state are only slightly different from the preindustrial steady-state values, as expected from the fact
that the ocean carbon inventory remains relatively stable.

Interestingly, if sediments are included in the model, the final value of $\Delta^{14}C_{atm}$ is much higher (~91 permil).

In this case, a perturbation in $k_w$ leads to a very rapid initial increase of $\Delta^{14}C_{atm}$ (~42 permil), and a much slower
subsequent increase of $\Delta^{14}C_{atm}$ (~49 permil). The latter has an e-folding time scale T of about 14,200 years. This slow
doubling of the initial $\Delta^{14}C_{atm}$ increase is unexpected, but can be explained by the fact that a reduction of $k_w$ involves
also a reduction of air-sea $O_2$ gas exchange in the deep water formation regions, decreasing the oceanic oxygen that
is available for transport to the deep ocean. This, in turn, implies lower oxygen concentrations in the water column
and the sediment pore waters, decreasing the rate of POC remineralization in the sediments. Reducing this has the
overall effect of enhancing POC sedimentation on the sea floor, causing the ocean carbon inventory to decrease. As
shown in Fig. 5f, the total oceanic amount of carbon decreases by 5.9 percent in the new steady state, resulting in
elevated $\Delta^{14}C$ values for the surface (+56 permil), deep (+30 permil), and global (+37 permil) ocean as well as for the
atmosphere (+91 permil) (see Fig. 6). Note that the increase in $\Delta^{14}C_{atm}$ is not accompanied by a significant change in
the atmospheric carbon inventory, which decreases by only 2.2 to 3.3 percent. The air-sea equilibration time scale for
$CO_2$ by gas exchange is about 1 year for a ~75-m thick surface mixed layer (Broecker and Peng, 1974), which is much
smaller than the ventilation time scale for the deep ocean (on the order of several hundred years or more). One would
therefore expect that the oceanic uptake of $CO_2$ demonstrates only a very small response to changes in $k_w$.

Overall, findings from these sensitivity experiments demonstrate that (1) the response of $\Delta^{14}C_{atm}$ to changes

in the internal parameters of the ocean carbon cycle, in contrast to $^{14}C$ production changes, depends strongly on
whether or not the balance between terrestrial weathering and sedimentation on the sea floor is simulated, (2) the e-
folding time scale for the initial adjustment of $\Delta^{14}C_{atm}$ to ocean carbon cycle changes, i.e., changes in ocean circulation
and gas exchange, is shorter than that for production changes (i.e., ~600 years and ~180 years versus ~6170 years),
(3) air-sea gas exchange, in contrast to ocean circulation, has only a small effect on atmospheric $CO_2$, given that gas
exchange is not the rate-limiting step for oceanic $CO_2$ uptake, and (4) on time scales of tens of thousands of years
changes in the balance between weathering and sedimentation can potentially diminish (or elevate) the $\Delta^{14}C_{atm}$ value.
This is new, important information for future paleoclimate simulations and suggests that changes in $\Delta^{14}C_{atm}$ may be
overestimated (or underestimated) in models that do not simulate the interaction between sea floor sediments and the
overlying water column.

**3.2 Role of $^{14}C$ production in past atmospheric $\Delta^{14}C$ variability**

We now consider the component of past $\Delta^{14}C_{atm}$ variability caused by production changes alone. Fig. 7 shows the
results of model runs using different reconstructions of the $^{14}C$ production rate, as inferred from paleointensity data
and from ice-core $^{10}Be$ fluxes. The global carbon cycle is assumed to be constant and under preindustrial conditions
for these simulations (i.e., scenario MOD is used). Our analysis is restricted to the glacial portion of the record (50 to
18 kyr BP), in part because this is the time period which experiences the largest production changes, and in part
because we did not attempt to reproduce the ~80 ppm change in atmospheric $CO_2$ that occurred during the last glacial
termination. As we have already noted, much research over the last decades has attempted to explain the observed
glacial-interglacial variations in $\Delta^{14}C_{atm}$ and $CO_2$, and this was not the goal of this study.

At first glance, the millennial-scale structure of model-simulated $\Delta^{14}C_{atm}$ is comparable to that of the

reconstructions. These similarities appear to be highest for the oldest portion of the record, roughly before 30 kyr BP.
The model reproduces major features of the reconstructed $\Delta^{14}C_{atm}$ variability such as the large changes associated with
the Laschamp (~41 kyr BP) and Mono Lake (~34 kyr BP) geomagnetic excursions. These two events are clearly
expressed as distinct maxima in all model-simulated records. A more detailed comparison reveals a high correlation
between the modelled and reconstructed $\Delta^{14}C_{atm}$ values between 50 and 33 kyr BP. Of note is the better agreement
with the new Hulu Cave $\Delta^{14}C_{atm}$ dataset as compared to the IntCal13 calibration curve (i.e., Pearson correlation
coefficient $r$ of 0.96 versus 0.91). This is likely due to the fact that the Laschamp excursion is smoothed/smeared out
during the stacking process of the IntCal13 $\Delta^{14}C_{atm}$ datasets (Adolphi et al., 2018). The correlation between modelled
and reconstructed $\Delta^{14}C_{atm}$ is much weaker during the millennia after the Mono Lake excursion (33 to 18 kyr BP; $r =$
0.52 to 0.64). While it is clear that much of the millennial-scale variation in $\Delta^{14}C_{atm}$ is driven by past changes in $^{14}C$
production, the model fails to reproduce the glacial level of $\Delta^{14}C_{atm}$ and also does not capture the ~15,000-year
persistent elevation of $\Delta^{14}C_{atm}$ or the subsequent decrease of $\Delta^{14}C_{atm}$ after ~25 kyr BP.

The reconstructions suggest that the highest values of $\Delta^{14}C_{atm}$ occurred during the Laschamp excursion, with
a maximum value of ~595 permil at 41.1 kyr BP found in the IntCal13 record. The Hulu Cave record indicates even
higher values for the Laschamp event $\Delta^{14}C_{atm} =$ ~742 permil, at 39.7 kyr BP). In contrast, the model is able to simulate
maximum $\Delta^{14}C_{atm}$ values of only ~364 permil at 40.4 kyr BP, and ~236 permil at 40.5 kyr BP, as predicted by the
paleointensity-based and ice-core $^{10}Be$-based production rate estimates, respectively. Although the model is unable to
reproduce the reconstructed values of $\Delta^{14}C_{atm}$, the modelled amplitude of the variation in $\Delta^{14}C_{atm}$ in response to the
Laschamp event shows a reasonable agreement with the reconstructed amplitude of $\Delta^{14}C_{atm}$ change found in the
IntCal13 record (~240 permil). The $\Delta^{14}C_{atm}$ change predicted by paleointensity data has a maximal amplitude of about
320 permil, whereas the ice-core $^{10}Be$ data indicate a smaller amplitude (~224 permil). Note that the IntCal13 and
model-simulated amplitudes of the Laschamp-related $\Delta^{14}C_{atm}$ change are about two times smaller than that observed
in the Hulu Cave record (~575 permil), which is more likely to be correct.

Moving onto the full glacial record (50 to 18 kyr BP), there are considerable discrepancies between
reconstructed and modelled $\Delta^{14}C_{atm}$ ($\Delta\Delta^{14}C$; see Fig. 7). The use of ice-core $^{10}Be$ data to predict past changes in $\Delta^{14}C_{atm}$
results in the largest $\Delta\Delta^{14}C$, with offsets between the records as high as ~544 to 558 permil (root-mean-square error
$RMSE$ = 404 to 408 permil). Model-simulated $\Delta^{14}C_{atm}$ given by paleointensity data varies widely between the four
available reconstructions, yielding $\Delta\Delta^{14}C$ values of ~325 to 639 permil ($RMSE$ = 206 to 455 permil). Note that the
upper limit of the paleointensity-based $\Delta\Delta^{14}C$ overlaps with the ice-core $^{10}Be$-based $\Delta\Delta^{14}C$. Given the uncertainties
associated with the reconstruction of past changes in $^{14}C$ production, accurate predictions of its contribution to past
changes in $\Delta^{14}C_{atm}$ are challenging. Nonetheless, the substantial systematic offsets between the reconstructed and
model-simulated $\Delta^{14}C_{atm}$ records after ~33 kyr BP point toward insufficiently high $^{14}C$ production rates over this period
of time. The question arises as to whether another factor besides geomagnetic modulation of the cosmic ray intensity
was responsible for elevated glacial $\Delta^{14}C_{atm}$ levels. The effect of ocean carbon cycle changes on the evolution of
$\Delta^{14}C_{atm}$ is considered next.

**3.3 Carbon cycle contribution to high glacial atmospheric $\Delta^{14}C$ levels**

Here we investigate the magnitude and timing of the maximum possible $\Delta^{14}C_{atm}$ change during the last glacial period,
obtained by running the Bern3D model with eight different carbon cycle scenarios (see Table 1). For the sake of
clarity, we will discuss only the results of model runs using the mean paleointensity-based [14]C production rate, though
all available reconstructions were used. We emphasize that this is not a best-guess estimate of paleointensity-based
[14]C production. One should focus on the relative changes of $\Delta^{14}C_{atm}$ between model scenarios, and how specific carbon
cycle processes affect the glacial level of $\Delta^{14}C_{atm}$.

Modelled 50,000-year records of $\Delta^{14}C_{atm}$ and $CO_2$ as well as their reconstructed histories are shown in Fig.
8. In order to provide a basis for comparison of modelling efforts, the results of model run MOD (which assumes a
constant preindustrial carbon cycle) are presented. The influence of ocean carbon cycle changes on $\Delta^{14}C_{atm}$ was tested
in the other model runs. Interestingly, the forcing fields for model run PAL (orbital parameters, greenhouse gas
radiative forcing, and ice sheet extent) have only a minimal impact on $\Delta^{14}C_{atm}$. The PAL forcing fields also do not
achieve sufficiently low glacial $CO_2$ concentrations. Only a slight reduction of atmospheric $CO_2$ by ~20 ppm could be
achieved, which unrealistically occurs during the last glacial termination ($CO_2$ = 258.07 ppm, at 14.6 kyr BP). With
hypothetical carbon cycle changes, the agreement between observed and modelled $CO_2$ during the last glacial period
is good (as by design), but the deglacial $CO_2$ rise is lagged and ~60 ppm too small at 11 kyr BP. Since this study
focuses on glacial $\Delta^{14}C_{atm}$ levels before incipient deglaciation at ~18 kyr BP, we will not discuss the lag any further.

Model simulation of high glacial $\Delta^{14}C_{atm}$ levels can be significantly improved by considering hypothetical
carbon cycle changes in conjunction with PAL forcing. The amplitude of $\Delta^{14}C_{atm}$ change is highest for runs CIRC,
VENT, and VENTx. This behavior is due to the fact that, owing to a reduction of $\tau$, $K_V$, and $k_w$, strong vertical $\Delta^{14}C$
gradients in the ocean, as well as a large air-sea $\Delta^{14}C$ gradient, are established. As shown in Fig. 8, a more sluggish
ventilation of deep waters is clearly expressed as an increase in the model ocean's global average ideal age and surface-
and deep-water reservoir ages, where the latter two are calculated for the surface ocean and bottom water grid cells,
respectively. These are equivalent to radiocarbon reservoir age offsets following Soulet et al. (2016). The deep-water
reservoir age (i.e., B-Atm [14]C age offset, or B-Atm) provides a measure of the radiocarbon disequilibrium between
the deep ocean and the atmosphere, which arises due to the combined effect of air-sea gas exchange efficiency and
the deep ocean ventilation rate, whereas the effect of upper ocean stratification and/or sea ice on air-sea gas exchange
is particularly important for surface reservoir ages (i.e., surface R-age) (Skinner et al., 2019).

Driven by a reduction in ocean circulation, model run CIRC predicts a substantial increase in B-Atm during
the last glacial, which is defined here as 40 to 18 kyr BP to avoid biasing global mean estimates toward Laschamp
values. The global average glacial B-Atm predicted by CIRC is ~3225 [14]C years, representing an increase in B-Atm
of ~1599 [14]C years relative to the preindustrial value of ~1626 [14]C years. Model run VENT predicts a slightly larger
increase in glacial B-Atm due to the inhibition of air-sea gas exchange. The "oldest" glacial waters are found in model
run VENTx where air-sea gas exchange is severely restricted, yielding an increase in B-Atm of ~1912 [14]C years
(glacial B-Atm ~3538 [14]C years). The glacial B-Atm values given by runs CIRC, VENT, and VENTx, as well as the
~717 year increase in ideal age during the last glacial relative to preindustrial, suggest that the glacial deep ocean was
about two times older than its preindustrial counterpart. Comparison of our LGM B-Atm estimates (range of 3682 to
3962 $^{14}$C years) with the compiled LGM marine radiocarbon data of Skinner et al. (2017) demonstrate that the carbon
cycle scenarios are extreme, although it should be noted that Skinner et al. consider a wider depth range (~500 to 5000
m) of the ocean than we do. Skinner et al. (2017) predict a global average LGM B-Atm value of ~2048 $^{14}$C years, an
increase of ~689 $^{14}$C years relative to preindustrial. Turning our comparison to surface reservoir ages, we note that
our global average LGM surface R-age of ~1132 $^{14}$C years from runs VENT and VENTx is comparable to the ~1241
$^{14}$C years obtained by Skinner et al. (2017) for the LGM. The model-based estimates of surface R-age from Butzin et
al. (2017) indicate a much lower LGM value of ~780 $^{14}$C years, and values ranging from 540 to 1250 $^{14}$C years between
50 and 25 kyr BP. Note that these estimates are based on model-simulated values between 50°N and 50°S. If the polar
regions are included in the calculation (see Fig. 8c), their surface R-age estimates become comparable to our glacial
values (range of 911 to 1354 $^{14}$C years), and between about 34 and 22 kyr BP can exceed them, including even those
from model runs VENT and VENTx, unless $\Delta^{14}C_{atm}$ and $CO_2$ are prescribed (dashed colored lines in Fig. 8c) as in the
simulation by Butzin et al. (2017).

Indirect evidence for deep water ageing can be provided by the occurrence of depleted ocean interior oxygen
levels, due to the progressive consumption of dissolved oxygen during organic matter remineralization in the water
column. This situation is amplified by the slow escape of accumulating remineralized carbon in the ocean interior
(see, e.g., Skinner et al., 2017), leading to higher values of apparent oxygen utilization (AOU = $O_{2,pre} - O_2$). These
two concepts (increased AOU and increased B-Atm) taken together signal a significant reduction in deep ocean
ventilation characterized by a decrease in the exchange rate between younger (higher $\Delta^{14}C$) surface waters and older
($^{14}$C-depleted), carbon-rich deep waters. Model runs CIRC, VENT, and VENTx do indeed indicate a large increase in
AOU of about 95 mmol m$^{-3}$ from its preindustrial value of ~150 mmol m$^{-3}$. The reason for this AOU increase is that
a reduction of deep ocean ventilation permits enhanced accumulation of remineralized carbon in the ocean interior
and therefore a more efficient biological carbon pump. Model runs BIO, PHYS-BIO, and PHYS-BIOx allow us to
investigate the impact of other biological carbon pump changes on $\Delta^{14}C_{atm}$ and $CO_2$ (i.e., changes in the CaCO$_3$-to-
POC export ratio and POC remineralization length scale). While these changes lead to an effective atmospheric $CO_2$
drawdown mechanism, model results confirm that their effect on $\Delta^{14}C_{atm}$ is much less important (see Fig. 8).

Model run VENTx gives the best results with respect to glacial levels of $\Delta^{14}C_{atm}$, with a maximum
underestimation of ~202 to 229 permil ($RMSE$ = 103 to 110 permil) and a relatively good correlation ($r$ = 0.79 to
0.91). Only one model parameter was changed for run VENTx as compared to runs CIRC and VENT, namely, the
polar gas transfer velocity $k_w$ was reduced to 0 percent of its preindustrial value during the last glacial. In this extreme
scenario, we assume that sea ice cover extended in the northern hemisphere as far south as 60°N and in the southern
hemisphere as far north as 48°S, which is not supported by the reconstructions (Gersonde et al., 2005; Allen et al.,
2011). Nonetheless, considering extreme assumptions about polar air-sea exchange efficiency under glacial climate
conditions is interesting for two reasons: (1) a change in gas exchange hardly affects the atmospheric $CO_2$
concentration, and (2) an additional change of $\Delta^{14}C_{atm}$ could possibly be achieved on a time scale of tens of thousands
of years by changing the balance between weathering and sedimentation (see Sect. 3.1.3). This behavior has important
implications for the glacial atmosphere, which is characterized by high $\Delta^{14}C$ levels in conjunction with low but
relatively stable $CO_2$ concentrations. In contrast to a change in ocean circulation, air-sea gas exchange is a dedicated
$\Delta^{14}C_{atm}$ "control knob" that can be invoked by models for a further increase of $\Delta^{14}C_{atm}$ without changing atmospheric
$CO_2$. Here, an additional increase in $\Delta^{14}C_{atm}$ of ~130 permil relative to CIRC and VENT is achieved if gas exchange
is reduced permanently to 0 percent in the polar regions.

While the modelled $\Delta^{14}C_{atm}$ values obtained by VENTx show rather good agreement with the reconstructions
between 50 and 33 kyr BP ($r$ = 0.92 to 0.96; *RMSE* = 74 to 102 permil), considerable discrepancies remain for the
younger portion of the record. The analysis shown in Fig. 9 illustrates that even with extreme changes in the ocean
carbon cycle it is very difficult to reproduce the reconstructed $\Delta^{14}C_{atm}$ values after ~33 kyr BP. During this period of
time, VENTx underestimates $\Delta^{14}C_{atm}$ by up to ~203 permil (*RMSE* = 118 to 128 permil), and is very poorly ($r$ = 0.1)
correlated with the reconstructions, confirming that there are still considerable gaps in our understanding. Although it
may be possible that permanent North Atlantic-Arctic and Antarctic sea ice cover extended to lower and higher
latitudes than previously reconstructed, we conclude from our model study that even extreme assumptions about sea
ice cover are insufficient to explain the elevated $\Delta^{14}C_{atm}$ levels after ~33 kyr BP. It appears instead that the glacial $^{14}C$
production rate was higher than previously estimated and/or the reconstruction of glacial $\Delta^{14}C_{atm}$ levels is biased high.
The older portion of the $\Delta^{14}C_{atm}$ record is based on data from archives other than tree rings (i.e., plant macrofossils,
speleothems, corals, and foraminifera) (Reimer et al., 2013), providing, except for the Lake Suigetsu plant macrofossil
data (Bronk Ramsey et al., 2012), only indirect measurements of $\Delta^{14}C_{atm}$. Note that these data show uncertainty in
calendar age that propagate into the estimation of past $\Delta^{14}C_{atm}$ levels.

Large uncertainties in the pre-Holocene $^{14}C$ production rate also hamper our qualitative and quantitative
interpretation of the $\Delta^{14}C_{atm}$ record. There is considerable disagreement between the available reconstructions of past
changes in $^{14}C$ production (Fig. 1). Paleointensity-based estimates typically predict higher $^{14}C$ production rates than
ice-core $^{10}Be$-based ones. An exception is the paleointensity stack from Channell et al. (2018), which predicts lower
production rates. But, irrespective of the scatter, it is clear that all of the $^{14}C$ production rate estimates are insufficiently
high to explain the elevated $\Delta^{14}C_{atm}$ levels during the last glacial. Given the uncertainties in these estimates, it is very
difficult to quantitatively describe the role of the ocean carbon cycle in determining the $\Delta^{14}C$ and $CO_2$ levels in the
glacial atmosphere.

**3.4 Reconstructing the $^{14}C$ production rate by deconvolving the atmospheric $\Delta^{14}C$ record**

The unresolved discrepancy between reconstructed and model-simulated $\Delta^{14}C_{atm}$ raises the question how the $^{14}C$
production rate would have had to evolve to be consistent with the IntCal13 calibration curve or the new Hulu Cave
$\Delta^{14}C_{atm}$ dataset. This question is addressed by deconvolving the $\Delta^{14}C_{atm}$ reconstruction over the last 50 kyr, using the
Bern3D carbon cycle model forced with reconstructed histories of $\Delta^{14}C_{atm}$ and $CO_2$ (see Eq. [2]). The carbon cycle
scenarios described in Table 1, with the exception of MOD, are used in order to provide an estimate of the uncertainty
associated with the model's glacial ocean carbon cycle. We note that the carbon cycle scenarios are not designed to
capture the specific features of the last glacial termination, and therefore the results of the deconvolution over this
time period must be considered very preliminary (and regarded as tentative). A detailed analysis of the Holocene $^{14}$C
production rate is available in the literature (Roth and Joos, 2013). Finally, we consider the uncertainties associated
with the older portion of the $\Delta^{14}C_{atm}$ record by deconvolving both the IntCal13 and Hulu Cave $\Delta^{14}C_{atm}$ records. Hulu
Cave data overlap with IntCal13 between ~10.6 and 33.3 kyr BP (Cheng et al., 2018), as expected from the fact that
IntCal13 between 10.6 and 26.8 kyr BP is based in part on Hulu Cave stalagmite H82 (Southon et al., 2012), whereas
there are substantial offsets before ~30 kyr BP.

Fig. 10 shows the new, model-based reconstruction of past changes in $^{14}$C production compared with
available measurement-based reconstructions. Before the onset of the Laschamp excursion at ~42 kyr BP, production
rates as inferred from the Hulu Cave record are near modern levels, whereas those obtained from the IntCal13 record
are somewhat higher than modern. As expected, peak production occurs during the Laschamp event (~42 to 40 kyr
BP), with the Hulu Cave dataset yielding the largest amplitude (factor of ~2 greater than modern). The IntCal13 record
predicts a smaller amplitude of ~1.6 times the modern value. Both $\Delta^{14}C_{atm}$ records predict production minima at ~37
kyr BP (~7 percent higher than modern) and ~32 kyr BP (~5 percent higher than modern), interrupted by a prominent
peak (factors of ~1.5 and ~1.4, respectively) during the Mono Lake geomagnetic excursion (~34 kyr BP), though the
details of the timing and structure differ between the two records. Between 32 and 22 kyr BP, model-based estimates
of the $^{14}$C production rate are ~1.3 times the modern value, which then decrease to around modern levels by HS1 (~18
kyr BP).

Model-based estimates of $^{14}$C production during the last glacial are typically higher than paleointensity-based
and ice-core $^{10}$Be-based ones, as expected from Sect. 3.2. Between 32 and 22 kyr BP, the deconvolutions of the
IntCal13 and Hulu Cave $\Delta^{14}C_{atm}$ records give estimates that are about 17.5 percent higher than the reconstructions. It
is important to note that the differences between the reconstructions based on proxy data (i.e., paleointensity data and
ice-core $^{10}$Be fluxes) are as large as the differences between our deconvolution results and the reconstructions (see
Table 2). As shown in Fig. 11, it is extremely difficult to reconcile the discrepancies between measurement- and
model-based $^{14}$C production on the basis of carbon cycle changes alone. Nonetheless, the fact remains that two
independent estimates of the $^{14}$C production rate (i.e., estimates inferred from paleointensity data and from ice-core
$^{10}$Be fluxes) show systematically lower rates than those obtained by our model-based deconvolution of $\Delta^{14}C_{atm}$, in
particular between 32 and 22 kyr BP. The differences between the production rate results shown in Fig. 10 and Fig.
11 and Table 2 stem from various uncertainties that are discussed next.

Uncertainties associated with the glacial ocean carbon cycle (Fig. 10, colored shading; Fig. 11, colored lines)
are systematic in our approach. The deconvolutions, e.g., of the Hulu Cave $\Delta^{14}C_{atm}$ record, under different model
scenarios are offset against one another, whereas the millennial-scale variability is maintained (see Fig. 11). We do
not attempt to resolve uncertainties associated with Dansgaard-Oeschger warming events and related Antarctic and
tropical climatic excursions in the model runs. Such climatic events may have influenced the atmospheric radiocarbon
budget, but their influence on long-term variations in $\Delta^{14}C_{atm}$, and therefore inferred production rates, is presumably
limited. As may be expected, the lowest production rates (the lowest $F_{as}$ values) are found in VENTx and the highest
in scenarios PAL and BIO, mirroring the high and low glacial $\Delta^{14}C_{atm}$ levels achieved by these model scenarios as
discussed in Sect. 3.3. Note that there is a large uncertainty in the model-based $^{14}$C production rate stemming from
uncertainties associated with the reconstruction of past changes in $\Delta^{14}C_{atm}$, in particular the older portion of the $\Delta^{14}C_{atm}$
record.

A shortcoming of paleointensity-based reconstructions of the $^{14}$C production rate is that they neglect changes

in the solar modulation of the cosmic radiation. The solar modulation potential, which describes the impact of the
solar magnetic field on isotope production, varied between 100 and 1200 MeV during the Holocene on decadal to
centennial time scales, with a median value of approximately 565 MeV (Roth and Joos, 2013). A halving of the solar
modulation potential (e.g., from 600 to 300 MeV) increases the $^{14}$C production rate by about 25 percent for the modern
geomagnetic field strength (Roth and Joos, 2013; see their Fig. 13). This sensitivity remains similar when changes in
the strength of the geomagnetic field are limited as during the last ~35 kyr (Muscheler and Heikkilä, 2011). A shift to
lower solar modulation potential could have materialized if the sun spent on average more time in the postulated
"Grand Minimum" mode (Usoskin et al., 2014) during the last glacial than during the Holocene. The sensitivity of
isotope production to variations in solar modulation potential becomes large during the Laschamp event when the
intensity of the geomagnetic field was close to zero and changes in the solar modulation of the cosmic ray flux may
have a discernible impact on the high $\Delta^{14}C_{atm}$ levels found over this period. A reduction of the solar modulation
potential from 600 to 0 MeV would double $^{14}$C production during times of zero geomagnetic field strength (Masarik
and Beer, 2009). However, it is likely that changes in the solar modulation potential were insufficient to explain the
discrepancy between paleointensity-based production rate estimates and the results of our deconvolution, in particular
for the post-Laschamp period and for the reconstruction by Channell et al. (2018). Uncertainties associated with the
paleointensity-based reconstructions stem also from uncertainties in estimating the age-scales of the marine sediments
and the geomagnetic field data.

The ice-core $^{10}$Be-based reconstruction of past changes in $^{14}$C production reflects, by definition, the combined

influence of changes in the solar and geomagnetic modulation of the cosmic ray flux reaching the Earth. This method,
therefore, avoids a fundamental shortcoming of reconstructions based on geomagnetic field data. The assumption is
that the $^{10}$Be and $^{36}$Cl deposited on polar ice and measured in ice cores scales with the amount of cosmogenic isotopes
in the atmosphere. A difficulty is to extrapolate measurements from a single or a few locations to the global
atmosphere. Changes in climate influence atmospheric transport and deposition of $^{10}$Be as well as the snow
accumulation rate, which affect the ice-core $^{10}$Be concentration (Elsässer et al., 2015). Furthermore, the sensitivity of
$^{10}$Be in polar ice versus the sensitivity of total production to magnetic field variations, or "polar bias", is a point of
debate, but atmospheric transport models (Heikkilae et al., 2009; Field et al., 2006) and data analyses (Bard et al.,
1997; Adolphi and Muscheler, 2016; Adolphi et al., 2018) reach different conclusions about its existence and
magnitude. If a polar bias was present, it would lead to an underestimation of the geomagnetic modulation of the ice-
core [10]Be flux, and therefore variations in the [10]Be-based [14]C production rate would also be underestimated. However,
the mismatch of up to ~544 to 558 permil between reconstructed and modelled [10]Be-based $\Delta^{14}C_{atm}$ during the last
glacial (see Fig. 7c) appears to be much too large to be reconciled by considering uncertainties in the polar bias alone.
Furthermore, this mismatch with reconstructed $\Delta^{14}C_{atm}$ is qualitatively similar when using paleointensity-based [14]C
production rates that do not suffer from a polar bias (Fig. 7c).

Given the uncertainties associated with the proxy records, it may not be surprising that estimates of the [14]C
production rate for the last 50 kyr, as obtained by three fundamentally different methods (geomagnetic field data from
marine sediments, [10]Be and [36]Cl measurements in polar ice cores, and model-based deconvolution of $\Delta^{14}C_{atm}$), disagree
with one another, typically by order 10 percent and sometimes by up to 100 percent. At the same time, it is intriguing
that two independent estimates of the [14]C production rate (i.e., estimates inferred from paleointensity and ice-core [10]Be
data) give values that are systematically lower than what is required to match the $\Delta^{14}C_{atm}$ reconstruction.

**4 Summary and conclusions**

It is generally assumed that $\Delta^{14}C_{atm}$ is controlled by abiotic processes such as atmospheric [14]C production, air-sea gas
exchange, and ocean circulation and mixing. Here, results from sensitivity experiments with the Bern3D earth system
model of intermediate complexity suggest that $\Delta^{14}C_{atm}$ is potentially quite sensitive to the interaction with the ocean
sediments on multimillennial time scales. This rather surprising result is due to the coupling of ocean circulation and
the sedimentation of biogenic material on the sea floor via the biological carbon pump, which has important
implications for the ocean carbon inventory. If the model's ocean carbon cycle is sufficiently perturbed, e.g., by
changing the inputs or parameters controlling ocean circulation and/or gas exchange, the imbalance between
weathering and sedimentation has a significant impact on the total oceanic amount of carbon. On time scales of tens
of thousands of years this slow change in the ocean carbon inventory influences the partitioning of [14]C/C between the
ocean and atmosphere, and thus also oceanic $\Delta^{14}C$ and $\Delta^{14}C_{atm}$. This is important information for long-term climate
studies and paleoclimate modelling efforts concerning $\Delta^{14}C_{atm}$. Note that the representation of terrestrial weathering
and sea floor sedimentation in the Bern3D is necessarily simplified compared to reality. Nonetheless, a change in the
ocean carbon inventory linked with the weathering/sedimentation balance should be discussed as one of the potentially
important factors affecting $\Delta^{14}C_{atm}$ during the last glacial period.

The reason for the high $\Delta^{14}C$ values exhibited by the glacial atmosphere is still not clear. In order to
investigate potential mechanisms governing glacial $\Delta^{14}C_{atm}$ levels, the Bern3D model is again used as a tool. Results
of model simulations forced only by production changes point out that none of the available reconstructions of the [14]C
production rate can explain the full amplitude of $\Delta^{14}C_{atm}$ change during the last glacial. In order to test the sensitivity
of the model results with respect to the ocean carbon cycle state, various model parameters, i.e., different sets of
physical and biogeochemical parameters, were "tuned" to match the glacial $CO_2$ level. From this, we find that $\Delta^{14}C_{atm}$
is most sensitive to changes in physical model parameters, in particular those controlling ocean circulation and gas
exchange. In order to achieve an $\Delta^{14}C_{atm}$ value close to the glacial level, the gas transfer velocity in the polar regions
had to be reduced by 100 percent. If interpreted as being due to a greater extent of permanent sea ice cover, a reduction
in polar air-sea exchange efficiency is a possible explanation for high glacial $\Delta^{14}C_{atm}$ levels. Although this hypothesis
is compelling, such a scenario is not supported by the proxy records of Antarctic sea ice cover (Gersonde et al., 2005;
Allen et al., 2011) and the $^{13}C/^{12}C$ ratio of atmospheric $CO_2$ (Eggleston et al., 2016).
Atmospheric $\Delta^{14}C$ that is modelled at any point in time reflects $^{14}C$ production at that point, as well as the
legacy of past production and carbon cycle changes. The question arises as to whether our conclusions are affected by
unaccounted legacy effects, e.g., linked to the preindustrial spin-up simulation or model-diagnosed production rates.
Transient simulations forced by reconstructed changes in $^{14}C$ production (Sect. 3.2 and 3.3) are initialized at 70 kyr
BP, but their interpretation is restricted to the last 50,000 years of the integration to minimize legacy effects from
model spin-up. Available reconstructions of the $^{14}C$ production rate in relative units (Sect. 2.5) are applied as a scale
factor to the preindustrial steady-state absolute value, which is diagnosed by running the Bern3D model to equilibrium
under preindustrial boundary conditions. This approach represents an approximation and equilibrium conditions do
not fully apply. Indeed, there is a mismatch between reconstructed and modelled $\Delta^{14}C_{atm}$ at the preindustrial (see Fig.
8a). This mismatch is on the order of a few percent or less and adjusting the base level of production accordingly
would not remove the large mismatch between reconstructed and modelled $\Delta^{14}C_{atm}$ during the last glacial. In addition,
the uncertainty in the absolute value of the preindustrial production rate is on the order of 15%, primarily due to the
uncertainties in the preindustrial ocean radiocarbon inventory (see Roth and Joos, 2013, Sect. 3.2). This potential
systematic bias, however, does not affect our conclusions as we consider normalized production rate changes (see Fig.
7, 10, and 11).
Before model-simulated $\Delta^{14}C_{atm}$ can be taken seriously, it must be demonstrated that the reconstruction of
past changes in $^{14}C$ production is reliable. There is, however, a substantial amount of scatter in the paleointensity-
based and ice-core $^{10}Be$-based estimates of $^{14}C$ production. Here we adopt an alternative approach to estimating the
$^{14}C$ production rate, which would indeed benefit from further constraints and lines of supporting evidence. Our
deconvolution-based approach assumes that the $^{14}C$ production rate can be derived from an atmospheric radiocarbon
budget, constructed using a prognostic carbon cycle model combined with the $\Delta^{14}C_{atm}$ record. Here, non-equilibrium
effects are fully accounted for by transient simulations where $\Delta^{14}C_{atm}$ and $CO_2$ are prescribed following their
reconstructed histories (Sect. 3.4). Yet, these simulations indicate that the discrepancy between measurement- and
model-based estimates of the $^{14}C$ production rate remains for the last glacial (Fig. 10b). This would suggest that
unaccounted legacy effects do not significantly affect our conclusions. Our model results imply that the glacial $^{14}C$
production rate as inferred from paleointensity data and ice-core $^{10}Be$ fluxes may be underestimated by about 15
percent between 32 and 22 kyr BP, a time interval which appears to be an important piece of the glacial-interglacial
$\Delta^{14}C_{atm}$ puzzle. Note that our model-based estimates are associated with uncertainties arising from the reconstruction
of the older portion of the $\Delta^{14}C_{atm}$ record and from the model simulation of the glacial ocean carbon cycle (e.g.,
uncertainties in the glacial ocean circulation and air-sea $CO_2$ fluxes). An improved understanding of the role of $^{14}$C
production in past changes of $\Delta^{14}C_{atm}$ would open up the possibility of attributing model deficiencies to real changes
in the ocean carbon cycle, but there is as yet no emerging single record of the $^{14}$C production rate.
Progress in several different areas may help to resolve the glacial-interglacial radiocarbon problem.
Additional records of glacial $\Delta^{14}C_{atm}$ would help refine the older portion of the IntCal $\Delta^{14}$C record. Cosmogenic isotope
production records may be improved, e.g., by refining estimates of ice accumulation, by developing a better
understanding of $^{10}$Be transport and deposition during the glacial, by recovering additional long and continuous
records from Antarctic ice cores and including marine $^{10}$Be records, and by obtaining additional geomagnetic field
data. An expanded spatiotemporal observational coverage of $\Delta^{14}$C of DIC in the surface and deep ocean would help
narrow the time scales of surface-to-deep transport and air-sea equilibration of $\Delta^{14}$C, carbon and nutrients, and thereby
guide model-based analyses. Models should become more sophisticated and detailed in order to reproduce
successfully the glacial-interglacial changes in carbon and radiocarbon, by including exchange with sediments and the
lithosphere and by better representing coastal processes, and by representing a wide variety of paleo proxies such as
$\delta^{13}$C, Nd isotopes, carbonate ion concentration, lysocline evolution, and paleo-productivity proxies in a 3-D dynamic
context for model evaluation. What is also missing are methods to quantify how the ocean carbon inventory, which
co-determines the $^{14}$C/C ratio and thus the $\Delta^{14}$C values in the ocean and atmosphere, has changed over the last 50,000
years. Ultimately, an improved knowledge of $^{14}$C production during the last glacial, as well as more robust constraints
on the prevailing climate conditions (e.g., ocean circulation, sea ice cover, and wind speed), are necessary to elucidate
the processes permitting mysteriously high $\Delta^{14}$C levels in the glacial atmosphere.
**Appendix A: Description of the Bern3D model**
The physical core of the Bern3D model is based on the 3-D rigid-lid ocean model of Edwards et al. (1998) as updated
by Edwards and Marsh (2005). The forcing fields for the model integration are monthly mean wind stress data taken
from NCEP/NCAR (Kalnay et al., 1996). Diapycnal mixing is parameterized with a uniform vertical diffusivity $K_V$ of
$2 \times 10^{-5}$ m s$^{-1}$. The parameterization of eddy-induced transport is separated from that of isopycnal mixing, using the
Gent-McWilliams skew flux (Griffies, 1998). Running at the same temporal and horizontal resolution, the one-layer
energy-moisture balance atmosphere model performs an analysis of the energy budget of the Earth by involving solar
radiation, infrared fluxes, evaporation and precipitation, and sensible and latent heat. The zonally averaged surface
albedo climatology is taken from Kukla and Robinson (1980). Transport of moisture is performed by diffusion and
advection and heat by eddy diffusion.
The Bern3D ocean carbon cycle model is based on the Ocean Carbon-Cycle Model Intercomparison Project
(OCMIP-2) protocols. Air-sea gas exchange is parameterized using the standard gas transfer formulation adopted for
OCMIP-2, except that the gas transfer velocity $k_w$ parameterization is a linear function of wind speed (Krakauer et
al., 2006) to which we have added a scale factor of 0.81 to match the observed global ocean inventory of bomb $^{14}$C
(Müller et al., 2008). It is assumed that $CO_2$ and $O_2$ are well-mixed in the atmosphere. Surface boundary conditions
also include a virtual-flux term for biogeochemical tracers (e.g., DIC and Alk) to account for their dilution or
concentration due to implicit freshwater fluxes. Following OCMIP-2 biotic protocol, new production is partitioned
into particulate and dissolved organic matter. Modifications from the original OCMIP-2 biotic protocol include the
prognostic formulation of new/export production as a function of light, temperature, and limiting nutrient
concentrations, where the nutrient uptake follows Michaelis-Menten kinetics. The production of biogenic $CaCO_3$ and
opal is computed on the basis of the modelled particulate organic carbon (POC) production and availability of silicate,
with a maximum possible fraction of $CaCO_3$ material that can be produced. This threshold value is represented by the
$CaCO_3$-to-POC export ratio. In the preindustrial control run, the global mean export ratio $rr$ is 0.082.

Biogenic particles that have been produced in the 75-m production zone are redistributed over the water

column in order to parameterize the downward particle flux through the water column. A power-law model referred
to as the Martin curve is used to describe the vertical POC flux profile, whereas both $CaCO_3$ and opal export are
redistributed over the water column with an exponential curve. POC is remineralized instantaneously back to dissolved
form according to Redfield stoichiometry and with a 250–m length scale $l_{POC}$ (i.e., in 250 m, the POC flux declines
by $1 - 1/e \approx 63$ percent). Likewise, $CaCO_3$ and opal are dissolved within one time step, with $e$-folding depths of
5066 and 10,000 m, respectively. Biogenic particles reaching the model's sea floor form the upper boundary condition
of the 10-layer sediment model after Heinze et al. (1999) and Gehlen et al. (2006). The sediment model includes four
solid sediment components (POC, $CaCO_3$, opal, and clay) and is based on the sediment advection and accumulation
scheme as in the work of Archer et al. (1993). The rate of POC remineralization in the sediments is primarily
determined by the pore water concentration of oxygen, whereas the mineral dissolution rate is governed by the
saturation state of sediment pore waters with respect to $CaCO_3$ or opal. Weathering (dissolution) of carbonate and
silicate rocks on land, phosphorous release by chemical weathering of rocks, and volcanic outgassing of $CO_2$ are
simulated as constant inputs of DIC, Alk, phosphate (P), and silicate (Si) to the ocean at rates intended to balance their
removal from the ocean by sedimentation on the sea floor. These weathering inputs are added as a constant increment
to each surface ocean grid cell along the coastlines. The preindustrial steady state of the model is used to diagnose the
weathering rates that are held fixed and constant throughout the simulations. Note that the preindustrial spin-up results
in steady-state values for weathering-derived inputs of DIC, Alk, P, and Si of 0.46 Gt C per year, 34.37 Tmol Alk per
year, 0.17 Tmol P per year, and 6.67 Tmol Si per year, respectively. These values are within the range of observational
estimates (see, e.g., Jeltsch-Thömmes et al., 2019). Additional details concerning the sediment model are provided in
Tschumi et al. (2011), while the appendix of Jeltsch-Thömmes et al. (2019) gives a detailed description of the
atmosphere–ocean–sediment spin-up.

The exchange of any isotopic perturbation between the atmosphere and the terrestrial biosphere is simulated

by use of the four-box model of Siegenthaler and Oeschger (1987). The terrestrial biosphere is represented by four
well-mixed compartments (ground vegetation plus leaves, wood, detritus, and soils), with a fixed total carbon
inventory of 2220 Gt C. Net primary production is balanced by respiration of detritus and soils, and is set to 60 Gt C
per year.

**Data availability.**  Model-simulated atmospheric $\Delta^{14}C$ presented in Fig. 7b and 8a, and model-based $^{14}C$ production
rates shown in Fig. 10a, are included in the Supplement. Other data generated or analyzed during this study can be
made available upon request to the corresponding author (A.D.).

**Author contribution.**  This study was designed by F.J. and A.D. with input from F.A. A.D. developed and
performed the model simulations. F.A. provided production data. A.D. wrote the manuscript with contributions from
the co-authors.

**Competing interests.**  The authors declare that they have no conflict of interest.

**Acknowledgements.**  This work was made possible by the Swiss National Science Foundation (#200020_172476)
and by the UniBE international 2021 fellowship program of the U. Bern. F.A. was supported by the Swedish
Research Council (Vetenskaprådet DNR: 2016-00218).

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

Table 1. Summary of model scenarios considered in this study. Initial conditions refer to the boundary conditions used
for the precursor spin-up simulation needed to initialize the transient simulation. These correspond either to
preindustrial (PI) or last glacial conditions. The paleoclimate forcing fields, i.e., Orb-GHG-Ice, are reconstructed
changes in orbital parameters (Berger, 1978), greenhouse gas radiative forcing based on reconstructed atmospheric
greenhouse gas histories (Köhler et al., 2017), and varying ice sheet extent scaled using the global benthic $\delta^{18}O$ stack
of Lisiecki and Stern (2016). Numbers refer to the scale factor values applied to the tunable model parameters $\tau$ (wind
stress scale factor), $K_V$ (vertical diffusivity), $k_w$ (gas transfer velocity), $rr$ (CaCO$_3$-to-POC export ratio), and $\ell_{POC}$
(POC remineralization length scale) at the last glacial maximum (LGM). These values were chosen in order to achieve
an atmospheric CO$_2$ concentration close to the LGM level, and are varied over time using the global benthic $\delta^{18}O$
stack. See Roth et al. (2014) for the Bern3D model parameter set. In all scenarios, the fully coupled model
configuration, including the major global carbon reservoirs (atmosphere, terrestrial biosphere, ocean, and sediments),
is used.

| Scenario | Initial conditions | Paleoclimate forcing | Tunable parameters: scale factor at LGM | | | | |
|---|---|---|---|---|---|---|---|
| | | | $\tau$ | $K_V$ | $k_w$ | $rr$ | $\ell_{POC}$ |
| MOD | PI | - | - | - | - | - | - |
| PAL | Glacial | Orb-GHG-Ice | - | - | - | - | - |
| CIRC | Glacial | Orb-GHG-Ice | 0.4 | 0.4 | - | - | - |
| VENT | Glacial | Orb-GHG-Ice | 0.4 | 0.4 | 0.4 | - | - |
| VENTx | Glacial | Orb-GHG-Ice | 0.4 | 0.4 | 0.0 | - | - |
| BIO | Glacial | Orb-GHG-Ice | - | - | - | 0.8 | 1.2 |
| PHYS-BIO | Glacial | Orb-GHG-Ice | 0.7 | 0.7 | 0.7 | 0.7 | - |
| PHYS-BIOx | Glacial | Orb-GHG-Ice | 0.8 | 0.8 | 0.8 | 0.8 | 1.2 |



Table 2. Production rate estimates in relative units inferred from three fundamentally different reconstruction methods:
geomagnetic field data from marine sediments, [10]Be and [36]Cl measurements in polar ice cores, and model-based
deconvolution of atmospheric $\Delta^{14}$C. Laj00, Laj04, Now13, and Chn18 refer to the paleointensity-based reconstructions
of Laj et al. (2000), Laj et al. (2004), Nowaczyk et al. (2013), and Channell et al. (2018), respectively. Adp18 refers
to the ice-core [10]Be-based reconstruction of Adolphi et al. (2018). Int13 and Hul18 refer to the model-based
reconstructions from this study, using the IntCal13 calibration curve (Reimer et al., 2013) and the new Hulu Cave
$\Delta^{14}$C dataset (Cheng et al., 2018). The bold numbers show the mean production rates during the last glacial (50 to 18
kyr BP).

| Time (kyr BP) | Mean production rate (relative units) | | | | | | |
|---|---|---|---|---|---|---|---|
| | Laj00 | Laj04 | Now13 | Chn18 | Adp18 | Int13 | Hul18 |
| 50 to 42 | 1.08 | 1.04 | 1.12 | 1.08 | 1.01 | 1.23 | 1.14 |
| 42 to 37 | 1.57 | 1.56 | 1.71 | 1.36 | 1.44 | 1.45 | 1.67 |
| 37 to 32 | 1.19 | 1.09 | 1.35 | 0.98 | 1.10 | 1.25 | 1.28 |
| 32 to 22 | 1.22 | 1.15 | 1.29 | 0.92 | 0.99 | 1.31 | 1.31 |
| 22 to 18 | 1.31 | 1.20 | 1.17 | 0.81 | 0.98 | 1.11 | 1.11 |
| **50 to 18** | **1.25** | **1.18** | **1.31** | **1.01** | **1.08** | **1.28** | **1.29** |


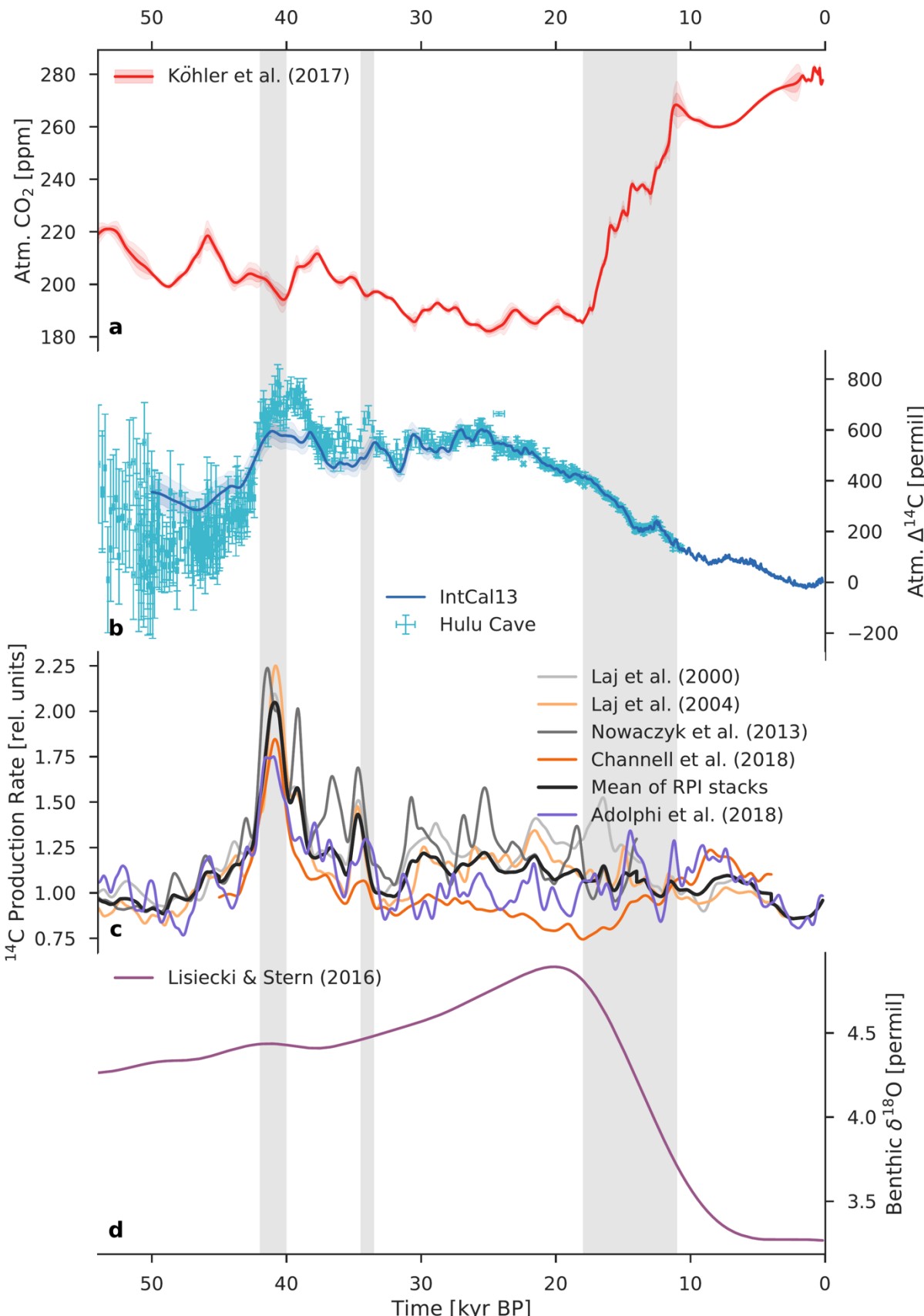

Fig. 1. Comparison of various paleoclimate records for the last 54 kyr. (a) Atmospheric $CO_2$ from the data compilation of Köhler et al. (2017). The light red envelope shows the uncertainty ($2\sigma$). (b) Atmospheric $\Delta^{14}C$ reconstructed from $^{14}C$ measurements on tree rings, plant macrofossils, speleothems, corals, and foraminifera. The light blue envelope shows the uncertainty ($2\sigma$) in the IntCal13 calibration curve (Reimer et al., 2013), whereas the Hulu Cave data (Cheng et al., 2018) are shown with error bars ($1\sigma$). Hulu Cave data are consistent with IntCal13 between ~10.6 and 33.3 kyr BP. For both records $\Delta^{14}C$ values were adjusted to the presently accepted value of the radiocarbon half-life (5700 years). (c) $^{14}C$ production rate in relative units reconstructed from paleointensity data (Laj et al., 2000; Laj et al., 2004; Nowaczyk et al., 2013; Channell et al., 2018) and from polar ice-core $^{10}Be$ fluxes (Adolphi et al., 2018). The heavy dark gray line is the mean paleointensity-based $^{14}C$ production rate. (d) Global benthic $\delta^{18}O$ stack, a proxy for ice volume, from Lisiecki and Stern (2016). Three vertical light gray bars indicate the Laschamp excursion (~41 kyr BP), when the Earth's geomagnetic dipole field intensity was close to zero, the Mono Lake geomagnetic excursion (~34 kyr BP), and the last glacial termination (~18 to 11 kyr BP), respectively.

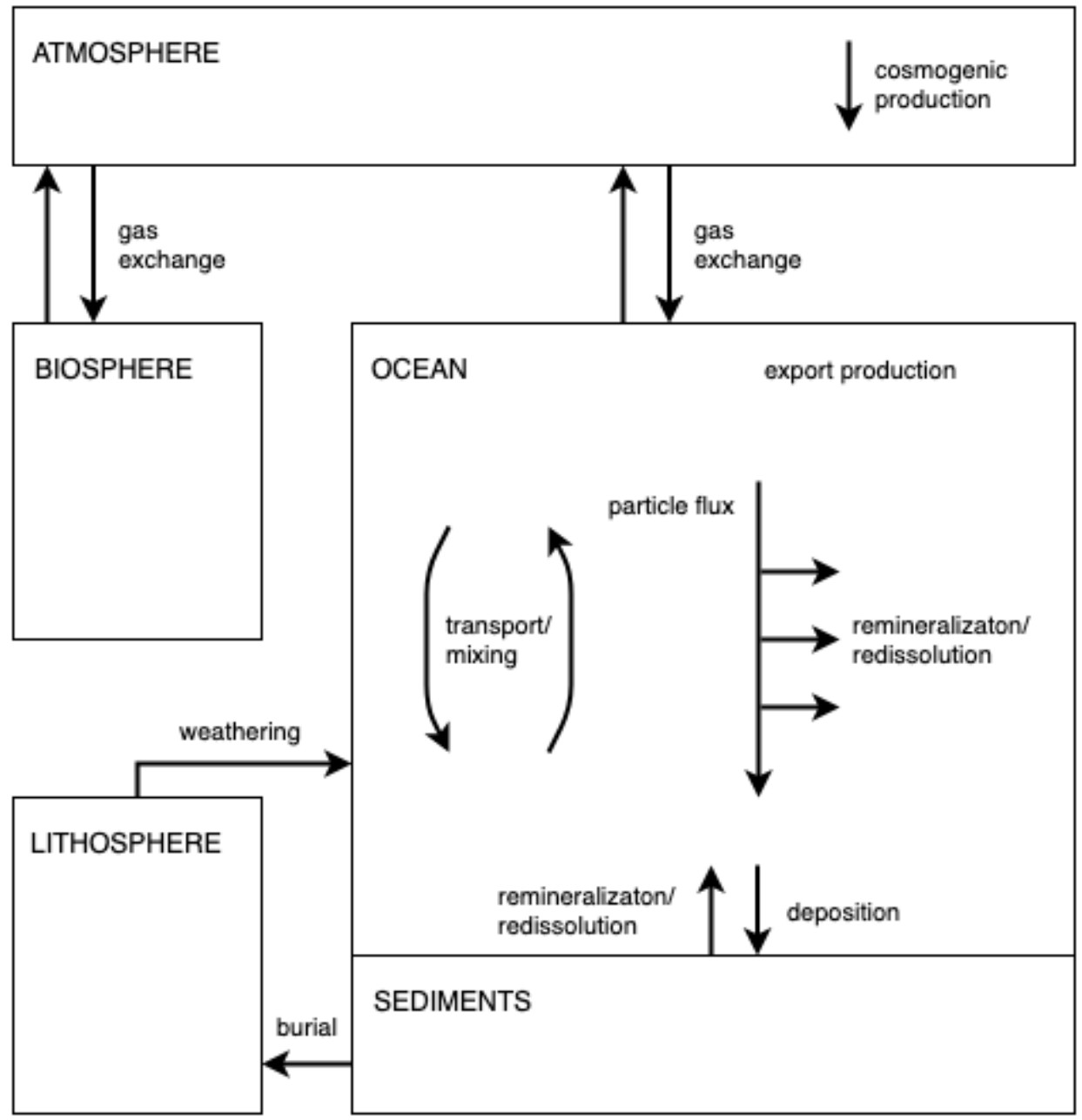


Fig. 2. Schematic diagram of the Bern3D carbon cycle model. The fully coupled model includes the major global
carbon reservoirs (atmosphere, terrestrial biosphere, ocean, and sediments) and the exchange fluxes between them.
Biogeochemical processes, namely, air-sea gas exchange, biological export production, and particle flux through the
water column, are parameterized by refined OCMIP-2 formulations. Details concerning the model are provided in
Sect. 2 and Appendix A.


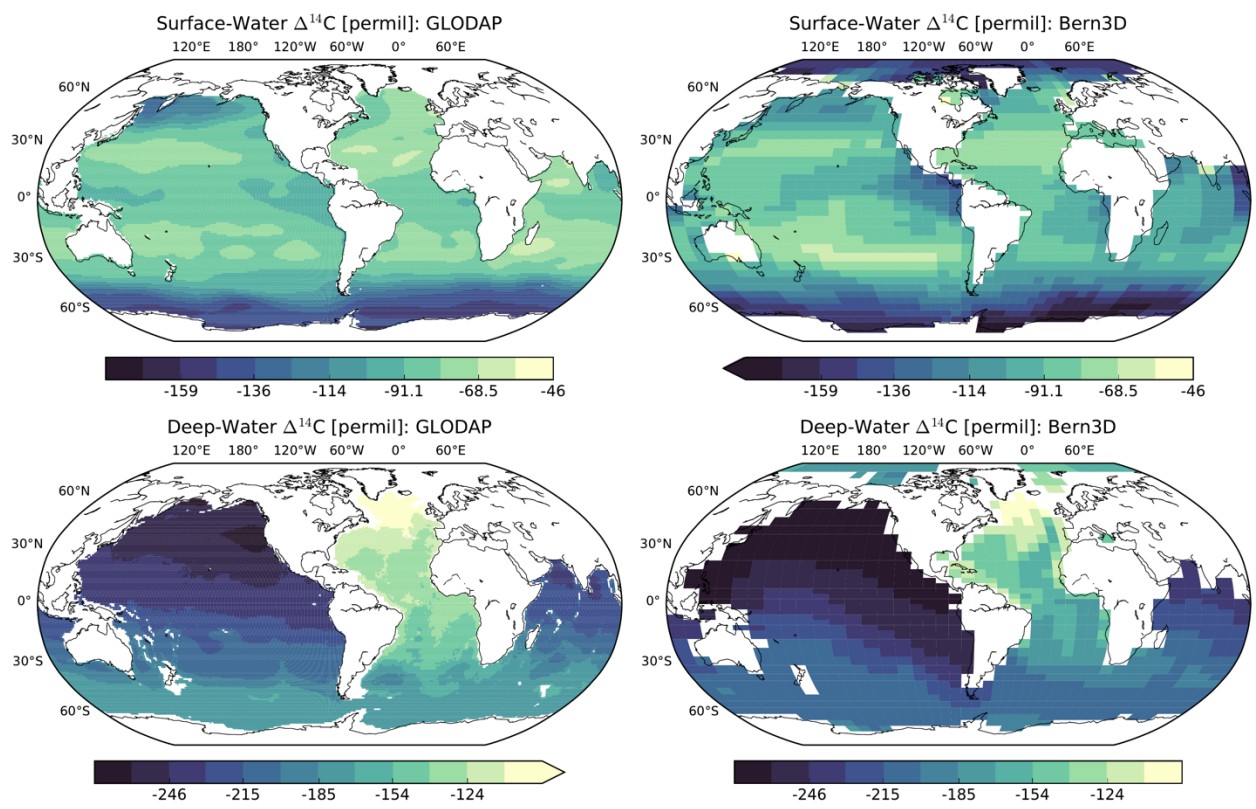

Fig. 3. Steady-state distribution of $\Delta^{14}C$ in the surface (< 100 m) and deep (> 1500 m) ocean for the preindustrial control run (right), compared to the distribution of $\Delta^{14}C$ based on the Global Ocean Data Analysis Project (GLODAP).

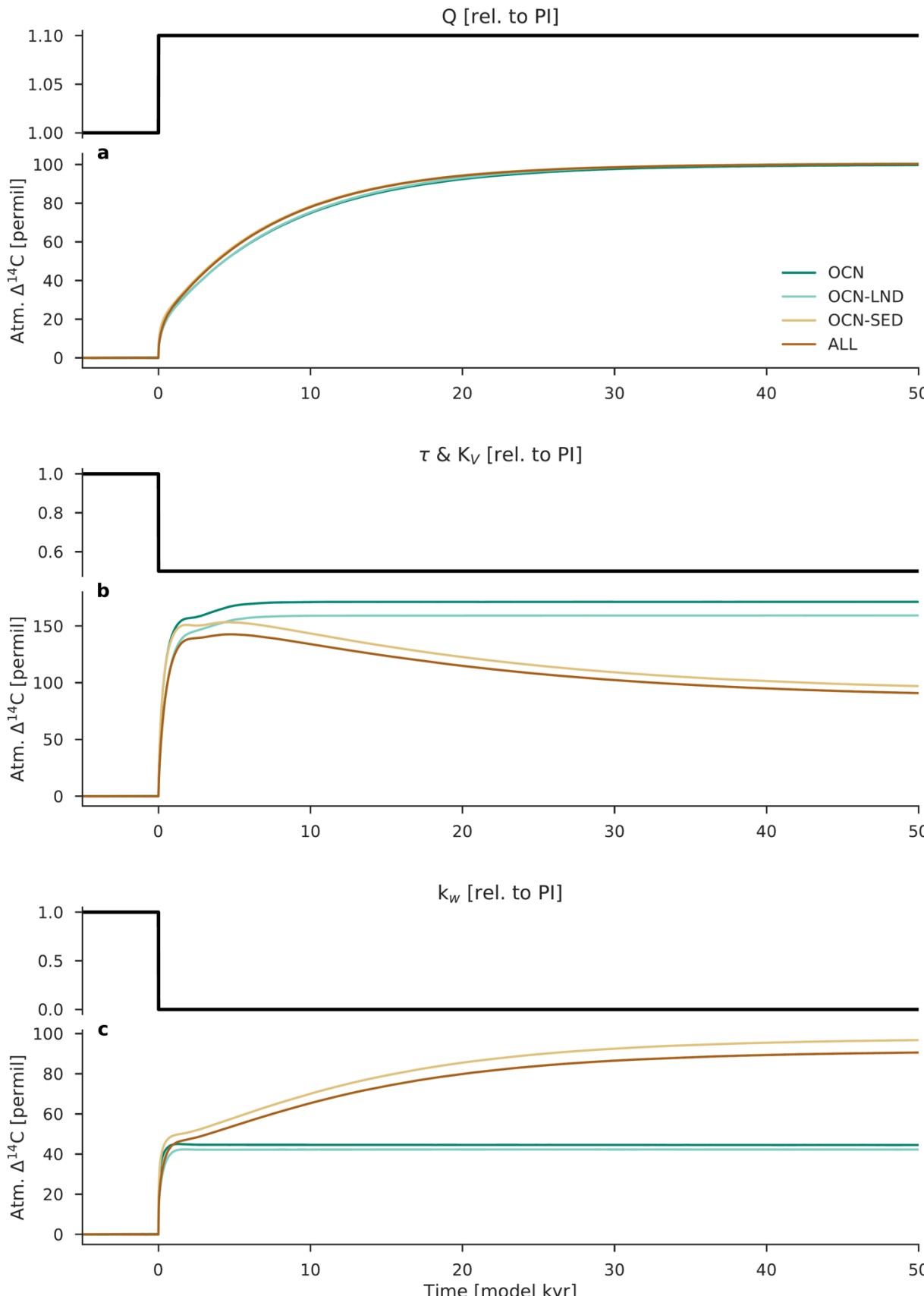


Fig. 4. Response of atmospheric $\Delta^{14}C$ to step changes in $^{14}C$ production, followed by step changes in the tunable model
parameters of the ocean carbon cycle. (a) $^{14}C$ production $Q$ is increased at time 0 from 100 to 110 percent of its
preindustrial value ("higher production" scenario). (b) Wind stress scale factor $\tau$ and vertical diffusivity $K_V$ are
decreased at time 0 from 100 to 50 percent of their preindustrial values ("reduced deep ocean ventilation" scenario).
(c) Gas transfer velocity $k_w$ is decreased at time 0 from 100 to 0 percent of its preindustrial value at the north ($> 60°N$)
and south ($> 48°S$) poles ("enhanced permanent sea ice cover" scenario). Four model configurations are considered.
The dark turquoise line shows the model results using the atmosphere–ocean (OCN) configuration, the light turquoise
line is the atmosphere–ocean–land (OCN-LND) configuration, the light brown line is the atmosphere–ocean–sediment
(OCN-SED) configuration, and the dark brown line is the atmosphere–ocean–land–sediment (ALL) configuration.




















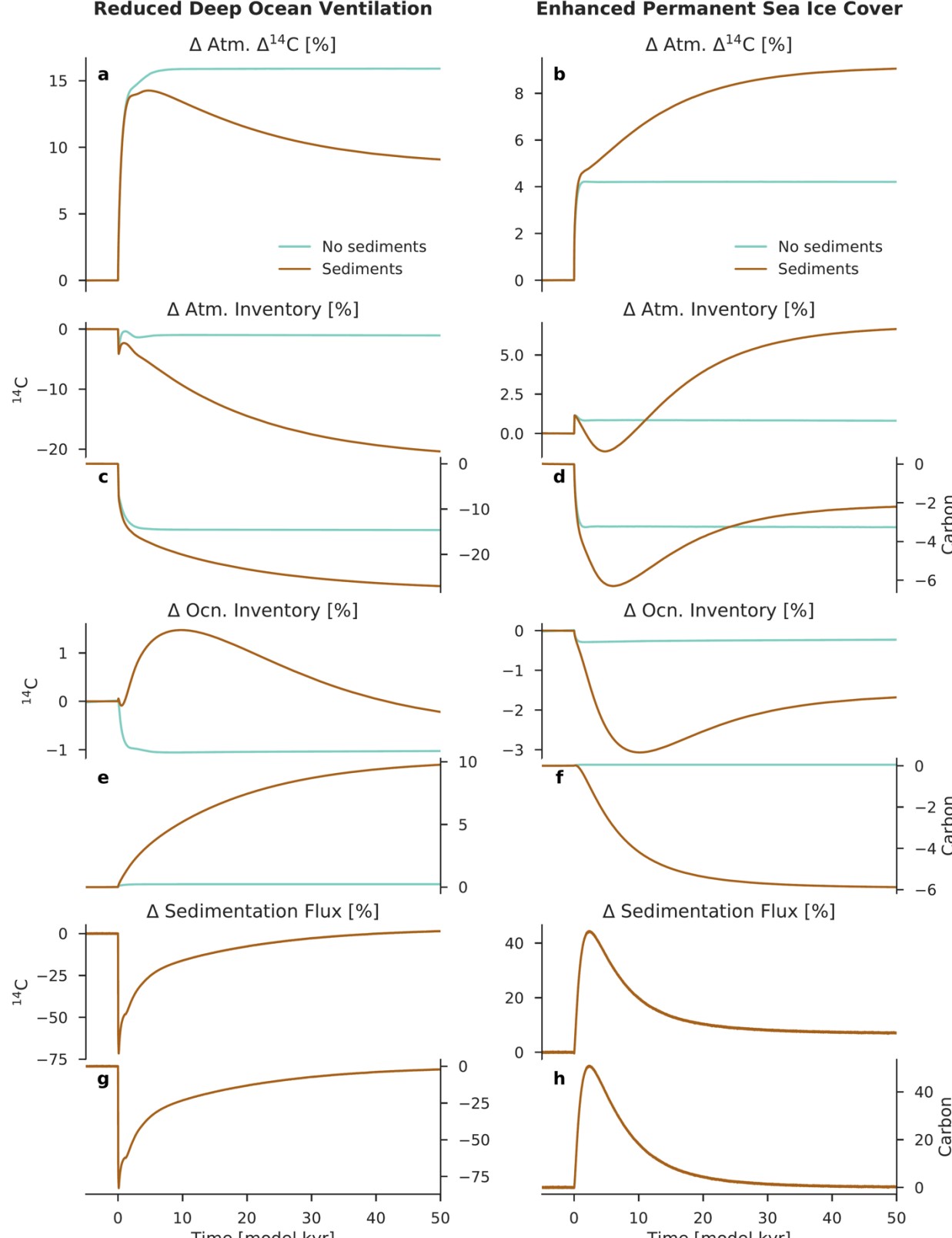


Fig. 5. Changes in carbon reservoir sizes and the sedimentation flux for the scenarios "reduced deep ocean ventilation"
(left) and "enhanced permanent sea ice cover" (right). The change in atmospheric $\Delta^{14}C$ is also shown (a, b). Anomalies
are expressed here as differences relative to the preindustrial steady state (in percent). Turquoise lines show the model
results using configuration OCN-LND (without sediments) and brown lines are configuration ALL (with sediments).
The y-axis on the left-hand side of each panel refers to changes in the $^{14}C$ inventory, whereas the y-axis on the right-
hand side of each panel refers to changes in the carbon inventory or flux.

























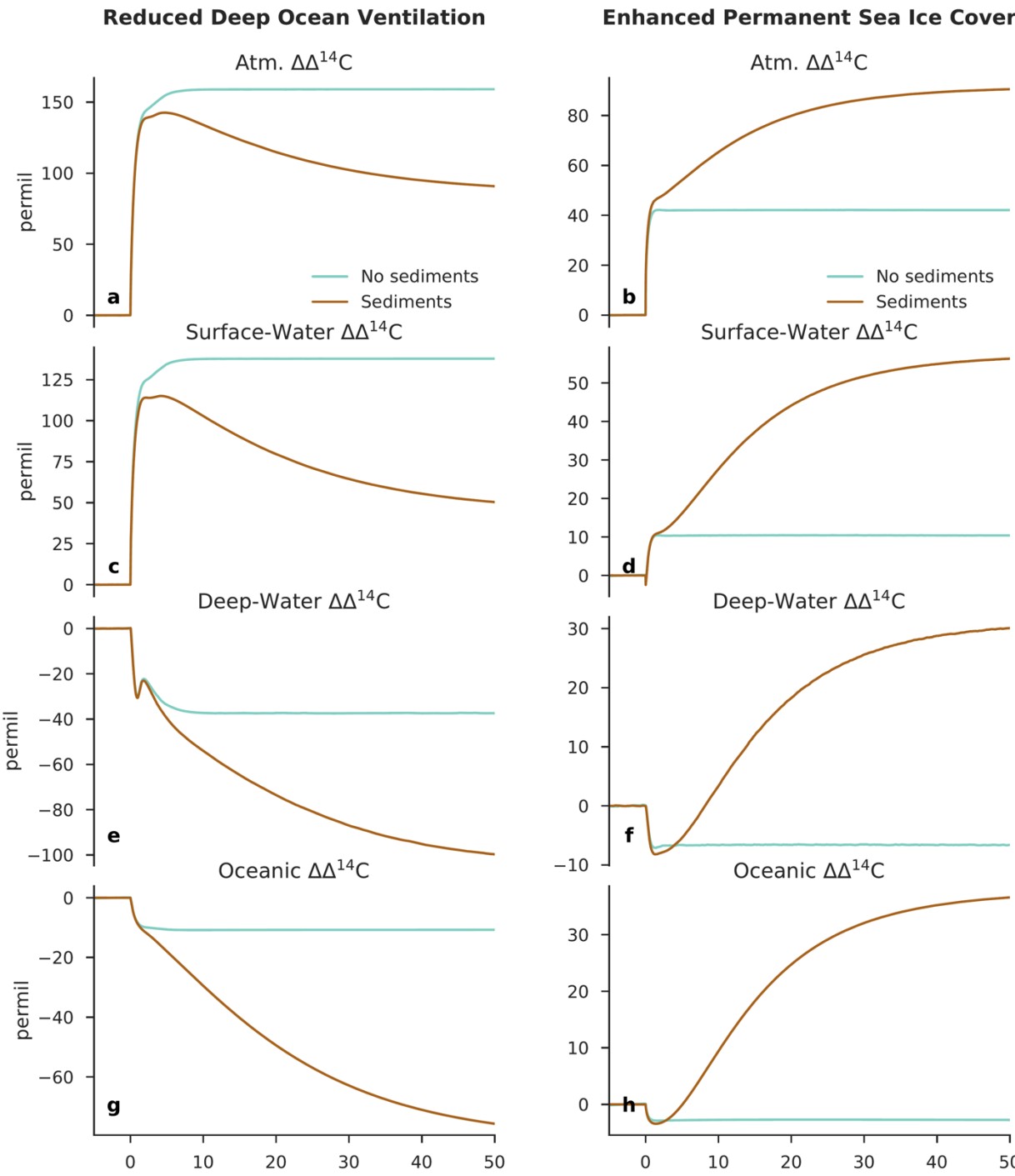

Fig. 6. Change in $\Delta^{14}C$ for the atmosphere, surface ocean, deep ocean, and global ocean for the scenarios "reduced
deep ocean ventilation" (left) and "enhanced permanent sea ice cover" (right). Anomalies are expressed here as
differences relative to the preindustrial steady state (in permil). Turquoise lines show the model results using
configuration OCN-LND (without sediments) and brown lines are configuration ALL (with sediments).

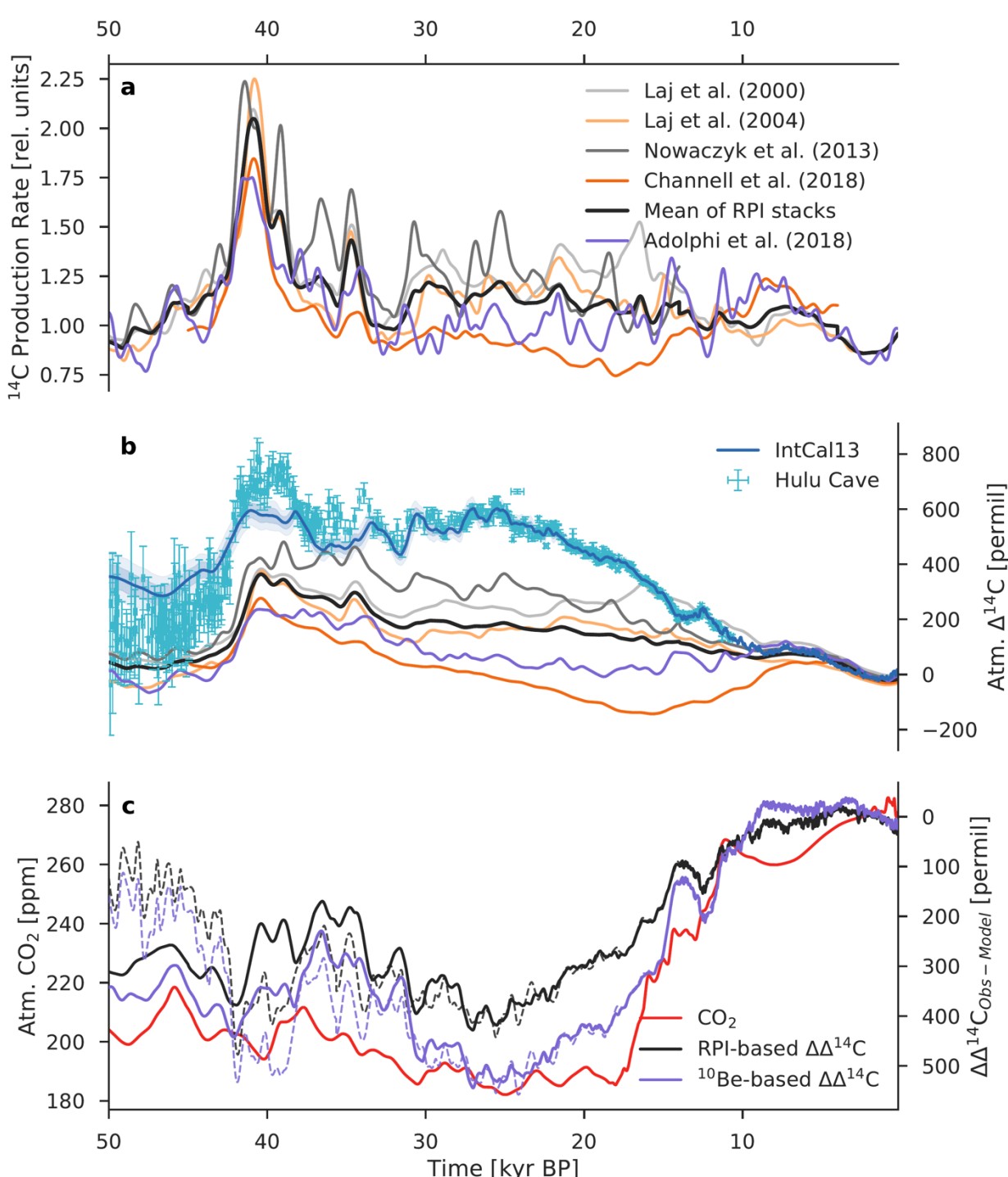

Fig. 7. Component of atmospheric $\Delta^{14}C$ variability caused by production changes alone. (a) Relative $^{14}C$ production rate as inferred from paleointensity data (gray) and from polar ice-core $^{10}Be$ fluxes (purple). The heavy dark gray line is the mean paleointensity-based $^{14}C$ production rate. (b) Modelled $\Delta^{14}C$ records based only on $^{14}C$ production changes, compared with the reconstructed IntCal13 and Hulu Cave $\Delta^{14}C$ records. The modelled records are given by scenario MOD that assumes a constant preindustrial carbon cycle. (c) Difference between reconstructed $\Delta^{14}C$ and model-simulated $\Delta^{14}C$ using averaged paleointensity data (RPI-based $\Delta\Delta^{14}C$; gray) and the ice-core $^{10}Be$ data of Adolphi et

al. (2018) ($^{10}$Be-based $\Delta\Delta^{14}$C; purple), compared with the atmospheric $CO_2$ record (red). Solid lines show the IntCal13–model difference, whereas dashed lines show the Hulu–model difference. The $\Delta\Delta^{14}$C curve indicates changes in $\Delta^{14}$C that can be attributed to some combination of carbon cycle changes, uncertainties in the reconstruction of the $^{14}$C production rate, and uncertainties in the IntCal13 and Hulu Cave $\Delta^{14}$C records.

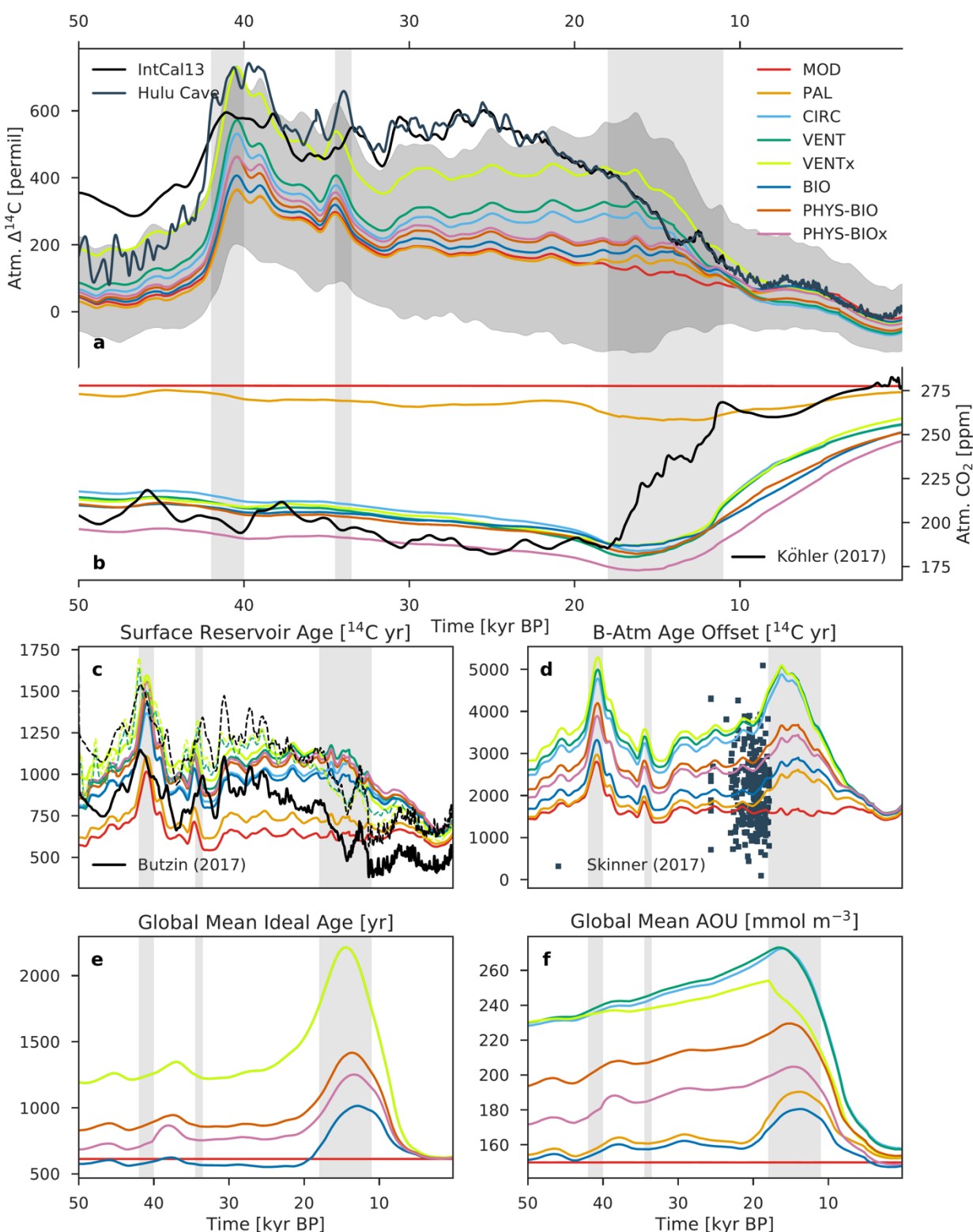


Fig. 8. Modelled records of atmospheric (a) $\Delta^{14}C$ and (b) $CO_2$, compared with their reconstructed histories (black and
dark blue lines). Also shown are modelled records of the global average (c) surface reservoir age and (d) B-Atm $^{14}C$
age offset, compared with a recent compilation of LGM marine radiocarbon data (dark blue squares) by Skinner et al.

(2017) and model-based surface reservoir age estimates between 50°N and 50°S (solid black line) and across all latitudes (dashed black line) from Butzin et al. (2017), as well as (e) ideal age and (f) apparent oxygen utilization (AOU). Colored lines show the results of model runs using the mean paleointensity-based $^{14}$C production rate and the eight different carbon cycle scenarios described in Sect. 2.4 and Table 1. The gray envelope in (a) shows the uncertainty (2σ) from all production rate reconstructions and carbon cycle scenarios, providing a bounded estimate of $\Delta^{14}$C change. The dashed colored lines in (c) show the surface reservoir age results from VENT and VENTx where atmospheric $\Delta^{14}$C and $CO_2$ are prescribed. Radiocarbon ventilation ages are expressed here as radiocarbon reservoir age offsets following Soulet et al. (2016) which are used extensively by the radiocarbon dating community.

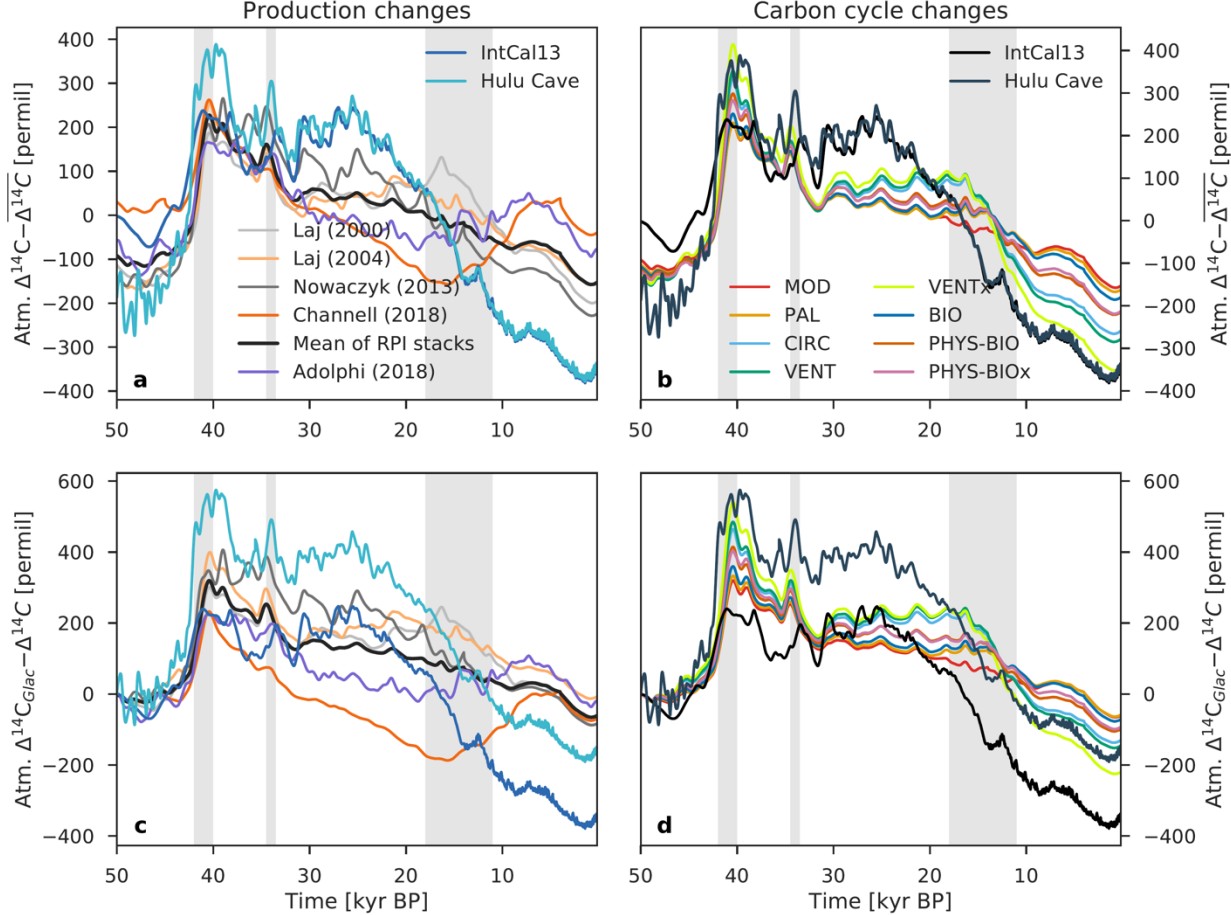

Fig. 9. Comparison of atmospheric $\Delta^{14}C$ variability caused by changes in the ocean carbon cycle (b, d) with production-driven changes in atmospheric $\Delta^{14}C$ using scenario MOD (a, c). For the analysis of carbon cycle changes, only the results of model runs using the mean paleointensity-based $^{14}C$ production rate are shown. The $\Delta^{14}C$ records in the upper panel (a, b) have been detrended by removing the mean, whereas the lower panel (c, d) shows $\Delta^{14}C$ anomalies expressed as differences relative to the $\Delta^{14}C$ value at 50 kyr BP. Three vertical light gray bars indicate the Laschamp (~41 kyr BP) and Mono Lake (~34 kyr BP) geomagnetic excursions, and the last glacial termination (~18 to 11 kyr BP).

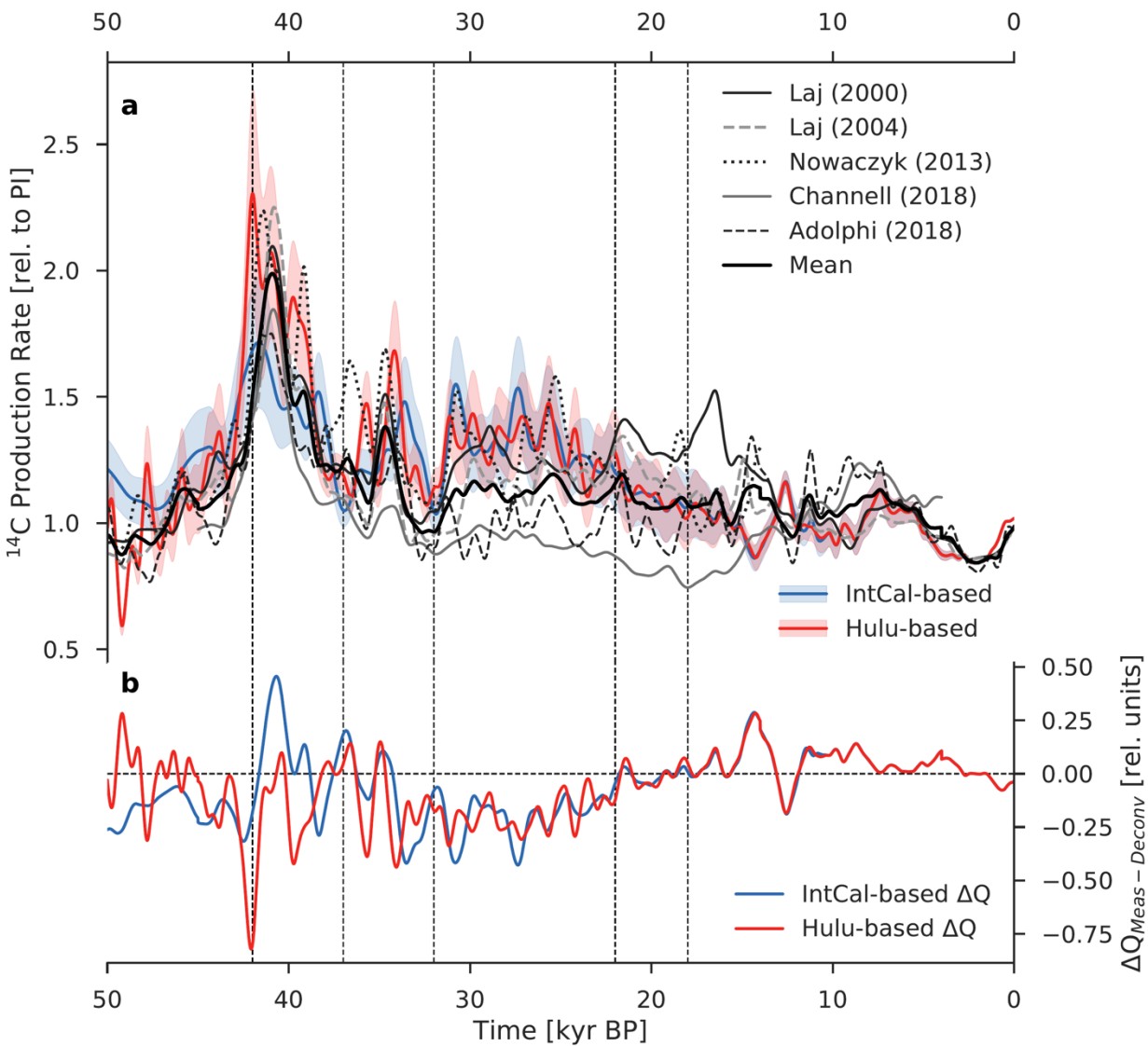

1474

Fig. 10. Comparison of $^{14}$C production rate estimates inferred from a deconvolution of the atmospheric $\Delta^{14}$C record and from paleointensity and ice-core $^{10}$Be data. (a) $^{14}$C production rate calculated as the sum of the modelled air-sea and atmosphere-land $^{14}CO_2$ fluxes and the reconstructed change in the atmospheric $^{14}$C inventory and loss of $^{14}$C due to radioactive decay (see Eq. [2]). Model-based $^{14}CO_2$ fluxes were obtained by forcing the Bern3D carbon cycle model with reconstructed variations in atmospheric $\Delta^{14}$C and $CO_2$ as well as seven different carbon cycle scenarios. Results of model runs using the IntCal13 calibration curve are shown in the light blue envelope ($2\sigma$), whereas the light red envelope ($2\sigma$) shows the results from simulations using the composite Hulu Cave (10.6 to 50 kyr BP) and IntCal13 (0 to 10.6 kyr BP) $\Delta^{14}$C record. The heavy black line is the mean of five available production rate reconstructions: Laj et al. (2000), Laj et al. (2004), Nowaczyk et al. (2013), Channell et al. (2018), and Adolphi et al. (2018). (b) Difference between the mean of the measurement-based production rate estimates (heavy black line) and estimates based on the deconvolution of the IntCal13 (IntCal-based $\Delta$Q; blue) and Hulu Cave (Hulu-based $\Delta$Q; red) $\Delta^{14}$C data.


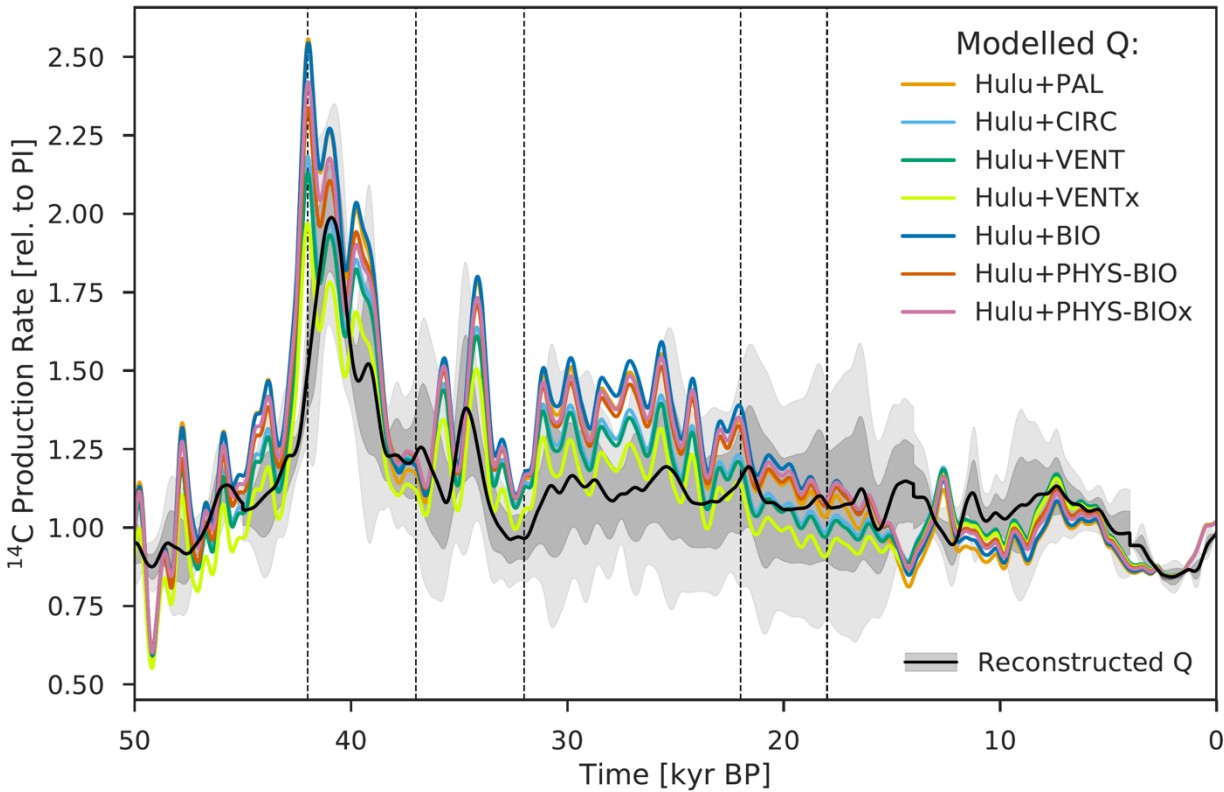


Fig. 11. Relative $^{14}$C production rate as inferred from the Bern3D model under seven carbon cycle scenarios (see Sect.
2.4). Estimates shown here are based on the composite Hulu Cave and IntCal13 $\Delta^{14}$C record. The black line is the
mean of the five production rate reconstructions shown in Fig. 10; the gray envelope shows its uncertainty (2σ).













