# Peer review of "Mysteriously high $\Delta^{14}$ C of the glacial atmosphere: Influence of 1 14C production and carbon cycle changes 2"

_Climate of the Past, 2019_

## Referee Comment (RC1) · Anonymous Referee #1 · 28 Jan 2020

This paper describes simulation experiments with the Bern3D model in order to understand past levels of atmospheric $\Delta^{14}$C reconstructed in the international effort IntCal13, especially how can be explained that the reconstructed values over the last 50 kyr went up to about 500-600‰. The conclusions made suggest that the consideration of sedimentation in the model is crucial for the understanding of $^{14}$C, and that reconstructed $^{14}$C production rates based on either geomagnetic field strength or $^{10}$Be and $^{36}$Cl from ice cores are not sufficient to produce with the Bern3D model atmospheric $\Delta^{14}$C that is in reasonable agreement with reconstructions. The authors therefore suggest, that alternatively $^{14}$C production rates based on their model output might be taken in the future, leaving the ultimate question why reconstructions and model results differ for

future efforts.

The study is timely and well set, and results are explained very well. I nevertheless have a few comments, which I ask to consider in a revision, which are partially fundamental. I believe in a necessary revision following especially my fundamental remarks below some of the more general remarks (especially in the abstract, introduction and conclusions) need to tone down and give also some more room for the shortcomings of the study itself (e.g. constant weathering rates, potentially missing silicate weathering, potentially missing volcanic $CO_2$ outgassing).

**Fundamental remarks:**

1. Weathering vs sedimentation: It is said that when sedimentation is included in the model the atmospheric $\Delta^{14}C$ strongly decreases in comparison to an atmosphere-ocean model version only. I believe this is naming the wrong process. My understanding of the model description is, that weathering is the process that brings $^{14}C$-free C into the system, so it is carbonate weathering, that is fundamental for the $^{14}C$ cycle. It is clear that once weathering input (of alkalinity and DIC to the ocean) is considered also sedimentation as sink needs to be implemented (otherwise the carbon cycle would run away with an ocean accumulating alkalinity, and subsequent changes to atmospheric $CO_2$ levels), but sedimentation is not the important process here that changes $\Delta^{14}C$. This might then also lead to a different name of the model configuration now called OCN-SED.

2. No details on weathering are given, but since it is said, that $^{14}C$-free C is entered via weathering I have to assume, this implies carbonate weathering. However, it need to be clarified (and maybe corrected?), that in carbonate weathering, 50% of the carbon that enters the ocean as weathering product (bicarbonate ion, $HCO_3^-$, which changes DIC and alkalinity in the ocean) comes from rocks

($^{14}$C-free), and 50% has its origin in atmospheric $CO_2$ with its atmospheric $^{14}$C-signature. For silicate weathering, also bringing $HCO_3^-$ to the ocean, 100% of the carbon has its origin in the atmosphere. Is silicate weathering considered? For details see, for example, Colbourn et al. (2013). Without checking on recent updates, I believe both silicate and carbonate weathering contributed about a similar amount of $HCO_3^-$ input into the ocean. At least in a study some years ago (Hartmann et al., 2009) in present day weathering the $CO_2$ consumption is twice as big in silicate than in carbonate weathering, but since in carbonate weathering 50% of the C has its origin from rocks, both processes should contribute about the same. Since weathering is the relevant process for this paper more details on its implementation in the model should be included. From the Appendix I understood, that weathering rates are constant in time, but please give their numbers, which would be especially of interest to other modellers doing similar things. Also consider in a discussion, that missing temporal changes in weathering rate might be one reason why reconstructed $\Delta^{14}$C (and $CO_2$) is not met with simulations. Having found, that the input of $^{14}$C-free carbon to the system is so important for an understanding of $\Delta^{14}$C brings me also to the quesion if $^{14}$C-free $CO_2$ outgassing from volcanos is considered, which might have similar effects on $^{14}$C. I understand that this has been investigated previously with the Bern3D model (Roth and Joos, 2012), but with focus on $^{13}$C. Maybe some more insights from previous simulations are possible here, at least in a discussion. At least please mention the applied $CO_2$ volcanic outgassing rates. Note, that there is a fundamental, analytical derived solution from the steady state assumption on volcanic $CO_2$ input being 50% of the $CO_2$ consumption by silicate weathering, which is of relevance for times longer than 100 kyr (briefly mentioned in Munhoven and François (1996) or in depth discussed on pages 80-81 of Munhoven (1997), http://www.astro.ulg.ac.be/~munhoven/en/PhDIndex.html). For shorter periods such as the last 50 kyr considered here, differences from this numbers are certainly possible, but this relationship gives a rough guideline, and

might explain long-term drifts in the C cycle, if not obeyed. Taken together, I have the impression, that no silicate weathering, and also no volcanic outgassing of $CO_2$ is considered here, which would indicate according to this theory no drift in the system, but also the missing of two important processes. If so, I am not saying, these should be implemented in the revision, but it needs to be stated clearly if and how they are (not) included. How does your weathering flux compare to others, e.g. Fig 7 in Brovkin et al. (2012) or Colbourn et al. (2013)?

3. Earlier simulation studies have shown, that to get the $^{14}$C cycle right, one needs to have the C cycle right as well. Köhler et al. (2006) has shown that previous studies (Beck et al., 2001; Hughen et al., 2004) focusing only on $^{14}$C, but showing no simulated $CO_2$, they therefore have very likely some deficits. For atmospheric $\Delta^{14}$C especially the air-sea gas exchange is important, which depends similarily on the gas exchange velocity ($k_w$, which is considered here in sensitivity experiments), but also on the $CO_2$ gradient between atmosphere and surface ocean. This implies that whenever simulated $CO_2$ differs from reconstructions there will also be an offset in simulated $\Delta^{14}$C from data. In a recent simulation effort for IntCal20 (the successor of IntCal13) the marine surface $\Delta^{14}$C has been simulated (Heaton et al., submitted). There, the importance of time-dependent changes in $CO_2$ has been as important for the simulated surface ocean $\Delta^{14}$C as that of climate change (temperature change, ocean circulation change etc), which via gas exchange would also feedback to atmosheric $\Delta^{14}$C. This is unpublished so far, but since it is submitted and will probably be available in due time I nevertheless mention it here.

4. The coauthor Florian Adolphi is also coauthor of the now submitted IntCal20 effort (updating the atmospheric $\Delta^{14}$C record), (Reimer et al., submitted) and should therefore be aware of the large changes which occur between IntCal13 and IntCal20, namely the amplitude of the $\Delta^{14}$C maxima around 40 kyr increases in IntCal20 towards the Hulu Cave numbers. Maybe this should be briefly discussed

in an outlook.

**Minor issues in chronological order:**

1. The decay constant of $^{14}$C used here is based on a halflife of $^{14}$C of 5700 yr (here) but of 5730 yr in Intcal13 (and IntCal20) which produced the atmospheric $\Delta^{14}$C record. If you consider the decay of $^{14}$C over 40 kyr (the time of the maximum in atm $\Delta^{14}$C) with either 5700 vs 5730 yr you get a 2.5% smaller number when based on 5700 yr, although the halflife time differed only by 0.5%. This difference is small when compared to the difference of IntCal13 and the Hule Cave data, but should nevertheless be mentioned.

2. line 173: Please be specific, how $^{14}$C is fractionation corrected.

3. line 180-181: Weathering is prescribed as constant input of DIC, but no $^{14}$C. See fundamental comment above, but no matter where the C of weathering comes from the input in the ocean should be a $HCO_3^-$, changing both DIC and alkalinity in the ocean. I hope this is only a too simplified description here, but has implemented correctly in the model. Please revise.

4. line 184: For the preindustrial spin-up $CO_2$ is fixed to 278.05 ppm. Why this number, would not 278 ppm do the same job? Is this OCMIP protocol? Also: What would be the internally calculated preindustrial $CO_2$? By prescribing $CO_2$ concentration during spin-up C is added or extracted from the simulated system, which might be a potential source of bias. How long is the spin-up time?

5. line 229: Consider citing the underlying ice core paper, from which the greenhouse gases splines provided by Köhler et al (2017) have been calculated.

6. line 232: Global benthic $\delta^{18}O$ is not only a global ice volume proxy, but has also a considerable contribution from deep ocean temperature, see papers of the van de Wal group from Utrecht Universitiy on the deconvolution (e.g. Bintanja and van de Wal, 2008). Taken this knowledge into consideration, would this change your approach how sea level is changing? You might also discuss how different sea level reconstructions vary, e.g. see three different sea level reconstructions in Fig 1f of Hasenclever et al. (2017), and what this uncertainty in sea level might introduce into your approach.

7. lines 223-236: Model description says that greenhouse gas radiative forcing has been taken from data, thus I assume that $CO_2$ seen by the carbon cycle is never prescribed, but always model-internally calculated. Please state this explicitly (or the correct version of this sentence, if this was not the case). However, you might also consider one scenario in which $CO_2$ is prescribed for the C cycle from data (similar as in Butzin et al., 2017, in which atmospheric $CO_2$ and $\Delta^{14}C$ has been prescribed by data), since this would bring your simulated C cycle as close to observations as probably possible, which might further reduce the bias in $^{14}C$ (see also fundamental remarks above).

8. Obtained surface reservoir ages (Fig 8c) might be compared with data and other models, e.g. see Butzin et al. (2017); Skinner et al. (2019). Benthic-atmospheric offsets (Fig 8d) might be compared for the LGM with the data compilation of Skinner et al. (2017). Note, surface reservoir ages might vary a lot as function of latitude, so this Fig 8c needs more information on averaging; even better: might be revised and thus restricted to sea ice-free areas only.

9. Please state somewhere the absolute (PI) values of those parameters which are changed in your sensitivity experiments, maybe in Table 1?

10. Table 2: $^{14}C$ production rates is given in relative units, relative to what? Probably preindustrial state. I also do not remember if the $^{14}C$ production rate in absolute

numbers is once given in the manuscript for preindustrial state, please insert somewhere.

11. Fig 4 captions does not need a description of the different colors of the lines, since a legend is given in the figures themselves.

**References**

Beck, J. W., Richards, D. A., Edwards, R. L., Silverman, B. W., Smart, P. L., Donahue, D. J., Hererra-Osterheld, S., Burr, G. S., Calsoyas, L., Jull, A. J. T., and Biddulph, D.: Extremely large variations of atmospheric $^{14}$C concentration during the Last Glacial Period, Science, 292, 2453–2458, 2001.

Bintanja, R. and van de Wal, R. S. W.: North American ice-sheet dynamics and the onset of the 100,000-year glacial cycles, Nature, 454, 869–872, doi:10.1038/nature07158, 2008.

Brovkin, V., Ganopolski, A., Archer, D., and Munhoven, G.: Glacial $CO_2$ cycle as a succession of key physical and biogeochemical processes, Climate of the Past, 8, 251–264, doi:10.5194/cp-8-251-2012, 2012.

Butzin, M., Köhler, P., and Lohmann, G.: Marine Radiocarbon Reservoir Age Simulations for the Past 50,000 Years, Geophysical Research Letters, 44, 8473–8480, doi:10.1002/2017GL074688, 2017.

Colbourn, G., Ridgwell, A., and Lenton, T. M.: The Rock Geochemical Model (RokGeM) v0.9, Geoscientific Model Development, 6, 1543–1573, doi:10.5194/gmd-6-1543-2013, 2013.

Hartmann, J., Jansen, N., Dürr, H. H., Kempe, S., and Köhler, P.: Global $CO_2$-consumption by chemical weathering: What is the contribution of highly active weathering regions?, Global and Planetary Change, 69, 185–194, doi:10.1016/j.gloplacha.2009.07.007, 2009.

Hasenclever, J., Knorr, G., Rüpke, L., Köhler, P., Morgan, J., Garofalo, K., Barker, S., Lohmann, G., and Hall, I.: Sea level fall during glaciation stabilized atmospheric $CO_2$ by enhanced volcanic degassing, Nature Communications, 8, 15 867, doi:10.1038/ncomms15867, 2017.

Heaton, T. J., Köhler, P., Butzin, M., Bard, E., Reimer, R. W., Austin, W. E. N., Ramsey, C. B., Grootes, P. M., Hughen, K. A., Kromer, B., Reimer, P. J., Adkins, J., Burke, A., Cook, M. S., Olsen, J., and Skinner, L. C.: Marine20 — the marine radiocarbon age calibration curve (0–55,000 cal BP), Radiocarbon, submitted.

Hughen, K., Lehman, S., Southon, J., Overpeck, J., Marchal, O., Herring, C., and Turnbull, J.: $^{14}$C activity and global carbon cycle changes over the past 50,000 years, Science, 303, 202–207, doi:10.1126/science.1090300, 2004.

Köhler, P., Muscheler, R., and Fischer, H.: A model-based interpretation of low frequency changes in the carbon cycle during the last 120 000 years and its implications for the reconstruction of atmospheric $\Delta^{14}$C, Geochemistry, Geophysics, Geosystems, 7, Q11N06, doi:10.1029/2005GC001228, 2006.

Munhoven, G.: Modelling glacial-interglacial atmospheric $CO_2$ variations: the role of continental weathering, Ph.D. thesis, Université de Liège, Liège, Belgium, 1997.

Munhoven, G. and François, L. M.: Glacial-interglacial variability of atmospheric $CO_2$ due to changing continental silicate rock weathering: a model study, Journal of Geophysical Research, 101(D16), 21 423–21 437, 1996.

Reimer, P. J., Austin, W. E. N., Bard, E., Bayliss, A., Blackwell, P. G., Bronk Ramsey, C., Butzin, M., Cheng, H., Edwards, R. L., Friedrich, M., Grootes, P. M., Guilderson, T. P., Hajdas, I., Heaton, T. J., Hogg, A. G., Hughen, K. A., Kromer, B., Manning, S. W., Muscheler, R., Palmer, J. G., Pearson, C., van der Plicht, H., Reimer, R. W., Richards, D. A., Scott, E. M., Southon, J. R., Turney, C. S. M., Wacker, L., Adophi, F., Büntgen, U., Capano, M., Fahrni, S., Fogtmann-Schulz, A., Friedrich, R., Köhler, P., Kudsk, S., Miyake, F., Olsen, J., Reinig, F., Sakamoto, M., Sookdeo, A., and Talamo, S.: The IntCal20 Northern Hemisphere radiocarbon age calibration curve (0–55 kcal BP), Radiocarbon, submitted.

Roth, R. and Joos, F.: Model limits on the role of volcanic carbon emissions in regulating glacial-interglacial $CO_2$ variations, Earth and Planetary Science Letters, 329 - 330, 141 – 149, 2012.

Skinner, L. C., Primeau, F., Freeman, E., de la Fuente, M., Goodwin, P. A., Gottschalk, J., Huang, E., McCave, I. N., Noble, T. L., and Scrivner, A. E.: Radiocarbon constraints on the glacial ocean circulation and its impact on atmospheric CO2, Nature Communications, 8, 16 010 EP, doi:10.1038/ncomms16010, 2017.

Skinner, L. C., Muschitiello, F., and Scrivner, A. E.: Marine Reservoir Age Variability Over the Last Deglaciation: Implications for Marine Carbon Cycling and Prospects for Regional Radiocarbon Calibrations, Paleoceanography and Paleoclimatology, 34, 1807–1815, doi:10.1029/2019PA003667, 2019.

---

## Referee Comment (RC2) · Luke Skinner (Referee) · 27 Feb 2020

This is a highly thought provoking, and a substantial, study that tackles a long standing and important puzzle. Arguably, if one wanted an illustration of just how much we have yet to resolve regarding global carbon cycle change on glacial-interglacial and millennial timescales, one need look no further than an attempted explanation of past atmospheric carbon isotope (radiocarbon and d13C) variability. This study demands some stamina of the reader; however the effort is rewarded. The study clarifies a number of fundamental principles concerning radiocarbon cycling and the controls on atmospheric radiocarbon across a range of equilibration/waiting times, and sheds new light

on the role of sedimentary fluxes in setting atmospheric radiocarbon activity. Although the latter point is not completely new (e.g. Kohler et al., 2006), and the basic principles of radiocarbon inventory balance are well established, I do not know of any previous systematic exploration of the effects of variable sedimentary outputs on atmospheric radiocarbon.

Probably the main drawback of this work is that a full sensitivity study of the sediment cycle is not provided, and that only sedimentary *feedbacks* on the simulated ocean dynamical/gas exchange changes are explored. We therefore do not get a full 'map' of the expected impacts of variable sediment fluxes (e.g. including reconstructed changes in sedimentary organic carbon and carbonate fluxes over the last glacial cycle). It is natural, and I think a strength of the study, that after mulling over this manuscript a whole host of additional model scenarios come to mind, including e.g. different forcing functions (e.g. scaling tuned parameters to atmospheric CO2 instead of benthic d18O), or different deconvolutions (e.g. of carbonate/POC export, instead of radiocarbon production rates). However, it would not be reasonable to suggest such alternatives as revisions, and therefore I would suggest that this study is ready for publication in Climate of the Past, perhaps subject to technical revisions, in order to give the authors a chance to make small changes in light of comments they receive. Ultimately, i think that the effort required of the reader by this manuscript would be even more greatly rewarded if, by way of conclusion, the authors included a more specific list of the observations/tests that could help to finally resolve the 'atmospheric D14C puzzle', in light of this study's findings.

Below I provide a list of specific comments that I think might be useful to the authors in revising their manuscript for publication.

1. Line 11: I feel that the term 'mystery interval' has become current without having a particularly clear meaning; it seems to be used to refer to a chronozone, for which there already is a name (Heinrich Stadial 1, etc...). Furthermore, the 'mysterious' part of the interval seems to be perceived differently by different people; is it the atmospheric radiocarbon decline, the proposed lack of marine radiocarbon activity increase, the entire 'mystery' of deglaciation? Not everyone shares the same notions regarding such 'mysteries', particularly regarding the marine radiocarbon inventory change, against which the term 'mystery interval' seems to have been directed. I would like to stick my neck out and suggest that this term has served its purpose in stimulating interest in a topic, and no longer serves a purpose for clear communication of a specific idea. I would therefore propose that the authors refer to other more clearly established chronozone designations, or even dates if these are trusted sufficiently.

2. Line 18: I think the word "more" can be dispensed with, here and elsewhere. One wonders: more than what?

3. Line 40: here and throughout the manuscript I was not sure whether "millennial-scale" was a helpful designation, as it made me think of variability associated with Dansgaard-Oeschger events. Perhaps the term, or another such as "short term" etc..., can be defined clearly when first used?

4. Line 60: same thoughts as above regarding the term "mystery interval"; if it is coincident with HS1, then we should use that term instead I think. At the time of the Broecker and Barker (2007) study there was proposed to be a lack of evidence for a radiocarbon depleted ocean interior at the LGM, and a subsequent increase in its radiocarbon activity; however, this is arguably no longer the case.

5. Line 67: probably best to be more specific, e.g. "...used only high accumulation sites, and square barrel gravity cores with minimal sediment disturbance.."

6. Line 72, last sentence of the paragraph: I don't mean to suggest that ther is anyh I gotta incorrect about this sentence, but I found this to be an odd way of phrasing things. To me there is one question, "why was atmospheric radiocarbon activity so high during the last glacial (including well after the Laschamp excursion)", which entails a subsidiary question, "how much did production changes contribute to this elevated atmospheric radiocarbon activity".

7. Line 95: I would say that the time required for ocean ventilation is not "up to", but rather "over" 1000yrs. Perhaps Primeau (2005) can be referenced for this. 8. Line 100: multi-millennial timescales?

9. Line 106: Andrey Ganopolski would disagree (see Ganopolski et al., CP, 2017). Perhaps this statement should be modified to say that it is currently not possible to do so without the use of any parameterisations of key processes, or something more specific?

10. Line 113: this sentence seems to suggest that the main proposals for explaining glacial-interglacial CO2 involve exchanges with the solid earth, but this is not really true. Arguably, as has been sketched out many times before, including in a recent review (Galbraith 2020), the "ingredients" for glacial-interglacial CO2 change are well accounted for, it is their 'calibration' and organisation within an orbital pacing framework that remains elusive.

11. Line 118: in idealised settings..

12. Line 122: here and throughout the manuscript it would be best to suffix D14Catm, so that we know what reservoir is referred to.

13. Line 125: is it not more accurate to state that the production rate is inferred from an atmospheric radiocarbon budget, combined with a range of hypothetical radiocarbon and carbon cycle scenarios?

14. Line 159: air-sea equilibration times are very different, which is potentially important...

15. Line 169: perhaps Stuiver et al. 1978 should be referenced.

16. Line 170: would it be clearer to state that DI14C is simulated, separately from DIC?

17. Line 189: I wonder if this is not a major part of the whole problem with simulating atmospheric radiocarbon in the past? If the modern (pre-industrial) state is in fact far

from equilibrium then this would mean that production rates are all miscalibrated. Why not explore the possibility that production rates are higher than required for equilibrium, e.g. due to ongoing equilibration of sedimentation following the deglaciation and early Holocene? It seems to me that the very conclusions of this study require that this be explored as a possibility. More specifically, and perhaps I am not getting this right.. we might expect that, following the expansion of the terrestrial biosphere during the Holocene (and the removal of carbon from the atmosphere-ocean system, causing a slow reduction of 'young' carbonate sediment output from the ocean), the radiocarbon inventory of the ocean and atmosphere should be on a slow disequilibrium downward trend, so that a higher radiocarbon production would be needed to get today's radiocarbon activity as an equilibrium state. Is that correct? Or is it the opposite? In any event, one has a sneaking suspicion that this sort of thing might be important here.

18. Line 220: "...levels, given available 14C production scenarios."

19. Line 244: Why was benthic d18O chosen? It is a smooth, slow function that lags behind most of the climatic processes that were important for the carbon cycle. Although it might seem circular, I don't think it is any more ad hoc to scale these parameters to atmospheric CO2 instead.. having rapid jumps in HS1 and the YD, and a faster change than benthic d18O, might help with getting the deglacial CO2 change 'right' (for parameterised reasons).

20. Line 254: the cited study is based entirely on the 'plateau tuning' approach, which may be questioned. Perhaps best to also cite Skinner et al. (2017) who showed that the LGM ocean was 'older' pretty conclusively with a range of other data.

21. Line 283: It seems crucially important to me that the 10Be and 36Cl flux records from the ice cores are NOT consistent with the final age scale that they are all placed on. As far as I can tell from Adolphi et al. (2018), the ice core data were converted to fluxes based on each ice core's individual age scale, and then they were all placed on the GICC05 age-scale, whereas Channell et al. (2018) argued that this age scale

implies very different fluxes. Surely the ice core cosmogenic nuclide data ALL need to be placed on the same age scale and THEN the fluxes should be calculated and 'stacked'. I think this is a really crucial thing, and I am really confused as to why the specialists working with these isotope records take a different approach that surely produces incorrect fluxes. A basic test I would propose is: are the individual ice core flux records consistent with the accumulation rates that are implied for each ice core by the GICC05 age scale? If not, they need to be corrected, surely. I suspect this will only make matters worse for reconciling everything, but it is still important to consider carefully.

22. Line 355: note again that this conflicts with the premise that the modern state is at equilibrium!

23. Line 448: my intuition tells me that air-sea has exchange may have a small effect, but depending on the circulation state. Is it not possible that changes in air-sea exchange might combine non-linearly with particular changes in the circulation geometry?

24. Line 460: Although I see why the authors try to wiggle free from resolving the deglacial CO2 problem, I think it is it entirely possible to set it aside, and I also think it is basically not true that the study deals only with the glacial portion of the record. It is the glacial versus interglacial amplitude of atmospheric D14C that is of concern, and therefore the change across the deglaciation is entirely relevant! In fact, as suggested below, I would propose provocatively that this study shows that atmospheric radiocarbon can be explained reasonably well up until the deglaciation, and that it is the modern radiocarbon activity that defies explanation. I wonder what the authors think of this contention.

25. Line 537, the discussion of simulated B-Atm values: why do the authors not refer at all to published data for comparison? The compilation of Skinner et al. (2017) estimated, with the available data, that the global average ageing of the ocean at the

LGM was 'only' ~689 14C years. This is relevant here, and indeed it would suggest that all of the model scenarios produce rather extreme outcomes as compared to available data.

26. Line 545: I think it is worth specifying in what ways these indirect methods are also potentially inaccurate, due to different processes affecting e.g. oxygen and radiocarbon.

27. Line 567: ..is a dedicated 'control knob', in the model.

28. Line 605: viewed as tentative, perhaps. The viewing is not tentative; the results are.

29. Line 676: is it worth stating by how much this polar bias would have to be in order to reconcile everything? Is that magnitude reasonable?

30. Line 703: in this paragraph the realism of the implied sea ice changes is discussed, but again no mention is made of what existing marine radiocarbon data imply. These are really important constraints to mention, surely.

31. Line 726: I couldn't help but feel that the conclusion of the study might be more hard hitting if we had a more specific 'shopping list' of things that could help to resolve this puzzle. For example, constraining the global marine radiocarbon inventory change across the deglaciation, estimating any gradient in cosmogenic nuclide production across latitudes (i.e. polar bias, perhaps from tropical ice cores?), estimates of global carbonate/POC export rates (which already exist incidentally; Cartapanis et al., 2016; 2018), etc...

32. Table 1: it would be helpful to specify here which simulations have active sediments included. Incidentally, why was the rain ratio changed in one simulartion?

33. Fig 3, caption: I think it is more mathematically correct to state <100m and >1500m, no?

34. Fig 7, caption, line 1203: I think it would be helpful to state "...using the mean reconstructed palaeointensity.."

35. Fig 8: shouldn't all the simulated D14Catm traces start at the same value and end at different values? Although this might look nasty, it suggests a different outlook in my view. Incidentally, the outputs in plots c and d are obvious candidates for comparison with existing data (e.g. Skinner et al., 2019, 20176), perhaps for a future study if not this one.

36. Fig 9: this is a fascinating figure, though I find it slightly problematic. First, what is the rationale for normalizing to the average D14Catm value 0-50ka? I think that plots a and b should be replaced with normalization to the final 'modern' value, and that plots c and d should be extended up to the present. The latter is surely important, as it shows how we (well, you!) can do a pretty good job at simulating the amplitude of D14Catm change in the glacial when tweaking all the model's knobs, but that we can't subsequently get the deglacial change to the modern value, just as we can't quite get the deglacial change in $CO_2$. I feel this must be significant... I wonder what the authors think.

37. Figure 10 and 11: I would suggest including a narrow plot at the base of each of these showing the offsets between simulated and observed values over time.

---

## Author Response (AR1)

We are grateful to the referees for their constructive feedback and the time they spent reviewing our manuscript. This helped us to improve the presentation, while results and conclusions remain unchanged. Below are our responses (in **bold**) to the referee comments (in *italics*). The list of relevant changes made in the manuscript are in red. Line numbers refer to the originally submitted manuscript.

Referee #1

*Fundamental remarks:*

*1. Weathering vs sedimentation: It is said that when sedimentation is included in the model the atmospheric Δ14C strongly decreases in comparison to an atmosphere-ocean model version only. I believe this is naming the wrong process. My understanding of the model description is, that weathering is the process that brings 14C-free C into the system, so it is carbonate weathering, that is fundamental for the 14C cycle. It is clear that once weathering input (of alkalinity and DIC to the ocean) is considered also sedimentation as sink needs to be implemented (otherwise the carbon cycle would run away with an ocean accumulating alkalinity, and subsequent changes to atmospheric CO2 levels), but sedimentation is not the important process here that changes Δ14C. This might then also lead to a different name of the model configuration now called OCN-SED.*

**Weathering fluxes are an important component of the global carbon cycle and input to the ocean from terrestrial weathering of carbonate and silicate rocks and volcanic emissions is included in our model to balance material loss by burial of particulate organic matter, calcium carbonate, and opal at the sea floor. See also our response to the following comment #2. Potential further changes in weathering fluxes may have been very important for past changes in $\delta^{13}C$ but are largely irrelevant for the present study with its focus on $\Delta^{14}C$. What we are interested in demonstrating here is that a change in the global ocean carbon inventory linked with the weathering/sedimentation balance is a potentially important factor affecting atmospheric $\Delta^{14}C$ levels. Also note that the nomenclature of the model configurations is meant to describe the global carbon reservoirs (model components) under consideration, so, e.g., model configuration OCN-SED includes the ocean model and the sediment model. We will modify the text to make this clearer.**

We have modified the discussion in Sect. 3.1 such that it is now clear that what's important for the long-term response of $\Delta^{14}C$ to perturbations in the ocean carbon cycle is the interaction with the ocean sediments and the imbalance between weathering and sedimentation, e.g., on lines 336-338:

The response of $\Delta^{14}C_{atm}$ to various perturbations depends on the magnitude of the change in the ocean carbon inventory, with a larger change achieved by considering the interaction with the ocean sediments and the imbalance between weathering and sedimentation (see Fig. 5e,f).

We have also added a more detailed description of the model's representation of weathering fluxes in Appendix A (see next point).

*2. No details on weathering are given, but since it is said, that 14C-free C is entered via weathering I have to assume, this implies carbonate weathering. However, it need to be clarified (and maybe corrected?), that in carbonate weathering, 50% of the carbon that enters the ocean as weathering product (bicarbonate ion, $HCO_3^-$, which changes DIC and alkalinity in the ocean) comes from rocks (14C-free), and 50% has its origin in atmospheric CO2 with its atmospheric 14C-signature. For silicate weathering, also bringing $HCO_3^-$ to the ocean, 100% of the carbon has its origin in the atmosphere. Is silicate weathering considered? For details see, for example, Colbourn et al. (2013). Without checking on recent updates, I believe both silicate and carbonate weathering contributed about a similar amount of $HCO_3^-$ input into the ocean. At least in a study some years ago (Hartmann et al., 2009) in present day weathering the CO2 consumption is twice as big in silicate than in carbonate weathering, but since in carbonate weathering 50% of the C has its origin from rocks, both processes should contribute about the same. Since weathering is the relevant process for this paper more details on its implementation in the model should be included. From the Appendix I understood, that weathering rates are constant in time, but please give their numbers, which would be especially of interest to other modellers doing similar things. Also consider in a discussion, that missing temporal changes in weathering rate might be one reason why reconstructed Δ14C (and CO2) is not met with simulations. Having found, that the input of 14C-free carbon to the system is so important for an understanding of Δ14C brings me also to the quesion if 14C-free CO2 outgassing from volcanos is considered, which might have similar effects on 14C. I understand that this has been investigated previously with the Bern3D model (Roth and Joos, 2012), but with focus on 13C. Maybe some more insights from previous simulations are possible here, at least in a discussion. At least please mention the applied CO2 volcanic outgassing rates. Note, that there is a fundamental, analytical derived solution from the steady state assumption on volcanic CO2 input being 50% of the CO2 consumption by silicate weathering, which is of relevance for times longer than 100 kyr (briefly mentioned in Munhoven and François (1996) or in depth discussed on pages 80-81 of Munhoven (1997), http://www.astro.ulg.ac.be/~munhoven/en/PhDIndex.html). For shorter periods such as the last 50 kyr considered here, differences from this numbers are certainly possible, but this relationship gives a rough guideline, and might explain long-term drifts in the C cycle, if not obeyed. Taken together, I have the impression, that no silicate weathering, and also no volcanic outgassing of CO2 is considered here, which would indicate according to this theory no drift in the system, but also the missing of two important processes. If so, I am not saying, these should be implemented in the revision, but it needs to be stated clearly if and how they are (not) included. How does your weathering flux compare to others, e.g. Fig 7 in Brovkin et al. (2012) or Colbourn et al. (2013)?*

**We are afraid that there has been a misunderstanding with regard to the model representation of terrestrial weathering fluxes. Apparently the referee missed the**

description of the sediment model and weathering fluxes given in Appendix A (lines 756-771). We will include the preindustrial steady-state values for weathering rates in Sect. 2.2 and provide additional details on the representation of terrestrial weathering in Appendix A. For convenience, the approach is described here in detail.

The Bern3D model simulates net ocean-sediment exchange, sediment stocks, and burial fluxes from the ocean sediments to the lithosphere of P, Si, Alk, DIC, $DI^{14}C$, and $DI^{13}C$ using a 10-layer ocean sediment model. In steady state, the net loss fluxes from the ocean to the sediments and lithosphere are compensated by corresponding input fluxes to the ocean, termed "weathering fluxes". These input fluxes are thought to represent the fluxes from weathering (dissolution) of carbonate and silicate rocks on land, of phosphorous release by rock weathering, and from $CO_2$ emissions from volcanic activity. The ocean inventories of P, Si, and Alk and atmospheric $CO_2$ and its isotopic signature are prescribed during the atmosphere-ocean-sediment spin-up. Any loss of P, Si, Alk, DIC and $DI^{13}C$ by net fluxes to the sediments is compensated by a corresponding input flux during the spin-up phase. Input fluxes are added uniformly to the coastal surface ocean. At the beginning of transient simulations, the global input fluxes of P, Si, Alk, DIC and $DI^{13}C$ are set equal to the burial fluxes diagnosed at the end of the model spin-up. These input fluxes are jointly denoted as "weathering fluxes". Radiocarbon is transferred from the ocean to the sediments and lithosphere where it decays, but no radiocarbon is added to the ocean from "weathering" as old rocks are radiocarbon free (or "radiocarbon dead").

The preindustrial spin-up results in steady-state values for weathering-derived inputs (and hence steady-state burial rates) of DIC, Alk, P, and Si of 0.46 Gt C per year, 34.37 Tmol $HCO_3^-$ per year, 0.17 Tmol P per year, and 6.67 Tmol Si per year, respectively. These values are within the range of observational estimates (see, e.g., Jeltsch-Thömmes et al., 2019, Table 1). The weathering input of Alk as $HCO_3^-$ is also comparable to the global riverine bicarbonate flux presented in Brovkin et al. (2012), very close to their interglacial estimates (36 to 38 Tmol per year) but lower than their LGM estimates (almost 50 Tmol per year).

These input fluxes may be further attributed to the weathering of organic material, $CaCO_3$, and $CaSiO_3$ on land, and to volcanic $CO_2$ outgassing. The flux of phosphorus (P) is assigned to weathering of organic material, and the related carbon (C) and Alk fluxes are computed by multiplication of the P flux with the Redfield ratio for organic matter stoichiometry (C:P:Alk = 117:1:-17). Similarly, the silicon (Si) flux is assigned to $CaSiO_3$ weathering, and the related Alk flux is computed using Si:Alk = 1:2 based on the simplified $CaSiO_3$ weathering reaction: $2CO_2 + CaSiO_3 + H_2O \rightarrow 2HCO_3^- + Ca^{2+} + SiO_2$ (Colbourn et al., 2013). As Colbourn et al., we sidestep the carbon flux from the atmosphere to the ocean. The remaining Alk flux is attributed to $CaCO_3$ weathering with the stoichiometric ratio C:Alk = 1:2 based on the $CaCO_3$ dissolution reaction: $CO_2 + H_2O + CaCO_3 \rightarrow Ca^{2+} + 2HCO_3^-$. The volcanic outgassing flux is the remaining flux needed to balance the C input flux. The diagnosed fluxes at the end of the spin-up are 0.24 Gt C per year for terrestrial weathering of organic material, 0.13 Gt C per year for terrestrial $CaCO_3$ weathering, 0.09 Gt C per year for volcanic $CO_2$ outgassing, and 6.67 Tmol Si per year for terrestrial $CaSiO_3$ weathering.

For simplicity, we kept weathering fluxes constant during transient simulations, and weathering feedbacks (Jeltsch-Thömmes and Joos, 2020; Colbourne et al., 2013) were not enabled in this study. This does not affect our results and conclusions. First, changes in weathering fluxes have no influence on the ocean radiocarbon inventory as weathering fluxes are "radiocarbon dead". Second, the impact of potential changes in weathering fluxes on atmospheric $CO_2$ and the ocean and sediment carbon inventories, which would influence atmospheric $\Delta^{14}C$, is implicitly considered in our sensitivity experiments where $CO_2$ and carbon inventories are forced to vary. Third, we note that there is a large uncertainty in the dissolution rates of carbonate and silicate rocks on land (terrestrial weathering) over time, and that these weathering reactions represent a very long-term sink of atmospheric $CO_2$. In particular, weathering of silicate rocks on land is occurring too slowly (on a time scale of hundreds of thousands of years) to be important on the time scale relevant for this study (~50,000 years). Furthermore, Roth and Joos (2013) demonstrated that even massive changes in volcanic emissions cause changes in atmospheric $\Delta^{14}C$ that are much smaller than the discrepancies between reconstructed and modelled $\Delta^{14}C$.

Finally, we would like to point out that because DIC and Alk are conservative with respect to changes in state (temperature, salinity, and pressure) during mixing, both are carried as tracers in ocean carbon cycle models like the Bern3D. Together they completely determine the $CO_2$ system in seawater ($H^+$, $pCO_2$, $H_2CO_3^*$, $HCO_3^-$, and $CO_3^{2-}$), using the well-known carbonate chemistry routines. These parameters can be used to compute, e.g., air-sea $CO_2$ and $^{14}CO_2$ fluxes or the saturation state of seawater with respect to $CaCO_3$.

We have added the preindustrial steady-state values for weathering-derived inputs to Sect. 2.2, line 194:

Note the preindustrial spin-up results in steady-state values for weathering-derived inputs of DIC, Alk, P, and Si of 0.46 Gt C per year, 34.37 Tmol $HCO_3^-$ per year, 0.17 Tmol P per year, and 6.67 Tmol Si per year, respectively. These terrestrial weathering rates were chosen to balance the sedimentation rates on the sea floor and are held fixed and constant throughout the simulations.

We have also added more detailed description of the model's representation of weathering fluxes in Appendix A:

Weathering (dissolution) of carbonate and silicate rocks on land, phosphorous release by chemical weathering of rocks, and volcanic outgassing of $CO_2$ are simulated as constant inputs of DIC, Alk (as bicarbonate ion, $HCO_3^-$), phosphate (P), and silicate (Si) to the ocean at rates intended to balance their removal from the ocean by sedimentation on the sea floor. These weathering inputs are added as a constant increment to each surface ocean grid cell along the coastlines. The preindustrial steady state of the model is used to diagnose the weathering rates that are held fixed and constant throughout the simulations. Note that the preindustrial spin-up results in steady-state values for weathering-derived inputs of DIC, Alk, P, and Si of 0.46 Gt C per year, 34.37 Tmol $HCO_3^-$ per year, 0.17 Tmol P per year, and 6.67 Tmol Si per year, respectively. These values are within the range of observational estimates (see, e.g., Jeltsch-Thömmes et al., 2019). Additional details concerning the sediment model are provided in Tschumi et al. (2011), while the appendix of Jeltsch-Thömmes et al. (2019) gives a detailed description of the atmosphere–ocean–sediment spin-up.

*3. Earlier simulation studies have shown, that to get the 14C cycle right, one needs to have the C cycle right as well. Köhler et al. (2006) has shown that previous studies (Beck et al., 2001; Hughen et al., 2004) focusing only on 14C, but showing no simulated CO2, they therefore have very likely some deficits. For atmospheric Δ14C especially the air-sea gas exchange is important, which depends similarily on the gas exchange velocity (kw, which is considered here in sensitivity experiments), but also on the CO2 gradient between atmosphere and surface ocean. This implies that whenever simulated CO2 differs from reconstructions there will also be an offset in simulated Δ14C from data. In a recent simulation effort for IntCal20 (the successor of IntCal13) the marine surface Δ14C has been simulated (Heaton et al., submitted). There, the importance of time-dependent changes in CO2 has been as important for the simulated surface ocean Δ14C as that of climate change (temperature change, ocean circulation change etc), which via gas exchange would also feedback to atmosheric Δ14C. This is unpublished so far, but since it is submitted and will probably be available in due time I nevertheless mention it here.*

**We thank the referee for bringing this very interesting-sounding work to our attention. We look forward to reading it once it becomes publicly available.**

**It is well known that gross isotopic air-sea fluxes scale with atmospheric $CO_2$ and the $^{14}C/C$ ratio. In 6 of our 8 model carbon cycle scenarios, parameter values were selected to reproduce low glacial atmospheric $CO_2$ concentrations. Note also that atmospheric $CO_2$ was prescribed in the model runs where the $^{14}C$ production rate is deconvolved from the model results (i.e., Sect. 3.4). Thus, temporal changes in atmospheric $CO_2$ are taken into account by our model simulations.**

We have added the following text after the second paragraph in Sect. 2.4:

Variations in atmospheric $CO_2$ govern how fast $\Delta^{14}C$ signatures are passed between the atmosphere and ocean. Gross fluxes of $^{14}C$ between the atmosphere and ocean, and vice versa, scale with atmospheric $pCO_2$ and its $^{14}C/C$ ratio. It is therefore important to reproduce low glacial atmospheric $CO_2$ concentrations in at least some of the model scenarios, thereby capturing the influence of temporal changes in $CO_2$ on the air-sea exchange of $^{14}C$.

*4. The coauthor Florian Adolphi is also coauthor of the nows ubmitted IntCal20 effort (updating the atmospheric Δ14C record), (Reimer et al., submitted) and should therefore be aware of the large changes which occur between IntCal13 and IntCal20, namely the amplitude of the Δ14C maxima around 40 kyr increases in IntCal20 towards the Hulu Cave numbers. Maybe this should be briefly discussed in an outlook.*

**The backbone of the new IntCal20 calibration curve is the Hulu Cave $\Delta^{14}$C dataset from Cheng et al. (2018) that we use in this work, since IntCal20 is not yet published. Essentially all datasets underlying the IntCal20 curve are tied to the Hulu Cave record, either via time scales (Lake Suigetsu plant macrofossil data) or marine reservoir corrections (marine records). Hence, IntCal20 looks more or less like the Hulu record. Since we are not discussing the fine-scale structure of the record but rather the large-scale changes in $\Delta^{14}$C, using IntCal20 would not significantly impact our conclusions.**

We have added the following text to the third paragraph in Sect. 2.5:

Note that although the forthcoming IntCal20 calibration curve (Reimer et al., in press) will be the new standard atmospheric radiocarbon record for the last 55,000 years, essentially all data underlying IntCal20 before 13.9 kyr BP are tied to the Hulu Cave dataset, either via time scales (Lake Suigetsu plant macrofossil data) or marine reservoir corrections (marine records). Hence, the IntCal20 and Hulu Cave $\Delta^{14}C_{atm}$ records are very similar and using IntCal20 would not impact our conclusions.

*Minor issues in chronological order:*

*1. The decay constant of 14C used here is based on a halflife of 14C of 5700 yr (here) but of 5730 yr in Intcal13 (and IntCal20) which produced the atmospheric Δ14C record. If you consider the decay of 14C over 40 kyr (the time of the maximum in atm Δ14C) with either 5700 vs 5730 yr you get a 2.5% smaller number when based on 5700 yr, although the halflife time differed only by 0.5%. This difference is small when compared to the difference of IntCal13 and the Hule Cave data, but should nevertheless be mentioned.*

**As mentioned in Fig. 1 caption, reconstructed $\Delta^{14}$C values taken from the IntCal13 calibration curve and the Hulu Cave dataset were adjusted to the presently accepted value of the radiocarbon half-life (5700 years), allowing comparison with our modelled $\Delta^{14}$C values.**

*2. line 173: Please be specific, how 14C is fractionation corrected.*

**Radiocarbon measurements are generally reported as $\Delta^{14}$C which includes a correction for fractionation effects. The measured $\delta^{13}$C value is used to remove the effects of isotopic fractionation. As indicated in Sect. 2.2, this model study simulates a $^{14}$C concentration that is "fractionation corrected". What we mean by this is as follows. To model $^{14}$C, the Bern3D**

neglects effects due to fractionation during gas exchange and photosynthesis, which means that model results for $\Delta^{14}C$ are directly comparable to measurements reported as $\Delta^{14}C$. If we were interested in dealing with absolute values of the $^{14}C$ concentration over time, then a correction is needed to account for fractionation effects, using the following equation (see Orr et al., 2017, Eq. A3 and associated discussion in the Appendix):

$$^{14}C = \left( ^{14}C_{model} \Big/ \left[ 1 - 2 \left( \frac{\delta^{13}C + 25}{1000} \right) \right] \right) {}^{14}r_{std} \tag{1}$$

For $^{13}C$, the Bern3D includes fractionation effects during gas exchange and photosynthesis. Eq. 1 together with modelled $\delta^{13}C$ values could be used to compute corrections for atmospheric and oceanic $^{14}C$, were we interested in looking at absolute values of the $^{14}C$ concentration.

The text near line 173 is modified to read:

In this model study, $\Delta^{14}C$ is treated as a diagnostic variable using the two-tracer approach of OCMIP-2. Rather than treating the $^{14}C$/C ratio as a single tracer, fractionation-corrected $^{14}C$ is carried independently from the carbon tracer.

*3. line 180-181: Weathering is prescribed as constant input of DIC, but no 14C. See fundamental comment above, but no matter where the C of weathering comes from the input in the ocean should be a $HCO-3$ , changing both DIC and alkalinity in the ocean. I hope this is only a too simplified description here, but has implemented correctly in the model. Please revise.*

This comment has already been addressed in our response to comment #2 of the referee's "fundamental remarks".

*4. line 184: For the preindustrial spin-up CO2 is fixed to 278.05 ppm. Why this number, would not 278 ppm do the same job? Is this OCMIP protocol? Also: What would be the internally calculated preindustrial CO2? By prescribing CO2 concentration during spin-up C is added or extracted from the simulated system, which might be a potential source of bias. How long is the spin-up time?*

This is irrelevant. The reason for this atmospheric $CO_2$ value is that it is the nominal value for year 1750 in one of our in-house $CO_2$ data compilations, but yes, holding atmospheric $CO_2$ constant at 278 ppm would do the same job. In the preindustrial spin-up simulation, the model is run to equilibrium over a ~50,000-year integration, as mentioned in line 186. During this spin-up, atmospheric $CO_2$ is held constant at 278.05 ppm and $\Delta^{14}C$ at 0 permil. These are the values that the atmospheric carbon and $^{14}C$ tracers see. The ocean carbon and radiocarbon inventories change in response to this forcing. After spin-up the ocean is in equilibrium with the atmosphere.

*5. line 229: Consider citing the underlying ice core paper, from which the greenhouse gases splines provided by Köhler et al (2017) have been calculated.*

**Agreed. We will cite Enting (1987) for the spline smoothing method.**

We have now cited Enting (1987):

(data compilation of Köhler et al., 2017, as splined using the spline smoothing method of Enting, 1987)

*6. line 232: Global benthic δ18O is not only a global ice volume proxy, but has also a considerable contribution from deep ocean temperature, see papers of the van de Wal group from Utrecht Universitiy on the deconvolution (e.g. Bintanja and van de Wal, 2008). Taken this knowledge into consideration, would this change your approach how sea level is changing? You might also discuss how different sea level reconstructions vary, e.g. see three different sea level reconstructions in Fig 1f of Hasenclever et al. (2017), and what this uncertainty in sea level might introduce into your approach.*

**No, this would not significantly change our approach or our results. As explained in the manuscript, we use the global benthic $\delta^{18}O$ stack to scale the ice sheet size for periods where no reconstructions are available. The tie points of this scaling, i.e., the LGM and preindustrial ice sheet reconstructions, remain unaffected by the scaling method, i.e. whether the scaling is done with $\delta^{18}O$ corrected for deep ocean temperature or not. Ice sheet size is important for albedo, salinity and latent heat fluxes, but has little influence on modelled atmospheric $\Delta^{14}C$ (e.g., the small difference between scenarios MOD and PAL in Fig. 8).**

*7. lines 223-236: Model description says that greenhouse gas radiative forcing has been taken from data, thus I assume that CO2 seen by the carbon cycle is never prescribed, but always model-internally calculated. Please state this explicitly (or the correct version of this sentence, if this was not the case). However, you might also consider one scenario in which CO2 is prescribed for the C cycle from data (similar as in Butzin et al., 2017, in which atmospheric CO2 and Δ14C has been prescribed by data), since this would bring your simulated C cycle as close to observations as probably possible, which might further reduce the bias in 14C (see also fundamental remarks above).*

**Although the radiative forcing for $CO_2$ is prescribed, the atmospheric $CO_2$ concentration is allowed to evolve freely, except in the simulations described in Sect. 2.5. We will clarify this point in Sect. 2.4.**

**As discussed in lines 302-315, our approach to estimating the $^{14}C$ production rate over the last 50 kyr relies on model simulations forced by reconstructed changes in atmospheric $\Delta^{14}C$**

and $CO_2$ as well as 7 different carbon cycle scenarios. None of the model runs are able to reproduce the reconstructed variations in [14]C production during the last glacial, especially between 32 and 22 kyr BP (see Sect. 3.4 and Fig. 10 and 11). Thus, the discrepancy between reconstructions and model results remains even when prescribing atmospheric $CO_2$.

We have clarified in Sect. 2.4 that atmospheric $CO_2$ is allowed to evolve freely:

Note that, although the radiative forcing for $CO_2$ is prescribed, the atmospheric $CO_2$ concentration is allowed to evolve freely, except in the simulations described in Sect. 2.5.

*8. Obtained surface reservoir ages (Fig 8c) might be compared with data and other models, e.g. see Butzin et al. (2017); Skinner et al. (2019). Benthic-atmospheric offsets (Fig 8d) might be compared for the LGM with the data compilation of Skinner et al. (2017). Note, surface reservoir ages might vary a lot as function of latitude, so this Fig 8c needs more information on averaging; even better: might be revised and thus restricted to sea ice-free areas only.*

We agree comparison with measurement- and model-based estimates of radiocarbon reservoir age offsets from, e.g., Skinner et al. (2017) and Butzin et al. (2017), is a missed opportunity. It was a sacrifice made to reduce the length of an already very lengthy manuscript. Nonetheless, some intriguing points can be made by such a comparison, so we will incorporate it into Sect. 3.3 and Fig. 8.

Comparison of our LGM B-Atm age offset estimates from runs CIRC, VENT, and VENTx (range of 3682 to 3962 [14]C years) with the compiled LGM marine radiocarbon data of Skinner et al. (2017) demonstrate that the carbon cycle scenarios are extreme, although it should be noted that they consider a wider depth range (~500 to 5000 m) of the ocean than we do. Skinner et al. (2017) predict a global average LGM B-Atm value of ~2048 [14]C years, an increase of ~689 [14]C years relative to preindustrial. Turning our comparison to surface reservoir ages, we note that our global average LGM surface reservoir age of ~1132 [14]C years from runs VENT and VENTx is comparable to the ~1241 [14]C years obtained by Skinner et al. (2017) for the LGM. The model-based estimates of surface reservoir age from Butzin et al. (2017) indicate a much lower LGM value of ~780 [14]C years, and values ranging from 540 to 1250 [14]C years between 50 and 25 kyr BP. Note that these estimates are based on model-simulated values between 50°N and 50°S. If the polar regions are included in the calculation (see Fig. 8c), their surface reservoir age estimates become comparable to our glacial values (range of 911 to 1354 [14]C years), and between about 34 and 22 kyr BP can exceed them, including even those from model runs VENT and VENTx, unless atmospheric $\Delta$[14]C and $CO_2$ are prescribed (dashed colored lines in Fig. 8c). Interestingly, this is also roughly the time period where our deconvolutions of the IntCal13 and Hulu Cave $\Delta$[14]C records give production rate estimates that are about 17.5 percent higher than the reconstructions, which indicates at the very least this is an important piece of the puzzle of the glacial-interglacial $\Delta^{14}C$ problem, given that the effect of upper ocean stratification and/or sea ice on air-sea gas exchange is particularly important for surface reservoir ages.

Comparison of our surface reservoir ages with estimates from Skinner et al. (2019) will have to await a future study. A clear picture of the spatiotemporal evolution of the global average surface reservoir age has yet to emerge, but the regionally distinct patterns as demonstrated by Skinner et al. (2019) have important implications for the calibration of marine radiocarbon samples. These results need to be scrutinized more carefully and investigated in more detail with models, and with experiments specifically designed with this question in mind.

[Figure]

Fig. 8. Modelled records of atmospheric (a) Δ¹⁴C and (b) CO₂, compared with their reconstructed histories (black and dark blue lines). Also shown are modelled records of the global average (c) surface reservoir age and (d) B-Atm ¹⁴C age offset, compared with a recent compilation of LGM marine radiocarbon data (dark blue squares) by Skinner et al.

(2017) and model-based surface reservoir age estimates between 50°N and 50°S (solid black line) and across all latitudes (dashed black line) from Butzin et al. (2017), as well as (e) ideal age and (f) apparent oxygen utilization (AOU). Colored lines show the results of model runs using the mean paleointensity-based $^{14}$C production rate and the eight different carbon cycle scenarios described in Sect. 2.4 and Table 1. The gray envelope in (a) shows the uncertainty (2σ) from all production rate reconstructions and carbon cycle scenarios, providing a bounded estimate of Δ$^{14}$C change. The dashed colored lines in (c) show the surface reservoir age results from VENT and VENTx where atmospheric Δ$^{14}$C and $CO_2$ are prescribed. Radiocarbon ventilation ages are expressed here as radiocarbon reservoir age offsets following Soulet et al. (2016) which are used extensively by the radiocarbon dating community.

We have added the observed and modelled records of Skinner et al. (2017) and Butzin et al. (2017), respectively, to Fig. 8d,c as well as added a paragraph to Sect. 3.3 that discusses the model-data comparison of B-Atm and surface R-age:

Driven by a reduction in ocean circulation, model run CIRC predicts a substantial increase in B-Atm during the last glacial, which is defined here as 40 to 18 kyr BP to avoid biasing global mean estimates toward Laschamp values. The global average glacial B-Atm predicted by CIRC is ~3225 $^{14}$C years, representing an increase in B-Atm of ~1599 $^{14}$C years relative to the preindustrial value of ~1626 $^{14}$C years. Model run VENT predicts a slightly larger increase in glacial B-Atm due to the inhibition of air-sea gas exchange. The "oldest" glacial waters are found in model run VENTx where air-sea gas exchange is severely restricted, yielding an increase in B-Atm of ~1912 $^{14}$C years (glacial B-Atm ~3538 $^{14}$C years). The glacial B-Atm values given by runs CIRC, VENT, and VENTx, as well as the ~717 year increase in ideal age during the last glacial relative to preindustrial, suggest that the glacial deep ocean was about two times older than its preindustrial counterpart. Comparison of our LGM B-Atm estimates (range of 3682 to 3962 $^{14}$C years) with the compiled LGM marine radiocarbon data of Skinner et al. (2017) demonstrate that the carbon cycle scenarios are extreme, although it should be noted that Skinner et al. consider a wider depth range (~500 to 5000 m) of the ocean than we do. Skinner et al. (2017) predict a global average LGM B-Atm value of ~2048 $^{14}$C years, an increase of ~689 $^{14}$C years relative to preindustrial. Turning our comparison to surface reservoir ages, we note that our global average LGM surface R-age of ~1132 $^{14}$C years from runs VENT and VENTx is comparable to the ~1241 $^{14}$C years obtained by Skinner et al. (2017) for the LGM. The model-based estimates of surface R-age from Butzin et al. (2017) indicate a much lower LGM value of ~780 $^{14}$C years, and values ranging from 540 to 1250 $^{14}$C years between 50 and 25 kyr BP. Note that these estimates are based on model-simulated values between 50°N and 50°S. If the polar regions are included in the calculation (see Fig. 8c), their surface R-age estimates become comparable to our glacial values (range of 911 to 1354 $^{14}$C years), and between about 34 and 22 kyr BP can exceed them, including even those from model runs VENT and VENTx, unless Δ$^{14}$C$_{atm}$ and $CO_2$ are prescribed (dashed colored lines in Fig. 8c) as in the simulation by Butzin et al. (2017).

*9. Please state somewhere the absolute (PI) values of those parameters which are changed in your sensitivity experiments, maybe in Table 1?*

**We would like to direct the referee, and the reader, to the appendix of Roth et al. (2014) for the Bern3D model parameter set.**

The following text has been added to the Table 1 caption:

See Roth et al. (2014) for the Bern3D model parameter set.

*10. Table 2: 14C production rates is given in relative units, relative to what? Probably preindustrial state. I also do not remember if the 14C production rate in absolute numbers is once given in the manuscript for preindustrial state, please insert somewhere.*

**Our model-based records of the global production rate of $^{14}$C are in units relative to the preindustrial value, as mentioned in lines 416-417. We will include the preindustrial steady-state absolute value of 443.9 mol $^{14}$C per year (1.66 atoms cm$^{-2}$ s$^{-1}$) in Sect. 2.2.**

We have added the preindustrial steady-state absolute value of atmospheric $^{14}$C production of 443.9 mol $^{14}$C per year (or 1.66 atoms cm$^{-2}$ s$^{-1}$) to Sect. 2.2.

*11. Fig 4 captions does not need a description of the different colors of the lines, since a legend is given in the figures themselves.*

**Here we reference the colored lines in order to remind the reader that their labels refer to the model configurations representing different combinations of global carbon reservoirs, which is important when comparing the response of $\Delta^{14}$C to the step changes.**

Referee #2

*1. Line 11: I feel that the term 'mystery interval' has become current without having a particularly clear meaning; it seems to be used to refer to a chronozone, for which there already is a name (Heinrich Stadial 1, etc...). Furthermore, the 'mysterious' part of the interval seems to be perceived differently by different people; is it the atmospheric radiocarbon decline, the proposed lack of marine radiocarbon activity increase, the entire 'mystery' of deglaciation? Not everyone shares the same notions regarding such 'mysteries', particularly regarding the marine radiocarbon inventory change, against which the term 'mystery interval' seems to have been directed. I would like to stick my neck out and suggest that this term has served its purpose in stimulating interest in a topic, and no longer serves a purpose for clear communication of a specific idea. I would therefore propose that the authors refer to other more clearly established chronozone designations, or even dates if these are trusted sufficiently.*

We agree with the referee that using the term "mystery interval" to refer to the sharp drop in $\Delta^{14}$C across Heinrich Stadial 1 ~17.5 to 14.5 kyr BP serves no purpose other than to stimulate interest. We will update the manuscript to be more precise, such that "mystery interval" is replaced by Heinrich Stadial 1.

We have replaced the term "mystery interval" with "Heinrich Stadial 1" throughout the revised manuscript.

*2. Line 18: I think the word "more" can be dispensed with, here and elsewhere. One wonders: more than what?*

Models allow us to investigate specific phenomena in more idealized settings compared to the "real world". However, we agree that, in this context, referring to such settings as "more idealized" rather than simply "idealized" is not very useful. The manuscript will be updated accordingly.

We have replaced the phrase "more idealized" with "idealized" throughout the revised manuscript.

*3. Line 40: here and throughout the manuscript I was not sure whether "millennial-scale" was a helpful designation, as it made me think of variability associated with Dansgaard-Oeschger events. Perhaps the term, or another such as "short term" etc..., can be defined clearly when first used?*

The primary focus of this work is on the specific mechanisms responsible for variations in atmospheric $\Delta^{14}$C on millennial time scales (i.e., time scale of thousands of years). We do not attempt to resolve more abrupt climate perturbations such as Dansgaard-Oeschger warming events, which is noted in lines 637-641 of the original manuscript. To avoid confusion, we will add a note of caution in Sect. 2.4 when we introduce the carbon cycle scenarios considered in the model runs.

We have added a note of caution when introducing the model carbon cycle scenarios in Sect. 2.4:

A note of caution. Because millennial-scale $\Delta^{14}C_{atm}$ variations during the last glacial are what we are interested in, we do not attempt to reproduce abrupt climate perturbations such as Dansgaard-Oeschger warming events in the model runs.

*4. Line 60: same thoughts as above regarding the term "mystery interval"; if it is coincident with HS1, then we should use that term instead I think. At the time of the Broecker and Barker (2007) study there was proposed to be a lack of evidence for a radiocarbon depleted ocean interior at the LGM, and a subsequent increase in its radiocarbon activity; however, this is arguably no longer the case.*

**This comment has already been addressed in our response to comment #1.**

*5. Line 67: probably best to be more specific, e.g. "...used only high accumulation sites, and square barrel gravity cores with minimal sediment disturbance.."*

**We agree with the referee that it would be valuable for the reader if we elaborated on the coring and sampling methods that minimize the influence of drilling disturbance. This will be done in a revised manuscript.**

We have included additional details on the coring and sampling methods that minimize the influence of drilling disturbance:

Paleointensity-based reconstructions are sensitive to coring disturbances of poorly consolidated sediments. The last 50 kyr are represented by the relatively slushy uppermost few meters of recovered marine sediment cores (Channell et al., 2018). Channell et al. (2018) preferentially selected cores recovered using conventional piston and square barrel gravity coring methods, and from sites with high mean (> 15 cm kyr$^{-1}$) sedimentation rates, so as to minimize the influence of drilling disturbance, and reached very different production rates than, e.g., Laj et al. (2000).

*6. Line 72, last sentence of the paragraph: I don't mean to suggest that ther is anyh I gotta incorrect about this sentence, but I found this to be an odd way of phrasing things. To me there is one question, "why was atmospheric radiocarbon activity so high during the last glacial (including well after the Laschamp excursion)", which entails a subsidiary question, "how much did production changes contribute to this elevated atmospheric radiocarbon activity".*

**We agree with the referee that it is unnecessary to make a distinction between the contribution of production changes to high glacial $\Delta^{14}$C levels and their contribution to the deglacial $\Delta^{14}$C decline. Our goal was to remind the reader that only if estimates of past changes in $^{14}$C production are robust can one improve assessments of the relative importance of the two fundamental mechanisms responsible for glacial-interglacial $\Delta^{14}$C changes (i.e., production and carbon cycle changes).**

We have reframed what the interpretation problem caused by uncertainties in past estimates of $^{14}$C production is:

The large uncertainties associated with the reconstruction of past changes in $^{14}$C production hamper our ability to predict reliably the extent to which production changes contributed to high glacial $\Delta^{14}$C$_{atm}$ levels. Only if estimates of past changes in $^{14}$C production are robust can one improve assessments of the relative importance of the two fundamental mechanisms responsible for glacial-interglacial $\Delta^{14}C$ changes: (1) production changes and (2) carbon cycle changes.

*7. Line 95: I would say that the time required for ocean ventilation is not "up to", but rather "over" 1000yrs. Perhaps Primeau (2005) can be referenced for this. 8. Line 100: multi-millennial timescales?*

**While the ventilation time scale for the deep ocean is typically of order 1000 years, we note that the deep ocean ventilation time scale can exceed 1000 years, as demonstrated by the modelling study of Primeau (2005). This time scale depends on which Ocean General Circulation Model and tracer was used to predict the time scale of the penetration of water from the surface into the ocean interior.**

We have qualified that the deep ocean ventilation time scale can exceed 1000 years per Primeau (2005).

*9. Line 106: Andrey Ganopolski would disagree (see Ganopolski et al., CP, 2017). Perhaps this statement should be modified to say that it is currently not possible to do so without the use of any parameterisations of key processes, or something more specific?*

**While Ganopolski & Brovkin (2017) reproduce the overall trends and more general features of glacial–interglacial variability of climate, ice sheets, and atmospheric $CO_2$ concentration using only orbital forcing to drive the CLIMBER-2 model, the finer-scale temporal dynamics of the simulated $CO_2$ evolution do not match the reconstructions. In particular, the model fails to simulate the correct timing of the deglacial $CO_2$ rise. In addition, the model underestimates the magnitude of the deglacial decline in atmospheric $\Delta^{14}C$. Therefore, we think it is reasonable to conclude that models cannot yet reproduce climate and atmospheric $CO_2$ variations on the basis of orbital forcing alone.**

*10. Line 113: this sentence seems to suggest that the main proposals for explaining glacial-interglacial CO2 involve exchanges with the solid earth, but this is not really true. Arguably, as has been sketched out many times before, including in a recent review (Galbraith 2020), the "ingredients" for glacial-interglacial CO2 change are well accounted for, it is their 'calibration' and organisation within an orbital pacing framework that remains elusive.*

**We agree with the referee. The text will be modified to highlight the role of ocean-based physical and biological mechanisms in explaining the glacial-interglacial variations in atmospheric $CO_2$, and to clarify that what is missing is a single framework in which these mechanisms are linked to each other in a predictable manner under the influence of orbital forcing.**

We have rephrased the discussion of the glacial-interglacial $CO_2$ problem:

A wide variety of mechanisms, both physical and biological, centered on or connected with the ocean, as well as exchange processes with the land biosphere, marine sediments, coral reefs, and the lithosphere, are thought to play a role in explaining the glacial-interglacial variations in atmospheric $CO_2$ (Archer et al., 2000; Fischer et al., 2010; Wallmann et al., 2016; Galbraith and Skinner, 2020), but how they interacted over time under the influence of orbital forcing remains elusive. We appear to still be missing a single framework in which these mechanisms are linked to each other in a predictable manner.

*11. Line 118: in idealised settings..*

**This comment has already been addressed in our response to comment #2.**

*12. Line 122: here and throughout the manuscript it would be best to suffix D14Catm, so that we know what reservoir is referred to.*

**We agree this notation would be useful for the reader and will apply it in a revised manuscript.**

We have replaced the term "$\Delta^{14}C$" with "$\Delta^{14}C_{atm}$", when appropriate, throughout the revised manuscript.

*13. Line 125: is it not more accurate to state that the production rate is inferred from an atmospheric radiocarbon budget, combined with a range of hypothetical radiocarbon and carbon cycle scenarios?*

**We agree with the referee it would be more precise to state that our model-based 50,000-year reconstruction of the $^{14}C$ production rate is based on an atmospheric radiocarbon budget that is put together by forcing the Bern3D carbon cycle model with reconstructed changes in atmospheric $\Delta^{14}C$ and $CO_2$ as well as carbon cycle scenarios.**

We have qualified that our new reconstruction of the $^{14}C$ production rate relies upon carbon cycle model simulations, i.e.:

"…a new 50,000-year record of the $^{14}C$ production rate, as inferred by deconvolving the reconstructed histories of $\Delta^{14}C_{atm}$ and $CO_2$ with a prognostic carbon cycle model and considering the uncertainties associated with the glacial-interglacial ocean carbon cycle."

*14. Line 159: air-sea equilibration times are very different, which is potentially important...*

**The air-sea equilibration time scale for $\Delta^{14}C$ by gas exchange depends in part on the gas transfer velocity, which is investigated in the sensitivity experiments presented in Sect. 3.1.3. These simulations demonstrate a modest response of $\Delta^{14}C$ of approximately 4-8‰ to a 100% reduction of the gas transfer velocity at the north (> 60°N) and south (> 48°S) poles.**

We have added the following text to line 165:

Air-sea gas exchange is parameterized using a modified version of the standard gas transfer formulation of OCMIP-2, with exchange rates that vary across time and space (see Appendix A for more details).

*15. Line 169: perhaps Stuiver et al. 1978 should be referenced.*

**We cited Stuiver and Polach (1977) in lines 35-36 of the original manuscript, but we see no reason why we should not cite them again in Sect. 2.2 as suggested.**

We have included a citation for Stuiver and Polach (1977) in Sect. 2.2.

*16. Line 170: would it be clearer to state that DI14C is simulated, separately from DIC?*

**We agree with the referee. We will modify Sect. 2.1 and 2.2 to clarify that $CO_2$, $^{14}CO_2$, DIC, and $DI^4C$ are all carried by the model, and are used to diagnose atmospheric and oceanic $\Delta^{14}C$.**

We have revised the description in Sect. 2.2 so that it is more obvious that the $^{14}C$ and carbon tracers are carried independently:

In this model study, $\Delta^{14}C$ is treated as a diagnostic variable using the two-tracer approach of OCMIP-2. Rather than treating the $^{14}C/C$ ratio as a single tracer, fractionation-corrected $^{14}C$ is carried independently from the carbon tracer. The modelled $^{14}C$ concentration is normalized by the standard ratio of the preindustrial atmosphere ($^{14}r_{std}$ = 1.170 x $10^{-12}$; Orr et al., 2017) in order to minimize the numerical error of carrying very small numbers. For comparison to observations, $\Delta^{14}C$ is calculated from the normalized and fractionation-corrected modelled $^{14}C$ concentration as follows:

$$\Delta^{14}C = 1000\left(^{14}r' - 1\right) \tag{1}$$

where $^{14}r'$ is the ratio of $^{14}C/C$ in either atmospheric $CO_2$ or oceanic DIC divided by $^{14}r_{std}$, depending on the reservoir being considered. The approach taken to simulate atmospheric $^{14}CO_2$ is analogous to the approach used for $CO_2$, except that the equation includes the terms due to atmospheric production and radioactive decay. For simulations where the sediment model is active, the oceanic DIC tracer sees a constant input from terrestrial weathering, whereas there is no weathering input of $DI^{14}C$ to the ocean (see Appendix A for more details).

*17. Line 189: I wonder if this is not a major part of the whole problem with simulating atmospheric radiocarbon in the past? If the modern (pre-industrial) state is in fact far from*

*equilibrium then this would mean that production rates are all miscalibrated. Why not explore the possibility that production rates are higher than required for equilibrium, e.g. due to ongoing equilibration of sedimentation following the deglaciation and early Holocene? It seems to me that the very conclusions of this study require that this be explored as a possibility. More specifically, and perhaps I am not getting this right.. we might expect that, following the expansion of the terrestrial biosphere during the Holocene (and the removal of carbon from the atmosphere-ocean system, causing a slow reduction of 'young' carbonate sediment output from the ocean), the radiocarbon inventory of the ocean and atmosphere should be on a slow disequilibrium downward trend, so that a higher radiocarbon production would be needed to get today's radiocarbon activity as an equilibrium state. Is that correct? Or is it the opposite? In any event, one has a sneaking suspicion that this sort of thing might be important here.*

This is a very interesting point, but our results suggest that such a disequilibrium effect is of relatively minor importance. Firstly, disequilibrium effects are fully accounted for in the model simulations where atmospheric $CO_2$ and $\Delta^{14}C$ are prescribed (see Sect. 2.5 and 3.4), given that the transient time evolution is modelled. Here, there is a major mismatch between the reconstructed production rates and those diagnosed from our simulations (see Fig. 10 and 11). Furthermore, as shown in Fig. 8a, the mismatch between reconstructed and modelled atmospheric $\Delta^{14}C$ at the preindustrial is on the order of a few percent and scaling the production records accordingly would not remove the mismatch in atmospheric $\Delta^{14}C$ during the last glacial period. We refrain from such a posteriori scaling as the mismatch in atmospheric $\Delta^{14}C$ at the preindustrial is likely related to the mismatch between observed and modelled atmospheric $CO_2$ (see Fig. 8b). What we will say here is that an incorrect preindustrial $^{14}C$ production rate would introduce a potential bias, leading to systematic underestimates (or overestimates) of atmospheric $\Delta^{14}C$ values over time. However, increasing (or decreasing) the base level of our production rate would not fix the glacial $\Delta^{14}C$ problem, i.e., the persistent elevation of $\Delta^{14}C$ after ~33 kyr BP. This can also be understood by Fig. 9.

The uncertainty in the preindustrial production rate is on the order of 15% due to the uncertainties in the preindustrial ocean radiocarbon inventory (see Roth and Joos, 2013, Sect. 3.2). This potential systematic bias was not considered by our model simulations, but it would not affect our analysis as we consider normalized production rate changes (see, e.g., Fig. 7, 10, and 11).

Finally, the preindustrial $^{14}C$ production rate Q of 1.66 atoms $cm^{-2}$ $s^{-1}$ that is diagnosed at the end of the preindustrial spin-up agrees reasonably well with independent estimates from production rate models, e.g., Masarik and Beer (1999, 2009) (Q = 2.05 atoms $cm^{-2}$ $s^{-1}$ for a solar modulation potential of 550 MeV) and Kovaltsov et al. (2012) (Q = 1.88 atoms $cm^{-2}$ $s^{-1}$ for the period 1750 to 1900 AD), and from Roth and Joos (2013) using an earlier Bern3D-LPX model version (Q = 1.75 atoms $cm^{-2}$ $s^{-1}$ for the period 1750 to 1900 AD).

We have added the following text after the second paragraph in the summary and conclusions section (line 715):

Atmospheric $\Delta^{14}C$ that is modelled at any point in time reflects $^{14}C$ production at that point, as well as the legacy of past production and carbon cycle changes. The question arises as to whether our conclusions are affected by unaccounted legacy effects, e.g., linked to the preindustrial spin-up simulation or model-diagnosed production rates. Transient simulations forced by reconstructed changes in $^{14}C$ production (Sect. 3.2 and 3.3) are initialized at 70 kyr BP, but their interpretation is restricted to the last 50,000 years of the integration to minimize legacy effects from model spin-up. Available reconstructions of the $^{14}C$ production rate in relative units (Sect. 2.5) are applied as a scale factor to the preindustrial steady-state absolute value, which is diagnosed by running the Bern3D model to equilibrium under preindustrial boundary conditions. This approach represents an approximation and equilibrium conditions do not fully apply. Indeed, there is a mismatch between reconstructed and modelled $\Delta^{14}C_{atm}$ at the preindustrial (see Fig. 8a). This mismatch is on the order of a few percent or less and adjusting the base level of production accordingly would not remove the large mismatch between reconstructed and modelled $\Delta^{14}C_{atm}$ during the last glacial. In addition, the uncertainty in the absolute value of the preindustrial production rate is on the order of 15%, primarily due to the uncertainties in the preindustrial ocean radiocarbon inventory (see Roth and Joos, 2013, Sect. 3.2). This potential systematic bias, however, does not affect our conclusions as we consider normalized production rate changes (see Fig. 7, 10, and 11).

We have also added the following text to the third paragraph in the summary and conclusions section:

Here, non-equilibrium effects are fully accounted for by transient simulations where $\Delta^{14}C_{atm}$ and $CO_2$ are prescribed (Sect. 3.4) following their reconstructed histories. Yet, these simulations indicate that the discrepancy between measurement- and model-based estimates of the $^{14}C$ production rate remains for the last glacial (Fig. 10b). This would suggest that unaccounted legacy effects do not significantly affect our conclusions.

*18. Line 220: "...levels, given available 14C production scenarios."*

**We agree with the referee it would be more precise to state that what we are interested in investigating is the extent to which changes in the ocean carbon cycle could explain high glacial $\Delta^{14}C$ levels, given available reconstructions of past changes in $^{14}C$ production.**

We have qualified this point in Sect. 2.4:

The goal is to investigate the extent to which changes in the ocean carbon cycle could explain high glacial $\Delta^{14}C_{atm}$ levels, given available reconstructions of past changes in $^{14}C$ production.

*19. Line 244: Why was benthic d18O chosen? It is a smooth, slow function that lags behind most of the climatic processes that were important for the carbon cycle. Although it might seem circular, I don't think it is any more ad hoc to scale these parameters to atmospheric CO2 instead.. having rapid jumps in HS1 and the YD, and a faster change than benthic d18O, might help with getting the deglacial CO2 change 'right' (for parameterised reasons).*

**We agree with the referee that a different scaling approach would be preferential when addressing the last glacial termination as benthic $\delta^{18}O$ lags the rise in atmospheric $CO_2$ and temperature as shown by Shackelton (2000). However, as our primary focus is on the last glacial period, a different scaling, e.g., by $CO_2$, would not change our conclusions.**

*20. Line 254: the cited study is based entirely on the 'plateau tuning' approach, which may be questioned. Perhaps best to also cite Skinner et al. (2017) who showed that the LGM ocean was 'older' pretty conclusively with a range of other data.*

**We agree with the referee that Skinner et al. (2017) would be a good study to cite here.**

We have added a citation for Skinner et al. (2017).

*21. Line 283: It seems crucially important to me that the 10Be and 36Cl flux records from the ice cores are NOT consistent with the final age scale that they are all placed on. As far as I can tell from Adolphi et al. (2018), the ice core data were converted to fluxes based on each ice core's individual age scale, and then they were all placed on the GICC05 age-scale, whereas Channell et al. (2018) argued that this age scale implies very different fluxes. Surely the ice core cosmogenic nuclide data ALL need to be placed on the same age scale and THEN the fluxes should be calculated and 'stacked'. I think this is a really crucial thing, and I am really confused as to why the specialists working with these isotope records take a different approach that surely produces incorrect fluxes. A basic test I would propose is: are the individual ice core flux records consistent with the accumulation rates that are implied for each ice core by the GICC05 age scale? If not, they need to be corrected, surely. I suspect this will only make matters worse for reconciling everything, but it is still important to consider carefully.*

**We are afraid that there has been a misunderstanding. The referee is correct that all time scale revisions impact ice-core accumulation rates and hence fluxes. We want to point out, however, that, as described in Adolphi et al. (2018) (Sect. 3.1, first paragraph), all ice cores were first placed on the same time scale (GICC05) before fluxes were calculated. Channell et al. (2018), on the other hand, describe the differences that arise from using the old ss09sea time scale (where accumulation rates are based on an empirical relationship with $\delta^{18}O$) instead of GICC05 (where they are based on the annual layer count) – so this does not apply to the record by Adolphi et al. (2018). And yes, as demonstrated by our results, using the GICC05 accumulation rates does make it more difficult to reconcile $^{14}C$ and $^{10}Be$ as compared to the ss09sea accumulation rates. As mentioned in lines 70-72, ice-core**

accumulation rates remain the largest source of systematic uncertainty in the $^{10}$Be-based production rate estimates. However, the largest systematic uncertainty in the calculation of accumulation rates comes from the correction of layer thinning through ice flow modelling, which is a slowly varying function of depth, and hence is relatively insensitive to minor corrections of the time scales themselves.

We have added the following text to line 288:

All ice cores were first placed on the same time scale (GICC05) before $^{10}$Be fluxes were calculated.

*22. Line 355: note again that this conflicts with the premise that the modern state is at equilibrium!*

**This comment has already been addressed in our response to comment #17.**

*23. Line 448: my intuition tells me that air-sea has exchange may have a small effect, but depending on the circulation state. Is it not possible that changes in air-sea exchange might combine non-linearly with particular changes in the circulation geometry?*

**As noted in lines 448-449, air-sea gas exchange has only a small effect on atmospheric $CO_2$ as compared to ocean circulation, given that the time scale of deep ocean ventilation (of the order of several hundred years to 1000 years or more) is much longer than the time scale of air-sea equilibration for $CO_2$ by gas exchange (approximately one year). In other words, the rate limiting step that determines the kinetics of the oceanic uptake of $CO_2$ is ocean circulation, not air-sea gas exchange. We will clarify this point in the third and fourth paragraphs of Sect. 3.1.3.**

In Sect. 3.1.3, we have included an explanation for why the increase in $\Delta^{14}C_{atm}$ induced by a change in the gas transfer velocity is not accompanied by a significant change in the atmospheric carbon inventory:

The air-sea equilibration time scale for $CO_2$ by gas exchange is about 1 year for a ~75-m thick surface mixed layer (Broecker and Peng, 1974), which is much smaller than the ventilation time scale for the deep ocean (on the order of several hundred years or more). One would therefore expect that the oceanic uptake of $CO_2$ demonstrates only a very small response to changes in $k_w$.

*24. Line 460: Although I see why the authors try to wiggle free from resolving the deglacial CO2 problem, I think it is it entirely possible to set it aside, and I also think it is basically not true that the study deals only with the glacial portion of the record. It is the glacial versus interglacial amplitude of atmospheric D14C that is of concern, and therefore the change across the deglaciation is entirely relevant! In fact, as suggested below, I would propose*

*provocatively that this study shows that atmospheric radiocarbon can be explained reasonably well up until the deglaciation, and that it is the modern radiocarbon activity that defies explanation. I wonder what the authors think of this contention.*

What we have tried to demonstrate with this work, especially by the analysis shown in Fig. 9, is that although models are able to reproduce successfully the high glacial $\Delta^{14}$C levels associated with the Laschamp (~41 kyr BP) event, it is very difficult to explain the persistence of relatively high $\Delta^{14}$C values after ~33 kyr BP, given available reconstructions of past changes in $^{14}$C production and extreme changes in the ocean carbon cycle. We think that this may be crucial for explaining the deglacial $\Delta^{14}$C decline, as the model representation of the mechanisms responsible for high glacial $\Delta^{14}$C levels will determine the carbon inventories of the different reservoirs prior to the deglacial $\Delta^{14}$C decline. And yes, the model fails to simulate the correct magnitude and timing of the deglacial $\Delta^{14}$C decline. But given that we did not attempt to reproduce accurately the observed glacial-interglacial variations in atmospheric $CO_2$ and $\Delta^{14}$C, this work seeks to highlight the persistent elevation of $\Delta^{14}$C after ~33 kyr BP as a major outstanding problem in our understanding of the atmospheric $\Delta^{14}$C record.

In other words, we can reach the amplitude of the Laschamp-related $\Delta^{14}$C change, but we cannot sustain the high levels during the last glacial nor can we get down low enough or fast enough during the last deglaciation.

*25. Line 537, the discussion of simulated B-Atm values: why do the authors not refer at all to published data for comparison? The compilation of Skinner et al. (2017) estimated, with the available data, that the global average ageing of the ocean at the LGM was 'only' ~689 14C years. This is relevant here, and indeed it would suggest that all of the model scenarios produce rather extreme outcomes as compared to available data.*

We agree comparison with measurement- and model-based estimates of radiocarbon reservoir age offsets from, e.g., Skinner et al. (2017) and Butzin et al. (2017), is a missed opportunity. It was a sacrifice made to reduce the length of an already very lengthy manuscript. Nonetheless, some intriguing points can be made by such a comparison, so we will incorporate it into Sect. 3.3 and Fig. 8.

Comparison of our LGM B-Atm age offset estimates from runs CIRC, VENT, and VENTx (range of 3682 to 3962 $^{14}$C years) with the compiled LGM marine radiocarbon data of Skinner et al. (2017) demonstrate that the carbon cycle scenarios are extreme, although it should be noted that they consider a wider depth range (~500 to 5000 m) of the ocean than we do. Skinner et al. (2017) predict a global average LGM B-Atm value of ~2048 $^{14}$C years, an increase of ~689 $^{14}$C years relative to preindustrial. Turning our comparison to surface reservoir ages, we note that our global average LGM surface reservoir age of ~1132 $^{14}$C years from runs VENT and VENTx is comparable to the ~1241 $^{14}$C years obtained by Skinner et al. (2017) for the LGM. The model-based estimates of surface reservoir age from Butzin et al. (2017) indicate a much lower LGM value of ~780 [14]C years, and values ranging from 540 to 1250 [14]C years between 50 and 25 kyr BP. Note that these estimates are based on model-simulated values between 50°N and 50°S. If the polar regions are included in the calculation (see Fig. 8c), their surface reservoir age estimates become comparable to our glacial values (range of 911 to 1354 [14]C years), and between about 34 and 22 kyr BP can exceed them, including even those from model runs VENT and VENTx, unless atmospheric $\Delta^{14}$C and $CO_2$ are prescribed (dashed colored lines in Fig. 8c). Interestingly, this is also roughly the time period where our deconvolutions of the IntCal13 and Hulu Cave $\Delta^{14}$C records give production rate estimates that are about 17.5 percent higher than the reconstructions, which indicates at the very least this is an important piece of the puzzle of the glacial-interglacial $\Delta^{14}$C problem, given that the effect of upper ocean stratification and/or sea ice on air-sea gas exchange is particularly important for surface reservoir ages.

[Figure]

Fig. 8. Modelled records of atmospheric (a) Δ¹⁴C and (b) CO₂, compared with their reconstructed histories (black and dark blue lines). Also shown are modelled records of the global average (c) surface reservoir age and (d) B-Atm ¹⁴C age offset, compared with a recent compilation of LGM marine radiocarbon data (dark blue squares) by Skinner et al.

**(2017) and model-based surface reservoir age estimates between 50°N and 50°S (solid black line) and across all latitudes (dashed black line) from Butzin et al. (2017), as well as (e) ideal age and (f) apparent oxygen utilization (AOU). Colored lines show the results of model runs using the mean paleointensity-based $^{14}$C production rate and the eight different carbon cycle scenarios described in Sect. 2.4 and Table 1. The gray envelope in (a) shows the uncertainty (2σ) from all production rate reconstructions and carbon cycle scenarios, providing a bounded estimate of Δ$^{14}$C change. The dashed colored lines in (c) show the surface reservoir age results from VENT and VENTx where atmospheric Δ$^{14}$C and $CO_2$ are prescribed. Radiocarbon ventilation ages are expressed here as radiocarbon reservoir age offsets following Soulet et al. (2016) which are used extensively by the radiocarbon dating community.**

We have added the observed and modelled records of Skinner et al. (2017) and Butzin et al. (2017), respectively, to Fig. 8d,c as well as added a paragraph to Sect. 3.3 that discusses the model-data comparison of B-Atm and surface R-age:

Driven by a reduction in ocean circulation, model run CIRC predicts a substantial increase in B-Atm during the last glacial, which is defined here as 40 to 18 kyr BP to avoid biasing global mean estimates toward Laschamp values. The global average glacial B-Atm predicted by CIRC is ~3225 $^{14}$C years, representing an increase in B-Atm of ~1599 $^{14}$C years relative to the preindustrial value of ~1626 $^{14}$C years. Model run VENT predicts a slightly larger increase in glacial B-Atm due to the inhibition of air-sea gas exchange. The "oldest" glacial waters are found in model run VENTx where air-sea gas exchange is severely restricted, yielding an increase in B-Atm of ~1912 $^{14}$C years (glacial B-Atm ~3538 $^{14}$C years). The glacial B-Atm values given by runs CIRC, VENT, and VENTx, as well as the ~717 year increase in ideal age during the last glacial relative to preindustrial, suggest that the glacial deep ocean was about two times older than its preindustrial counterpart. Comparison of our LGM B-Atm estimates (range of 3682 to 3962 $^{14}$C years) with the compiled LGM marine radiocarbon data of Skinner et al. (2017) demonstrate that the carbon cycle scenarios are extreme, although it should be noted that Skinner et al. consider a wider depth range (~500 to 5000 m) of the ocean than we do. Skinner et al. (2017) predict a global average LGM B-Atm value of ~2048 $^{14}$C years, an increase of ~689 $^{14}$C years relative to preindustrial. Turning our comparison to surface reservoir ages, we note that our global average LGM surface R-age of ~1132 $^{14}$C years from runs VENT and VENTx is comparable to the ~1241 $^{14}$C years obtained by Skinner et al. (2017) for the LGM. The model-based estimates of surface R-age from Butzin et al. (2017) indicate a much lower LGM value of ~780 $^{14}$C years, and values ranging from 540 to 1250 $^{14}$C years between 50 and 25 kyr BP. Note that these estimates are based on model-simulated values between 50°N and 50°S. If the polar regions are included in the calculation (see Fig. 8c), their surface R-age estimates become comparable to our glacial values (range of 911 to 1354 $^{14}$C years), and between about 34 and 22 kyr BP can exceed them, including even those from model runs VENT and VENTx, unless Δ$^{14}$C$_{atm}$ and $CO_2$ are prescribed (dashed colored lines in Fig. 8c) as in the simulation by Butzin et al. (2017).

*26. Line 545: I think it is worth specifying in what ways these indirect methods are also potentially inaccurate, due to different processes affecting e.g. oxygen and radiocarbon.*

**A comparison of modelled apparent oxygen utilization (AOU) with the model ocean's deep-water reservoir age (B-Atm age offset) is not meant to be taken as a direct comparison. The goal of showing the parallel occurrence of depleted ocean interior oxygen levels (i.e., increased AOU) was to provide the reader with additional (indirect) evidence that deep water ageing is occurring in the model runs that consider reductions in ocean circulation and air-sea gas exchange (e.g., scenarios CIRC, VENT, and VENTx). A significant reduction in deep ocean ventilation permits the enhanced accumulation of remineralized carbon in the ocean interior, and therefore the progressive consumption of dissolved oxygen, as well as an increase in the radiocarbon disequilibrium between the deep ocean and the atmosphere, due to a decrease in the rate of transport and mixing of younger (higher $\Delta^{14}C$) waters. These observations (increased AOU and increased B-Atm age offset) taken together suggest that deep water ageing is occurring. We will clarify this point further in the third and fourth paragraphs of Sect. 3.3.**

In Sect. 3.3, we have elaborated briefly on what the parallel occurrence of increased AOU and increased B-Atm indicates:

Indirect evidence for deep water ageing can be provided by the occurrence of depleted ocean interior oxygen levels, due to the progressive consumption of dissolved oxygen during organic matter remineralization in the water column. This situation is amplified by the slow escape of accumulating remineralized carbon in the ocean interior (see, e.g., Skinner et al., 2017), leading to higher values of apparent oxygen utilization ($AOU = O_{2,pre} - O_2$). These two concepts (increased AOU and increased B-Atm) taken together signal a significant reduction in deep ocean ventilation characterized by a decrease in the exchange rate between younger (higher $\Delta^{14}C$) surface waters and older ($^{14}C$-depleted), carbon-rich deep waters.

*27. Line 567: ..is a dedicated 'control knob', in the model.*

**We agree with the referee it would be prudent to clarify that air-sea gas exchange is a principal "control knob" governing atmospheric $\Delta^{14}C$ in a model framework.**

We have qualified this point in Sect. 3.3:

In contrast to a change in ocean circulation, air-sea gas exchange is a dedicated $\Delta^{14}C_{atm}$ "control knob" that can be invoked by models for a further increase of $\Delta^{14}C_{atm}$ without changing atmospheric $CO_2$.

28. Line 605: viewed as tentative, perhaps. The viewing is not tentative; the results are.

**Agreed.**

We have modified the text to read that the results must be "regarded as tentative".

*29. Line 676: is it worth stating by how much this polar bias would have to be in order to reconcile everything? Is that magnitude reasonable?*

**Interesting point. However, we would rather not discuss the polar bias further as we do not think that it can really reconcile everything. Firstly, the geomagnetic field reconstructions do not suffer from a polar bias and yet, cannot explain atmospheric $\Delta^{14}$C either. Secondly, as shown in Fig. 7c, the difference between reconstructed $\Delta^{14}$C and modelled $^{10}$Be (or RPI)-based $\Delta^{14}$C is changing over time and the largest changes of this difference occur between ~35 and 30 kyr BP and then during the last deglaciation, not during the Laschamp event as one might expect if these mismatches were due to a polar bias. Instead, production rates (as inferred from $^{10}$Be and RPI) were relatively stable across these two periods. Hence, it seems difficult to explain the mismatch by the presence of a polar bias alone.**

The text on line 676 has been modified to read:

If a polar bias was present, it would lead to an underestimation of the geomagnetic modulation of the ice-core $^{10}$Be flux, and therefore variations in the $^{10}$Be-based $^{14}$C production rate would also be underestimated. However, the mismatch of up to ~544 to 558 permil between reconstructed and modelled $^{10}$Be-based $\Delta^{14}$C$_{atm}$ during the last glacial (see Fig. 7c) appears to be much too large to be reconciled by considering uncertainties in the polar bias alone. Furthermore, this mismatch with reconstructed $\Delta^{14}$C$_{atm}$ is qualitatively similar when using paleointensity-based $^{14}$C production rates that do not suffer from a polar bias (Fig. 7c).

*30. Line 703: in this paragraph the realism of the implied sea ice changes is discussed, but again no mention is made of what existing marine radiocarbon data imply. These are really important constraints to mention, surely.*

**This comment has already been addressed in our response to comment #25.**

*31. Line 726: I couldn't help but feel that the conclusion of the study might be more hard hitting if we had a more specific 'shopping list' of things that could help to resolve this puzzle. For example, constraining the global marine radiocarbon inventory change across the deglaciation, estimating any gradient in cosmogenic nuclide production across latitudes (i.e. polar bias, perhaps from tropical ice cores?), estimates of global carbonate/POC export rates (which already exist incidentally; Cartapanis et al., 2016; 2018), etc...*

**What may help to resolve the glacial radiocarbon problem is progress in several different areas. Additional records of glacial atmospheric $\Delta^{14}$C would help to further refine the IntCal $\Delta^{14}$C record. Cosmogenic isotope production records may be improved, e.g., by refining**

estimates of ice accumulation, by developing a better understanding of [10]Be transport and deposition during the glacial, by recovering additional long and continuous records from Antarctic ice cores and including marine [10]Be records, and by obtaining additional geomagnetic data. An expanded spatiotemporal coverage of $\Delta^{14}$C of DIC in the surface and deep ocean would allow one to narrow the time scales of surface-to-deep transport and air-sea equilibration of $\Delta^{14}$C, carbon and nutrients, and thereby guide model-based analyses. Models should be improved to better represent the glacial cycles of carbon and radiocarbon, by taking into account exchange with sediments and the lithosphere, by better representing coastal processes, and by representing a wide variety of paleo proxies such as $\delta^{13}$C, Nd isotopes, carbonate ion concentration, lysocline evolution, and biological productivity proxies in a 3-D dynamic context. What is also missing are methods to quantify how the global ocean carbon inventory, which co-determines the [14]C/C ratio and thus $\Delta^{14}$C value in the ocean, has changed over the last 50,000 years.

We have added the following text to the last paragraph in the summary and conclusions section:

Progress in several different areas may help to resolve the glacial-interglacial radiocarbon problem. Additional records of glacial $\Delta^{14}$C$_{atm}$ would help refine the older portion of the IntCal $\Delta^{14}$C record. Cosmogenic isotope production records may be improved, e.g., by refining estimates of ice accumulation, by developing a better understanding of [10]Be transport and deposition during the glacial, by recovering additional long and continuous records from Antarctic ice cores and including marine [10]Be records, and by obtaining additional geomagnetic field data. An expanded spatiotemporal observational coverage of $\Delta^{14}$C of DIC in the surface and deep ocean would help narrow the time scales of surface-to-deep transport and air-sea equilibration of $\Delta^{14}$C, carbon and nutrients, and thereby guide model-based analyses. Models should become more sophisticated and detailed in order to reproduce successfully the glacial-interglacial changes in carbon and radiocarbon, by including exchange with sediments and the lithosphere and by better representing coastal processes, and by representing a wide variety of paleo proxies such as $\delta^{13}$C, Nd isotopes, carbonate ion concentration, lysocline evolution, and paleo-productivity proxies in a 3-D dynamic context for model evaluation. What is also missing are methods to quantify how the ocean carbon inventory, which co-determines the [14]C/C ratio and thus the $\Delta^{14}$C values in the ocean and atmosphere, has changed over the last 50,000 years.

*32. Table 1: it would be helpful to specify here which simulations have active sediments included. Incidentally, why was the rain ratio changed in one simulartion?*

**As mentioned in lines 209-210 of the original manuscript, all transient simulations are performed with Bern3D model configuration ALL, which is the atmosphere–ocean–land–sediment model configuration. Hence, transient simulations include the 10-layer sediment model of Heinze et al. (1999) and Gehlen et al. (2006). We will clarify this point in Table 1 caption.**

As discussed in lines 267-276 and summarized in Table 1, the $CaCO_3$-to-POC export ratio was changed over time in model scenarios BIO, PHYS-BIO, and PHYS-BIOx in order to investigate the impact of biological carbon pump changes on atmospheric $\Delta^{14}C$. While changes in the $CaCO_3$-to-POC export ratio are important for achieving glacial atmospheric $CO_2$ drawdown, our model results demonstrate that biogeochemical changes alone (scenario BIO) do not lead to an improved simulation of high glacial $\Delta^{14}C$ levels as compared to model runs invoking only physical changes (i.e., changes in ocean circulation and/or air-sea gas exchange). This is clearly illustrated by Fig. 8 and 9.

We have modified the Table 1 caption so that it is clear that the Bern3D model configuration with sediments was used for the transient historical simulations:

In all scenarios, the fully coupled model configuration, including the major global carbon reservoirs (atmosphere, terrestrial biosphere, ocean, and sediments), is used.

*33. Fig 3, caption: I think it is more mathematically correct to state <100m and >1500m, no?*

**Yes, this is a typo that will be corrected in a revised manuscript.**

This typo has been corrected in the revised manuscript.

*34. Fig 7, caption, line 1203: I think it would be helpful to state ". . .using the mean reconstructed palaeointensity.."*

**We agree it would be more precise to state that RPI-based $\Delta\Delta^{14}C$ is the difference between reconstructed $\Delta^{14}C$ and model-simulated $\Delta^{14}C$ based on the mean RPI-based $^{14}C$ production rate.**

We have revised the Fig. 7c caption to:

Difference between reconstructed $\Delta^{14}C$ and model-simulated $\Delta^{14}C$ using averaged paleointensity data (RPI-based $\Delta\Delta^{14}C$; gray) and the ice-core $^{10}Be$ data of Adolphi et al. (2018) ($^{10}Be$-based $\Delta\Delta^{14}C$; purple), compared with the atmospheric $CO_2$ record (red).

*35. Fig 8: shouldn't all the simulated D14Catm traces start at the same value and end at different values? Although this might look nasty, it suggests a different outlook in my view. Incidentally, the outputs in plots c and d are obvious candidates for comparison with existing data (e.g. Skinner et al., 2019, 20176), perhaps for a future study if not this one.*

**Since different carbon cycle scenarios (and therefore processes) were used to force the model into a glacial state over a 50,000-year integration, during which the glacial drawdown of atmospheric $CO_2$ was achieved, the model runs start from different global**

$^{14}$C/C distributions, and therefore different values of atmospheric $\Delta^{14}$C, at 70 kyr BP. The analysis presented in Fig. 9 effectively normalizes the various $\Delta^{14}$C records so that they are comparable, using two different "corrections".

We have now added the observed data of Skinner et al. (2017) and model results of Butzin et al. (2017), respectively, to Fig. 8d,c.

*36. Fig 9: this is a fascinating figure, though I find it slightly problematic. First, what is the rationale for normalizing to the average D14Catm value 0-50ka? I think that plots a and b should be replaced with normalization to the final 'modern' value, and that plots c and d should be extended up to the present. The latter is surely important, as it shows how we (well, you!) can do a pretty good job at simulating the amplitude of D14Catm change in the glacial when tweaking all the model's knobs, but that we can't subsequently get the deglacial change to the modern value, just as we can't quite get the deglacial change in CO2. I feel this must be significant... I wonder what the authors think.*

The reason for subtracting the mean value from the $\Delta^{14}$C records shown in Fig. 9a,b was to remove the offset/trend and emphasize the fluctuations in the $\Delta^{14}$C data about the overall trend. This is effectively an offset correction normalization. Here, we can see that none of the model runs are able to sustain the high $\Delta^{14}$C levels after the Mono Lake excursion or capture the sharp decline in $\Delta^{14}$C during the last deglaciation. We agree with the referee that the $\Delta^{14}$C records shown in Fig. 9c,d should be extended up to 0 kyr BP.

We do not think that showing $\Delta^{14}$C anomalies relative to the last millennium $\Delta^{14}$C value would provide much new information compared to the existing Fig. 7 and 8. Simulated modern $\Delta^{14}$C values shown in Fig. 7 and 8 are relatively close to observed last millennium values and the remaining discrepancy is small compared to the model-data mismatch during the last glacial period.

[Figure]

**Fig. 9.** Comparison of atmospheric Δ14C variability caused by changes in the ocean carbon cycle (b, d) with production-driven changes in atmospheric Δ14C using scenario MOD (a, c). For the analysis of carbon cycle changes, only the results of model runs using the mean paleointensity-based 14C production rate are shown. The Δ14C records in the upper panel (a, b) have been detrended by removing the mean, whereas the lower panel (c, d) shows Δ14C anomalies expressed as differences relative to the Δ14C value at 50 kyr BP. Three vertical light gray bars indicate the Laschamp (~41 kyr BP) and Mono Lake (~34 kyr BP) geomagnetic excursions, and the last glacial termination (~18 to 11 kyr BP).

We have extended the time axis of Fig. 9c,d to 0 kyr BP.

*37. Figure 10 and 11: I would suggest including a narrow plot at the base of each of these showing the offsets between simulated and observed values over time.*

**This is a difficult comparison to make as there is no one true (correct) target value. Nonetheless, we agree that such a comparison would allow the reader to more easily visualize the time periods where disagreement between the deconvolution- and measurement-based production rate estimates is highest, i.e., between 32 and 22 kyr BP.**

[Figure]

**Fig. 10.** Comparison of $^{14}$C production rate estimates inferred from a deconvolution of the atmospheric $\Delta^{14}$C record and from paleointensity and ice-core $^{10}$Be data. (a) $^{14}$C production rate calculated as the sum of the modelled air-sea and atmosphere-land $^{14}$CO$_2$ fluxes and the reconstructed change in the atmospheric $^{14}$C inventory and loss of $^{14}$C due to radioactive decay (see Eq. [2]). Model-based $^{14}$CO$_2$ fluxes were obtained by forcing the Bern3D carbon cycle model with reconstructed variations in atmospheric $\Delta^{14}$C and CO$_2$ as well as seven different carbon cycle scenarios. Results of model runs using the IntCal13 calibration curve are shown in the light blue envelope (2σ), whereas the light red envelope (2σ) shows the results from simulations using the composite Hulu Cave (10.6 to 50 kyr BP) and IntCal13 (0 to 10.6 kyr BP) $\Delta^{14}$C record. The heavy black line is the mean of five available production rate reconstructions: Laj et al. (2000), Laj et al. (2004), Nowaczyk et al. (2013), Channell et al. (2018), and Adolphi et al. (2018). (b) Difference between the mean of the measurement-based production rate estimates (heavy black line) and estimates based on the

**deconvolution of the IntCal13 (IntCal-based ΔQ; blue) and Hulu Cave (Hulu-based ΔQ; red) Δ$^{14}$C data.**

We have added a subplot to Fig. 10 showing the difference between the mean measurement-based $^{14}$C production rate and our new estimates based on the deconvolution of the IntCal13 and Hulu Cave Δ$^{14}$C records, and updated its caption to better describe the model-based deconvolution approach.

[revised manuscript text omitted]

---

## Referee Report (RR1)

**Review on**

*Mysteriously high $\Delta^{14}C$ of the glacial atmosphere: Influence of $^{14}C$ production and carbon cycle changes*

**by A Dinauer et al**

submitted to *Climate of the Past*, article reference: cp-2019-159

**Date: May 20, 2020**

I think the authors have done a great job in their respones to the reviews and in their submitted revision.

Below are some replies to their response on the reviewers, which might need some fine tuning:

1. Reviewer 1, fundamental points on weathering and sedimentation. I understand that you take the fluxes that leave to the ocean to the sediment as those which need to be put back in the ocean as weathering, sidestepping the atmosphere, as you said. Therefore, we find a closed loop of ocean-lithosphere interaction in Figure 2.

   The given stoichiometric ratios of Si:Alk and C:Alk for the given equations however do not make sense to me, since the atmosphere from which $CO_2$ comes from, is ignored. For example, for carbonate weathering it is said C:Alk is 1:2, but the input into the ocean is 2 $HCO_3^-$, thus 2 C and 2 Alk, which I would consider as 1:1 ratio in C:Alk. Or do I have to understand these given ratios, that you indeed put the losses in the deep ocean (here $CaCO_3$ or $CO_3^{2-}$ with a C:Alk of 1:2) as such in the weathering fluxes to the surface ocean? If so, it would imply that you do not put $HCO_3^-$ as weathering fluxes in the ocean, but $CO_3^{2-}$, which needs different wording, (e.g. compensation fluxes for sedimentary loss). In this respect in lines 205-206 and 868-869, there is something misleading: You mention there is an alkalinity input of 34.37 Tmol $HCO_3^-$/yr. However, the input of $HCO_3^-$ is a change in both alkalinity and DIC, so it is not clear to me if this fluxes is already included in (or in addition to) the DIC flux of 0.46 GtC/yr (which equals 38.33 TmolC/yr). However, maybe all is correct in a condensed sense which implicitly assumes the effect of the sidestepped

atmosphere, but it is maybe worth to rethink these statement. Since this discussion is only contained in the rebuttal, but not in the revised draft, it would probably not change the draft, but a clarification might nevertheless be helpful. Maybe you then also reconsider, that some of what is said here might be added to the draft (e.g. the Appendix with the model description).

2. Reviewer 1, minor point 1: It is not enough to mention in the caption of Fig 1, that the considered 14C half-life of 5700 years has been accounted for, this also needs to be stated in the main text, when Intcal13 and Hulu Cave 14C are introduced.

3. Reviewer 1, minor point 5: Considering to cite the underlying ice core papers (suggested by the reviewer) was refering to the papers with the data, not with the spline method, as done now by citing Enting (1987). In Köhler et al. (2017) the data availablity section ends with "When using these data, please consider citing the original publications from which the data underlying this compilation have been taken.", which would imply for the $CO_2$ spline of the last 55 kyr something like the following: The spline combines raw data from Talos Dome, Siple Dome, WAIS Divide, EPICA Dome C, EPICA Dronning Maud Land and Law Dome each on the most recent age scale, e.g. AICC2012, GICC05, WD2014. The relevant ice core data papers underlying the spline (Ahn et al., 2012; Ahn and Brook, 2014; Bauska et al., 2015; Bereiter et al., 2012; Lüthi et al., 2010; MacFarling-Meure et al., 2006; Marcott et al., 2014; Monnin et al., 2001, 2004; Rubino et al., 2013) and those related to the age models (Buizert et al., 2015; Sigl et al., 2016; Veres et al., 2013) are those which have been suggested to be mentioned by the reviewer initially.

**References**

[revised manuscript text omitted]